# Online Composite Optimization Between Stochastic and Adversarial Environments

**Yibo Wang**[1,2], **Sijia Chen**[1,2], **Wei Jiang**[1], **Wenhao Yang**[1,2], **Yuanyu Wan**[3,1], **Lijun Zhang**[1,2,*]

[1]National Key Laboratory for Novel Software Technology, Nanjing University, Nanjing, China
[2]School of Artificial Intelligence, Nanjing University, Nanjing, China
[3]School of Software Technology, Zhejiang University, Ningbo, China
{wangyb, chensj, jiangw, yangwh, zhanglj}@lamda.nju.edu.cn, wanyy@zju.edu.cn

## Abstract

We study online composite optimization under the Stochastically Extended Adversarial (SEA) model. Specifically, each loss function consists of two parts: a fixed non-smooth and convex regularizer, and a time-varying function which can be chosen either stochastically, adversarially, or in a manner that interpolates between the two extremes. In this setting, we show that for smooth and convex time-varying functions, optimistic composite mirror descent (OptCMD) can obtain an $\mathcal{O}(\sqrt{\sigma_{1:T}^2} + \sqrt{\Sigma_{1:T}^2})$ regret bound, where $\sigma_{1:T}^2$ and $\Sigma_{1:T}^2$ denote the cumulative stochastic variance and the cumulative adversarial variation of time-varying functions, respectively. For smooth and strongly convex time-varying functions, we establish an $\mathcal{O}((\sigma_{\max}^2 + \Sigma_{\max}^2) \log(\sigma_{1:T}^2 + \Sigma_{1:T}^2))$ regret bound, where $\sigma_{\max}^2$ and $\Sigma_{\max}^2$ denote the maximal stochastic variance and the maximal adversarial variation, respectively. For smooth and exp-concave time-varying functions, we achieve an $\mathcal{O}(d \log(\sigma_{1:T}^2 + \Sigma_{1:T}^2))$ bound where $d$ denotes the dimensionality. Moreover, to deal with the unknown function type in practical problems, we propose a multi-level *universal* algorithm that is able to achieve the desirable bounds for three types of time-varying functions simultaneously. It should be noticed that all our findings match existing bounds for the SEA model without the regularizer, which implies that there is *no price* in regret bounds for the benefits gained from the regularizer.

## 1   Introduction

Online composite optimization has drawn considerable attention in recent years [Duchi and Singer, 2009, Ghadimi and Lan, 2012, Lei and Zhou, 2017, Scroccaro et al., 2023]. Formally, it can be viewed as an iterative game between a learner and the environment. In each round $t$, the learner makes a decision $\mathbf{x}_t$ from a convex set $\mathcal{X} \subseteq \mathbb{R}^d$ and then suffers a loss $\phi_t(\mathbf{x}_t)$ in the form of

$$\phi_t(\mathbf{x}) = f_t(\mathbf{x}) + r(\mathbf{x}), \tag{1}$$

where $f_t(\cdot) : \mathcal{X} \to \mathbb{R}$ denotes the time-varying function that is chosen by the environment, and $r(\cdot) : \mathcal{X} \to \mathbb{R}$ denotes the fixed non-smooth and convex regularizer, such as the $\ell_1$-norm for sparse vectors and the trace norm for low-rank matrices [Langford et al., 2009, Flammarion and Bach, 2017, Zhang et al., 2019, Garber and Kaplan, 2019]. In the literature, online composite optimization is generally divided into two categories: stochastic composite optimization [Lan, 2012, 2016, Zhang et al., 2017, Lei and Tang, 2018, Lei et al., 2019, Kulunchakov and Mairal, 2019] and adversarial composite optimization [Xiao, 2009, Duchi et al., 2010, Mohri and Yang, 2016, Joulani et al., 2020, Yang et al., 2024c]. In the former one, $f_t(\cdot)$ is assumed to be independent and identically distributed (i.i.d.) over time; in the latter one, $f_t(\cdot)$ can be chosen arbitrarily or even adversarially. However,

---

[*]Lijun Zhang is the corresponding author.

38th Conference on Neural Information Processing Systems (NeurIPS 2024).

the environment in real-world scenarios is seldom purely stochastic or adversarial, but rather falls somewhere in between [Amir et al., 2020, Garber et al., 2020, Sherman et al., 2021, Zimmert and Seldin, 2021, Ito, 2021], and our understanding for the more common intermediate scenarios remains limited in online composite optimization.

Recently, Sachs et al. [2022] introduce the Stochastically Extended Adversarial (SEA) model as an intermediate setting in online optimization without regularizer, i.e., $r(\mathbf{x}) = 0$. In the SEA model, loss functions are not restricted to fully i.i.d. or adversarial; instead, the environment selects them from any intermediate state between the two extreme settings. To reflect how stochastic or adversarial the environments are, they introduce the cumulative stochastic variance $\sigma_{1:T}^2$ and the cumulative adversarial variation $\Sigma_{1:T}^2$, as shown below:

$$\sigma_{1:T}^2 = \mathbb{E}\left[\sum_{t=1}^{T} \sigma_t^2\right] \text{ and } \Sigma_{1:T}^2 = \mathbb{E}\left[\sum_{t=1}^{T} \sup_{\mathbf{x} \in \mathcal{X}} \|\nabla F_t(\mathbf{x}) - \nabla F_{t-1}(\mathbf{x})\|_2^2\right], \qquad (2)$$

where $\sigma_t^2 = \sup_{\mathbf{x} \in \mathcal{X}} \mathbb{E}_{f_t \sim \mathcal{D}_t}[\|\nabla f_t(\mathbf{x}) - \nabla F_t(\mathbf{x})\|_2^2]$ denotes the variance of gradients and $F_t(\mathbf{x}) = \mathbb{E}_{f_t \sim \mathcal{D}_t}[f_t(\mathbf{x})]$ denotes the expected function for $f_t(\cdot)$. For the SEA model, two classical algorithms in optimistic online learning [Rakhlin and Sridharan, 2013]—optimistic Follow-The-Regularized Leader (FTRL) [Sachs et al., 2022] and optimistic Online Mirror Descent (OMD) [Chen et al., 2023]—have been proven ensuring sublinear regret bounds with respect to both $\sigma_{1:T}^2$ and $\Sigma_{1:T}^2$.

In this paper, we extend the SEA model into online composite optimization, termed as composite SEA, to bridge the gap between stochastic and adversarial composite optimization. Specifically, in (1), $r(\cdot)$ remains fixed over time and $f_t(\cdot)$ is selected from a distribution $\mathcal{D}_t$, which is chosen by the environments either stochastically, adversarially, or in a manner that interpolates between the two extremes. The goal of composite SEA is to minimize the expected regret in terms of (1):

$$\mathbb{E}\left[\text{Regret}_T\right] = \mathbb{E}\left[\sum_{t=1}^{T} [f_t(\mathbf{x}_t) + r(\mathbf{x}_t)] - \min_{\mathbf{x} \in \mathcal{X}} \sum_{t=1}^{T} [f_t(\mathbf{x}) + r(\mathbf{x})]\right], \qquad (3)$$

which benchmarks the cumulative composite loss of the learner and that of the best fixed decision. To handle the composite setting, a natural impulse is to treat $f_t(\cdot) + r(\cdot)$ as one function and directly apply existing methods for the SEA model, i.e., optimistic FTRL and optimistic OMD. However, such a straightforward application is unsuitable because (i) these methods heavily rely on the smoothness of loss functions, but due to the non-smooth component $r(\cdot)$, the summation function $\phi_t(\cdot) = f_t(\cdot) + r(\cdot)$ loses this crucial property; (ii) directly applying these methods ignores the presence of $r(\cdot)$ and thus fails to gain the benefits from the regularizer, e.g., the sparsity induced by the $\ell_1$-norm. Therefore, a natural question arises *whether it is possible to deal with both the intermediate nature of environments and the composite structure of loss functions concurrently.*

We affirmatively answer the above question by revisiting a variant of optimistic OMD [Scroccaro et al., 2023], named Optimistic Composite Mirror Descent (OptCMD). This method inherits the idea of optimistic online learning [Rakhlin and Sridharan, 2013], i.e., exploiting estimates on upcoming loss functions for decision updates, and can disentangle $f_t(\cdot)$ and $r(\cdot)$ during the optimization, thereby effectively leveraging distinct properties of each. However, OptCMD is originally designed for adversarial composite optimization, and a simple extension will lead to unsatisfactory bounds in composite SEA. In this paper, we reanalyze OptCMD and show that with suitable configurations, OptCMD attains $\mathcal{O}(\sqrt{\sigma_{1:T}^2} + \sqrt{\Sigma_{1:T}^2})$, $\mathcal{O}((\sigma_{\max}^2 + \Sigma_{\max}^2)\log(\sigma_{1:T}^2 + \Sigma_{1:T}^2))$ and $\mathcal{O}(d\log(\sigma_{1:T}^2 + \Sigma_{1:T}^2))$ bounds for smooth and general convex, smooth and strongly convex, and smooth and exp-concave time-varying functions, respectively. Our findings generalize previous results in stochastic and adversarial composite optimization, and can reduce to them by specializing $\sigma_{1:T}^2$ and $\Sigma_{1:T}^2$. Moreover, our results coincide with existing bounds for the SEA model [Sachs et al., 2022, Chen et al., 2023], which indicates that there is *no price* in regret bounds for the benefits from $r(\cdot)$.

One concern of OptCMD is the requirement of the function type in advance, which is often impractical in real-world problems and motivates us to design a *universal* algorithm that is agnostic to the prior knowledge about loss functions. We note that recently, Yan et al. [2023] have introduced a universal algorithm for the SEA model without regularizer, based on the meta-expert structure in online learning [van Erven and Koolen, 2016, Mhammedi et al., 2019, Wang et al., 2019, Zhang et al., 2022]. However, their method [Yan et al., 2023] cannot naturally support the composite setting, as it highly depends on the smoothness of the (summation) loss functions, which does not necessarily hold in the composite SEA model. In this paper, we propose a *novel* universal algorithm for composite SEA,

based on the observations that (i) the regularizer remains fixed over time and is accessible to the meta-algorithm from the beginning round; (ii) the meta-algorithm is able to obtain expert decisions for the current round before estimating their performance. With these two facts, we can utilize the information about $r(\cdot)$ to track experts, ensuring the meta-regret will not deviate from that of the best one. Specifically, inspired by Yan et al. [2023], we employ a two-layer Multi-scale Multiplicative-weight with Correction (MsMwC) [Chen et al., 2021] as the meta-algorithm, and choose OptCMD as the expert-algorithm. To effectively estimate the expert performance, we first collect decisions of each expert for the current round, and then explicitly integrate them and expert performance on $r(\cdot)$ into the losses and optimisms used in MsMwC. Theoretical analysis demonstrates that, with the proposed integration, our universal algorithm can achieve desirable bounds for three types of loss functions simultaneously. We summarize our contributions as shown below.

- We first introduce the composite SEA model, which serves as an intermediate setting between stochastic and adversarial composite optimization and can naturally adapt to two extreme settings;
- For the composite SEA model, we demonstrate that OptCMD can attain the regret bounds of $\mathcal{O}(\sqrt{\sigma_{1:T}^2} + \sqrt{\Sigma_{1:T}^2})$, $\mathcal{O}((\sigma_{\max}^2 + \Sigma_{\max}^2)\log(\sigma_{1:T}^2 + \Sigma_{1:T}^2))$ and $\mathcal{O}(d\log(\sigma_{1:T}^2 + \Sigma_{1:T}^2))$ for smooth and general convex, smooth and strongly convex, and smooth and exp-concave time-varying functions, respectively. Owing to the versatility of $\sigma_{1:T}^2$ and $\Sigma_{1:T}^2$, these bounds can *recover* previous results in stochastic and adversarial composite optimization;
- Moreover, to handle the unknown function type, we propose a *new* universal algorithm in composite SEA and show that it can concurrently achieve desirable bounds for three kinds of functions;
- Finally, we discuss implications of our theoretical findings for two common intermediate examples with composite loss functions, and derive favorable bounds specific to these practical scenarios.

## 2 Related work

In this section, we briefly review related work on adversarial and stochastic composite optimization, the SEA model and universal online learning.

**Adversarial composite optimization.** In this setting, $f_t(\cdot)$ is assumed to be chosen by the environments arbitrarily or even adversarially, and the performance is measured by *regret* shown in brackets of (3). In the literature, the seminal work [Duchi and Singer, 2009] proposes the FOBOS method with $\mathcal{O}(\sqrt{T})$ and $\mathcal{O}(\lambda^{-1}\log T)$ regret bounds for general convex and $\lambda$-strongly convex $f_t(\cdot)$, respectively. Later, Xiao [2009] proposes the RDA method based on the primal-dual subgradient framework [Nesterov, 2009], and shows that RDA is able to generate sparser decisions than FOBOS when $r(\mathbf{x}) = \mu\|\mathbf{x}\|_1$ for some $\mu > 0$. In the same time, another subsequent work [Duchi et al., 2010] introduces the COMID method under the mirror descent framework [Beck and Teboulle, 2003], and achieves the same bounds as FOBOS. Recently, for $\alpha$-exp-concave $f_t(\cdot)$, Yang et al. [2024c] establish an $\mathcal{O}((d/\alpha)\log T)$ regret bound by proposing the ProxONS method.

Besides, there exist other powerful methods [Yang et al., 2014, Mohri and Yang, 2016, Joulani et al., 2020, Scroccaro et al., 2023] equipped with problem-dependent bounds. These bounds can safeguard the above results in the worst-case scenarios and become tighter when environments have special properties, such as smoothness. Among them, the most related work is by Scroccaro et al. [2023], who develop OptCMD based on the optimistic online learning framework [Rakhlin and Sridharan, 2013]. The key idea is to make an estimate for the upcoming loss function, which ensures tighter bounds when the estimate is accurate and still maintains the worst-case bound otherwise. By utilizing the smoothness of $f_t(\cdot)$, OptCMD achieves $\mathcal{O}(\sqrt{V_T})$ and $\mathcal{O}(\lambda^{-1}\log V_T)$ regret bounds for general convex and $\lambda$-strongly convex $f_t(\cdot)$, respectively, where $V_T = \sum_{t=1}^{T}\max_{\mathbf{x}\in\mathcal{X}}\|\nabla f_t(\mathbf{x}) - \nabla f_{t-1}(\mathbf{x})\|_2^2$ denotes the gradient-variation and can be small when $f_t(\cdot)$ gradually changes.

**Stochastic composite optimization.** In this setting, $f_t(\cdot)$ is assumed to be i.i.d. sampled from a fixed distribution $\mathcal{D}$, and the goal is to minimize the composite objective: $\min_{\mathbf{x}\in\mathcal{X}}\mathbb{E}_{f\sim\mathcal{D}}[f(\mathbf{x})] + r(\mathbf{x})$. The performance is measured by *excess risk*, which compares the solution with the optimal one, i.e., $\mathbb{E}_{f\sim\mathcal{D}}[f(\mathbf{x}_T)] + r(\mathbf{x}_T) - \min_{\mathbf{x}\in\mathcal{X}}\{\mathbb{E}_{f\sim\mathcal{D}}[f(\mathbf{x})] + r(\mathbf{x})\}$.

In the literature, there are a substantial body of methods designed for the stochastic setting [Ghadimi and Lan, 2012, Lan, 2012, 2016, Lei and Tang, 2018, Lei et al., 2019], and by utilizing the online-to-batch conversion technique [Cesa-Bianchi et al., 2004], existing methods for adversarial composite optimization [Xiao, 2009, Duchi et al., 2010, Yang et al., 2024c] can also be extended to the stochastic

scenarios. Specifically, Lan [2012] first establishes an $\mathcal{O}(1/\sqrt{T})$ excess risk bound for general convex objective, and then Lan [2016] improves the rate to $\mathcal{O}(1/T)$ for strongly convex objective. By utilizing the online-to-batch conversion, both RDA [Xiao, 2009] and COMID [Duchi et al., 2010] achieve $\mathcal{O}(1/\sqrt{T})$ and $\mathcal{O}(\log T/(\lambda T))$ excess risks for general convex and $\lambda$-strongly convex objective, respectively. When $f(\cdot)$ is $\alpha$-exp-concave, ProxONS [Yang et al., 2024c] ensures an $\mathcal{O}(d \log T/(\alpha T))$ excess bound.

**SEA model.** The SEA model is originally introduced by Sachs et al. [2022] as an intermediate setting in online optimization without regularizer. Moreover, they also propose two versatile quantities $\sigma_{1:T}^2$ and $\Sigma_{1:T}^2$ to reflect the stochastic and adversarial aspect of the environments, respectively. Theoretically, for the SEA model, Sachs et al. [2022] prove that optimistic FTRL enjoys the bounds of $\mathcal{O}(\sqrt{\sigma_{1:T}^2} + \sqrt{\Sigma_{1:T}^2})$ and $\mathcal{O}(\lambda^{-1}(\sigma_{\max}^2 + \Sigma_{\max}^2) \log T)$ for smooth and general convex, and smooth and $\lambda$-strongly convex loss functions, respectively. Later, Chen et al. [2023] demonstrate that optimistic OMD is able to attain the same bound for general convex losses and an improved $\mathcal{O}(\lambda^{-1}(\sigma_{\max}^2 + \Sigma_{\max}^2) \log(\sigma_{1:T}^2 + \Sigma_{1:T}^2))$ bound for the strongly convex losses. Moreover, they also establish a new $\mathcal{O}((d/\alpha) \log(\sigma_{1:T}^2 + \Sigma_{1:T}^2))$ bound for smooth and $\alpha$-exp-concave functions.

**Universal online learning.** The universal online learning is proposed to handle the uncertainty of the loss function types, when applying online algorithms to practical optimization problems. The center to universal algorithms is the powerful meta-expert structure [van Erven and Koolen, 2016, Mhammedi et al., 2019, van Erven et al., 2021], which is also widely-used in many other fields of online learning [Daniely et al., 2015, Jun et al., 2017a,b, Zhang et al., 2018, 2020, 2021, Cutkosky, 2020, Wan et al., 2021, 2022b, Wang et al., 2024, Wan et al., 2024a]. The key idea of meta-expert structure is to maintain multiple experts to process different types of loss functions and then, deploy a meta-algorithm to combine the decisions from experts [Wang et al., 2019, 2020, van Erven et al., 2021, Zhang et al., 2022, Yang et al., 2024a,b]. In the literature, the most related work is Yan et al. [2023], who propose a novel universal algorithm with a two-layer MSMWC [Chen et al., 2021] as the meta-algorithm. By maintaining multiple instances of optimistic OMD, their method can adapt to the SEA model and deliver the same bounds as those in Chen et al. [2023]. Recently, Yan et al. [2024] further improve this method by employing a simpler meta-algorithm while achieving optimal regret bounds for three types of functions. However, it should be noticed that these methods [Yan et al., 2023, 2024] heavily relies on the smoothness of the summation function (1), and thus cannot handle the composite structure in loss functions well, as previously discussed.

## 3 Preliminaries

In this section, we introduce some preliminaries, including standard assumptions and a brief review of OptCMD [Scroccaro et al., 2023].

### 3.1 Assumptions

We first list the common assumptions in prior studies [Duchi et al., 2010, Chen et al., 2023, 2024].

**Assumption 1.** *The convex decision set $\mathcal{X}$ belongs to an Euclidean ball with the diameter $D$.*

**Assumption 2.** *At each round $t$, the random function $f_t(\cdot)$ is $G$-Lipschitz over $\mathcal{X}$, i.e.,*
$$\forall \mathbf{x}, \mathbf{y} \in \mathcal{X}, \ |f_t(\mathbf{x}) - f_t(\mathbf{y})| \leq G \|\mathbf{x} - \mathbf{y}\|_2.$$

**Assumption 3.** *The regularized function $r(\cdot)$ is convex and bounded over $\mathcal{X}$, i.e.,*
$$\forall \mathbf{x}, \mathbf{y} \in \mathcal{X}, \ r(\mathbf{y}) \geq r(\mathbf{x}) + \nabla r(\mathbf{x})^\top (\mathbf{y} - \mathbf{x}) \text{ and } \forall \mathbf{x} \in \mathcal{X}, \ 0 \leq r(\mathbf{x}) \leq C.$$

**Assumption 4.** *At each round $t$, the expected function $F_t(\cdot)$ is $H$-smooth over $\mathcal{X}$, i.e.,*
$$\forall \mathbf{x}, \mathbf{y} \in \mathcal{X}, \ \|\nabla F_t(\mathbf{x}) - \nabla F_t(\mathbf{y})\|_2 \leq H \|\mathbf{x} - \mathbf{y}\|_2.$$

**Assumption 5.** *At each round $t$, the expected function $F_t(\cdot)$ is convex over $\mathcal{X}$, i.e.,*
$$\forall \mathbf{x}, \mathbf{y} \in \mathcal{X}, \ F_t(\mathbf{y}) \geq F_t(\mathbf{x}) + \nabla F_t(\mathbf{x})^\top (\mathbf{y} - \mathbf{x}).$$

**Assumption 6.** *At each round $t$, the expected function $F_t(\cdot)$ is $\lambda$-strongly convex over $\mathcal{X}$, i.e.,*
$$\forall \mathbf{x}, \mathbf{y} \in \mathcal{X}, \ F_t(\mathbf{y}) \geq F_t(\mathbf{x}) + \nabla F_t(\mathbf{x})^\top (\mathbf{y} - \mathbf{x}) + (\lambda/2)\|\mathbf{x} - \mathbf{y}\|_2^2.$$

---
**Algorithm 1** Optimistic Composite Mirror Descent (OptCMD)
---
**Initialization**: Let $\mathbf{x}_1 = \hat{\mathbf{x}}_1$ be an arbitrary point in $\mathcal{X}$.

1: **for** $t = 1$ to $T$ **do**
2:     Submit $\mathbf{x}_t$, and the environment chooses a distribution $\mathcal{D}_t$
3:     Suffer the composite loss $f_t(\mathbf{x}_t) + r(\mathbf{x}_t)$, where $f_t(\cdot)$ is sampled from $\mathcal{D}_t$
4:     Update $\hat{\mathbf{x}}_{t+1}$ and $\mathbf{x}_{t+1}$ according to (4) and (5), respectively
5: **end for**
---

**Assumption 7.** *All the variance of the gradients and the adversarial variations are bounded, i.e.,*

$$\forall t \in [T], \ \sigma_t^2 \leq \sigma_{\max}^2 \ and \ \sup_{\mathbf{x} \in \mathcal{X}} \|\nabla F_t(\mathbf{x}) - \nabla F_{t-1}(\mathbf{x})\|_2^2 \leq \Sigma_{\max}^2.$$

**Assumption 8.** *At each round $t$, the time-varying function $f_t(\cdot)$ is $\alpha$-exp-concave over $\mathcal{X}$, i.e., $\forall \mathbf{x} \in \mathcal{X}$, $\exp(-\alpha f_t(\mathbf{x}))$ is concave.*

### 3.2 Optimistic composite mirror descent

OptCMD is an algorithmic realization of the powerful optimistic online learning [Rakhlin and Sridharan, 2013] in online composite optimization. Specifically, in each round $t$, the learner submits a decision $\mathbf{x}_t$ and suffers a loss $f_t(\mathbf{x}_t) + r(\mathbf{x}_t)$. Then, the learner receives an optimism $M_{t+1}$ that serves as an optimistic prediction for the gradient of subsequent function $f_{t+1}(\cdot)$. After that, the learner performs the following update steps:

$$\hat{\mathbf{x}}_{t+1} = \underset{\mathbf{x} \in \mathcal{X}}{\arg\min} \left\{ \langle \nabla f_t(\mathbf{x}_t), \mathbf{x} \rangle + r(\mathbf{x}) + \mathcal{B}^{\mathcal{R}_t}(\mathbf{x}, \hat{\mathbf{x}}_t) \right\} \tag{4}$$

$$\mathbf{x}_{t+1} = \underset{\mathbf{x} \in \mathcal{X}}{\arg\min} \left\{ \langle M_{t+1}, \mathbf{x} \rangle + r(\mathbf{x}) + \mathcal{B}^{\mathcal{R}_{t+1}}(\mathbf{x}, \hat{\mathbf{x}}_{t+1}) \right\} \tag{5}$$

where $\mathcal{B}^{\mathcal{R}_t}(\mathbf{x}, \mathbf{y}) = \mathcal{R}_t(\mathbf{x}) - \mathcal{R}_t(\mathbf{y}) - \langle \nabla \mathcal{R}_t(\mathbf{y}), \mathbf{x} - \mathbf{y} \rangle$ denotes the Bregman divergence associated with a time-varying function $\mathcal{R}_t$, of which the specific form depends on the type of $f_t(\cdot)$ and will be illuminated later. Note that in (4) and (5), $r(\cdot)$ remains non-linearilzed, which allows the decisions $\{\mathbf{x}_t\}_{t=1}^T$ to possess properties externally conferred by $r(\cdot)$, such as the sparsity.

**Remark.** Scroccaro et al. [2023] specialize the estimate $M_{t+1} = \nabla \hat{f}_{t+1}(\hat{\mathbf{x}}_{t+1})$, where $\hat{f}_{t+1}(\cdot)$ denotes the function prediction of $f_{t+1}(\cdot)$ and is generally set as $f_t(\cdot)$ in practice. We hightlight that this specification is unsuitable for the composite SEA model and inadvertently introduces a dependency issue during the analysis. More detailed discussions can be found in Sections 4.1 and 4.2.

## 4 OptCMD for composite SEA

In this section, we provide our results for general convex, strongly convex and exp-concave cases in the composite SEA model. Due to space limitations, all the proofs are deferred to Appendix B.

### 4.1 General convex case

Initially, we consider the case where the regualizer $r(\cdot)$ is convex and non-smooth, and the expected functions $F_t(\cdot)$ are general convex and smooth. We choose the following configuration for this case.

$$\mathcal{R}_t(\mathbf{x}) = \frac{1}{2\eta_t}\|\mathbf{x}\|_2^2, \ \eta_t = \min\left\{ \frac{D}{\sqrt{1+\bar{V}_{t-1}}}, \frac{D}{\delta} \right\}, \ and \ M_{t+1} = \nabla f_t(\mathbf{x}_t), \tag{6}$$

where $\bar{V}_{t-1} = \sum_{s=1}^{t-1} \|\nabla f_s(\mathbf{x}_s) - \nabla f_{s-1}(\mathbf{x}_{s-1})\|_2^2$ and $\delta > 0$ denotes the hyperparameter. With the configuration in (6), we establish the following regret bound for the general convex case.

**Theorem 1.** *Under Assumptions 1, 2, 3, 4 and 5, Algorithm 1 ensures*

$$\mathbb{E}[\text{Regret}_T] = \mathcal{O}\left( \sqrt{\sigma_{1:T}^2} + \sqrt{\Sigma_{1:T}^2} \right)$$

*with the configuration in (6) where $\delta = 6\sqrt{2}HD$.*

**Remark.** For the fully adversarial environments where $\Sigma_{1:T}^2 = V_T$ and $\sigma_{1:T}^2 = 0$, our bound degenerates to $\mathcal{O}(\sqrt{V_T})$ matching the previous result of Scroccaro et al. [2023]. For the fully stochastic environments where $\Sigma_{1:T}^2 = 0$ and $\sigma_{1:T}^2 = \sigma^2 T$ with the stochastic variance $\sigma^2$, our result becomes $\mathcal{O}(\sqrt{T})$ regret bound and further delivers an $\mathcal{O}(1/\sqrt{T})$ excess risk bound by the online-to-batch conversion [Cesa-Bianchi et al., 2004], which coincides with the results of Lan [2012]. Furthermore, our finding also aligns with existing bounds for the SEA model without $r(\cdot)$ [Sachs et al., 2022, Chen et al., 2023]. In other words, our method pays *no price* in regret bounds for handling the additional regularizer.

Beyond the configuration in (6), we also employ those adopted by Scroccaro et al. [2023]:

$$\mathcal{R}_t(\mathbf{x}) = \frac{1}{2\eta_t}\|\mathbf{x}\|_2^2, \ \eta_t = \frac{1}{\sqrt{4H^2 + \bar{D}_{t-1}}}, \text{ and } M_{t+1} = \nabla f_t(\hat{\mathbf{x}}_{t+1}), \tag{7}$$

where $\bar{D}_0 = 0$ and $\bar{D}_{t-1} = \sum_{s=1}^{t-1} \|\nabla f_s(\hat{\mathbf{x}}_s) - \nabla f_{s-1}(\hat{\mathbf{x}}_s)\|_2^2$. By setting the configuration in (7), we obtain the following bound.

**Theorem 2.** *Under Assumptions 1, 2, 3, 4 and 5, with configuration in* (7)*, Algorithm 1 ensures*

$$\mathbb{E}\left[\mathrm{Regret}_T\right] = \mathcal{O}\left(\sqrt{\tilde{\sigma}_{1:T}^2} + \sqrt{\Sigma_{1:T}^2}\right),$$

*where $\tilde{\sigma}_{1:T}^2$ with $\tilde{\sigma}_t^2 = \mathbb{E}_{f_t \sim \mathcal{D}_t}[\sup_{\mathbf{x} \in \mathcal{X}} \|\nabla f_t(\mathbf{x}) - \nabla F_t(\mathbf{x})\|_2^2]$.*

**Remark.** This bound is less favorable than that in Theorem 1, as it scales with a new quantity $\tilde{\sigma}_{1:T}^2$, which also measures the stochasticity in composite SEA but is larger than $\sigma_{1:T}^2$ because of the fact that $\mathbb{E}_{f_t \sim \mathcal{D}_t}[\sup_{\mathbf{x} \in \mathcal{X}} \|\nabla f_t(\mathbf{x}) - \nabla F_t(\mathbf{x})\|_2^2] \geq \sup_{\mathbf{x} \in \mathcal{X}} \mathbb{E}_{f_t \sim \mathcal{D}_t}[\|\nabla f_t(\mathbf{x}) - \nabla F_t(\mathbf{x})\|_2^2]$, where the inequality is due to the convexity of supremum operator. This discrepancy arises from the inappropriate choice of $M_{t+1} = \nabla f_t(\hat{\mathbf{x}}_{t+1})$, where $\hat{\mathbf{x}}_{t+1}$ is generated based on $\nabla f_t(\mathbf{x}_t)$ shown in (4), which implies that $\hat{\mathbf{x}}_{t+1}$ has already incorporated partial information about $f_t(\cdot)$, leading to a dependence issue in analysis. To remove the dependency, we apply the supremum operator on $\hat{\mathbf{x}}_t$ over $\mathcal{X}$ (c.f. Lemma 4) but inevitably establish a reliance on $\tilde{\sigma}_{1:T}^2$ in the final bound.

### 4.2 Strongly convex case

In this part, we focus on the case where the regualizer $r(\cdot)$ is non-smooth and convex, and the expected functions $F_t(\cdot)$ are smooth and $\lambda$-strongly convex. We set OptCMD with the configurations:

$$\mathcal{R}_t(\mathbf{x}) = \frac{1}{2\eta_t}\|\mathbf{x}\|_2^2, \ \eta_t = \frac{2}{\delta + \lambda t}, \text{ and } M_{t+1} = \nabla f_t(\mathbf{x}_t), \tag{8}$$

where $\delta > 0$ denotes the hyperparameter. The theoretical guarantee for this case is shown below.

**Theorem 3.** *Under Assumptions 1, 2, 3, 4, 6 and 7, Algorithm 1 ensures*

$$\mathbb{E}\left[\mathrm{Regret}_T\right] = \mathcal{O}\left(\frac{1}{\lambda}\left(\sigma_{\max}^2 + \Sigma_{\max}^2\right) \log\left(\sigma_{1:T}^2 + \Sigma_{1:T}^2\right)\right).$$

*with the configuration in* (8) *where $\delta = 128H^2D^2$.*

**Remark.** Similar to Theorem 1, this bound reduces to $\mathcal{O}(\lambda^{-1} \log V_T)$ matching that of Scroccaro et al. [2023] when environments become fully adversarial, and derives the same $\mathcal{O}(\log T/(\lambda T))$ excess risk bound as that of Duchi and Singer [2009] and Xiao [2009] when environments become fully stochastic. Moreover, this bound also recovers that of Chen et al. [2023] for the SEA model.

Additionally, we further equip OptCMD with the original configuration of Scroccaro et al. [2023]:

$$\mathcal{R}_t(\mathbf{x}) = \frac{1}{2\eta_t}\|\mathbf{x}\|_2^2, \ \eta_t = \frac{1}{4H^2 + (\lambda/2G^2)\bar{D}_{t-1}}, \text{ and } M_{t+1} = \nabla f_t(\hat{\mathbf{x}}_{t+1}), \tag{9}$$

and obtain the following theorem.

**Theorem 4.** *Under Assumptions 1, 2, 3, 4 and 6, with the configuration in* (9)*, Algorithm 1 ensures*

$$\mathbb{E}\left[\mathrm{Regret}_T\right] = \mathcal{O}\left(\frac{G^2}{\lambda} \log\left(\tilde{\sigma}_{1:T}^2 + \Sigma_{1:T}^2\right)\right).$$

**Remark.** This bound suffers two limitations. Firstly, it depends on the unfavorable $\tilde{\sigma}_{1:T}^2$, which is due to the improper estimation $M_{t+1}$ in (9). Secondly, it scales with $\mathcal{O}(G^2)$ due to the choice of $\eta_t$ in (9), which is not tighter than $\mathcal{O}(\sigma_{\max}^2 + \Sigma_{\max}^2)$ in Theorem 3.

---
**Algorithm 2** A Universal Strategy for Composite SEA
---
**Initialization**: $\mathcal{M}_{\text{top}}$ with $\eta^k = (C_0 \cdot 2^k)^{-1}$ and $\hat{q}_1^k = (\eta^k)^2 / \sum_{k=1}^{K}(\eta^k)^2$; for each $k \in [K]$, $\mathcal{M}_{\text{mid}}^k$ with $\eta^{k,i} = 2\eta^k$ and $\hat{p}_1^{k,i} = 1/N$, and experts $\{E^{k,i}\}_{i \in [N]}$

1: **for** $t = 1$ **to** $T$ **do**
2:     Receive $\mathbf{x}_t^{k,i}$ from expert $E^{k,i}$, and update $\mathbf{q}_t$ and $\mathbf{p}_t^k$ according to (11)
3:     Compute $\mathbf{x}_t^k = \sum_i p_t^{k,i} \mathbf{x}_t^{k,i}$ and submit $\mathbf{x}_t = \sum_k q_t^k \mathbf{x}_t^k$
4:     Suffer the loss $f_t(\mathbf{x}_t) + r(\mathbf{x}_t)$ and observe the gradient $\mathbf{g}_t = \nabla f_t(\mathbf{x}_t)$
5:     Update $\hat{\mathbf{q}}_{t+1}$ and $\hat{\mathbf{p}}_{t+1}^k$ according to (12)
6:     Send the gradient $\mathbf{g}_t$ and the regularizer $r(\cdot)$ to experts
7: **end for**
---

### 4.3 Exp-concave case

We further investigate the exp-concave case where the regualizer $r(\cdot)$ is non-smooth and convex, and the individual functions $f_t(\cdot)$ are smooth and $\alpha$-exp-concave.

**Remark.** In this case, we require the exponential concavity on the individual function $f_t(\cdot)$ instead of the expected one $F_t(\cdot)$. This is due to the technical demand in analysis, and similar assumptions are also adopted by Chen et al. [2023] and Yang et al. [2024c].

In this case, we set OptCMD with the following configuration:

$$\mathcal{R}_t(\mathbf{x}) = \frac{1}{2}\|\mathbf{x}\|_{H_t}^2, \ H_t = \delta I + \frac{\beta G^2}{2}I + \frac{\beta}{2}\sum_{s=1}^{t-1} h_s, \text{ and } M_{t+1} = \nabla f_t(\mathbf{x}_t), \qquad (10)$$

where $\delta > 0$ denotes the hyperparameter, $I$ denotes the $d$-dimension identity matrix, $h_t = \nabla f_t(\mathbf{x}_t)\nabla f_t(\mathbf{x}_t)^\top$ and $\beta = (1/2)\min\{1/(4GD), \alpha\}$. The regret for this case is stated below.

**Theorem 5.** *Under Assumptions 1, 2, 3, 4 and 8, Algorithm 1 ensures*

$$\mathbb{E}[\text{Regret}_T] = \mathcal{O}\left(\frac{d}{\alpha}\log\left(\sigma_{1:T}^2 + \Sigma_{1:T}^2\right)\right).$$

*with the configuration in (10) where $\delta = 1$.*

**Remark.** For the fully adversarial setting, this bound degenerates to $\mathcal{O}((d/\alpha)\log V_T)$, which is the *first* problem-dependent bound for exp-concave time-varying functions in online composite optimization, and tighter than $\mathcal{O}((d/\alpha)\log T)$ by Yang et al. [2024c] in benign environments. For the fully stochastic setting, our result reduces to $\mathcal{O}((d/\alpha)\log T)$ regret bound and further implies the same excess risk of $\mathcal{O}(d\log T/(\alpha T))$ as that of Yang et al. [2024c] through the online-to-batch conversion. Furthermore, it also matches existing results for the SEA model [Chen et al., 2023].

## 5 The universal strategy

Although we have established favorable theoretical results in the composite SEA model, achieving these guarantees requires prior knowledge of the function type to set appropriate configurations, e.g., (6) for the general convex case. This requirement is often impractical in real-world scenarios and motivates us to design a universal strategy. In the previous study of Yan et al. [2023], based on the meta-expert framework [van Erven and Koolen, 2016, Wang et al., 2020, Zhang et al., 2022], they propose a multi-layer universal algorithm, which can adapt to the SEA model without regularizer by choosing optimistic OMD [Chen et al., 2023] as the expert-algorithm. Unfortunately, their method [Yan et al., 2023] requires the smoothness of the (summation) loss functions, which does not necessarily hold for (1), so that it cannot be directly applied to the composite SEA model. In this paper, we propose a *novel* universal algorithm for composite SEA. For the expert-algorithm, we choose OptCMD due to its ability in handling three cases in composite SEA. For the meta-algorithm, we design a new method that is able to effectively manage the non-smooth component $r(\cdot)$ in (1) and can thus track experts according to their performances in the composite SEA model.

Our method is based on two key observations. Firstly, $r(\cdot)$ in (1) is fixed over time and available to the meta-algorithm from the beginning round. Secondly, the meta-algorithm can access expert

decisions for the current round before the performance estimation. Therefore, the information about $r(\cdot)$ can be introduced into the expert tracking process, securing that the regret of meta-algorithm will not deviate from that of the best expert. Specifically, inspired by Yan et al. [2023], we employ a two-layer meta-algorithm, with each layer running an MsMwC algorithm [Chen et al., 2021] that maintains two weight sequences $\{\mathbf{q}_t, \hat{\mathbf{q}}_t \in \Delta^N\}_{t \in [T]}$, and a group of losses $\boldsymbol{l}_t = (l_t^1, \cdots, l_t^N)$ and optimisms $\boldsymbol{m}_t = (m_t^1, \cdots, m_t^N)$ to measure the performance of $N$ experts. To handle the composite loss, we explicitly incorporate $r(\cdot)$ into both $\boldsymbol{l}_t$ and $\boldsymbol{m}_t$. The incorporation serves two purposes: (i) it endows $\boldsymbol{l}_t$ with a composite structure, facilitating the estimation of expert performance on composite losses; (ii) it utilizes the cancellation between $\boldsymbol{l}_t$ and $\boldsymbol{m}_t$ to eliminate the non-smooth component in loss functions, i.e., $r(\cdot)$, so that the meta-algorithm is able to leverage the smoothness of $f_t(\cdot)$. In the following, we describe our universal algorithm, which is also summarized in Algorithm 2.

**Meta-algorithm.** Overall, our algorithm comprises two components: a two-layer meta-algorithm and a set of experts, together forming a three-layer structure. Specifically, on the top layer, we run one MsMwC named $\mathcal{M}_{\text{top}}$, which maintains the weights $\mathbf{q} \in \Delta_K$ assigned to $K = \mathcal{O}(\log T)$ MsMwCs, named $\mathcal{M}_{\text{mid}}$. On the middle layer, each $\mathcal{M}_{\text{mid}}^k$ ($k \in [K]$) maintains the weights $\mathbf{p}^k \in \Delta_N$ assigned to $N = \mathcal{O}(\log T)$ experts. At the round $t$, we receive the decision $\mathbf{x}_t^{k,i}$ from each expert $E^{k,i}$, and update the weights $\mathbf{q}_t$ and $\mathbf{p}_t^k$ according to:

$$\mathbf{q}_t = \underset{\mathbf{q} \in \Delta^K}{\arg\min} \left\{ \langle \boldsymbol{m}_t, \mathbf{q} \rangle + \mathcal{B}^{\psi_1}(\mathbf{q}, \hat{\mathbf{q}}_t) \right\}, \ \mathbf{p}_t^k = \underset{\mathbf{p} \in \Delta^N}{\arg\min} \left\{ \langle \boldsymbol{m}_t^k, \mathbf{p} \rangle + \mathcal{B}^{\psi_2}(\mathbf{p}, \hat{\mathbf{p}}_t^k) \right\}, \quad (11)$$

where $\psi_1(\mathbf{q}) = \sum_k (q^k \ln q^k)/\eta^k$ and $\psi_2(\mathbf{p}) = \sum_i (p^i \ln p^i)/\eta^{k,i}$ denote the negative entropy functions, and $\boldsymbol{m}_t^k$ and $\boldsymbol{m}_t$ denote the optimism used by $\mathcal{M}_{\text{mid}}^k$ and $\mathcal{M}_{\text{top}}$, respectively. Then, we compute the weighted average decision $\mathbf{x}_t^k = \sum_i p_t^{k,i} \mathbf{x}_t^{k,i}$ and submit $\mathbf{x}_t = \sum_k q_t^k \mathbf{x}_t^k$. Next, we suffer the loss $f_t(\mathbf{x}_t) + r(\mathbf{x}_t)$ and observe the gradient $\mathbf{g}_t = \nabla f_t(\mathbf{x}_t)$. We update $\hat{\mathbf{q}}_{t+1}$ and $\hat{\mathbf{p}}_{t+1}^k$ by

$$\hat{\mathbf{q}}_{t+1} = \underset{\mathbf{q} \in \Delta^K}{\arg\min} \left\{ \langle \boldsymbol{l}_t + \boldsymbol{a}_t, \mathbf{q} \rangle + \mathcal{B}^{\psi_1}(\mathbf{q}, \hat{\mathbf{q}}_t) \right\}, \ \hat{\mathbf{p}}_{t+1}^k = \underset{\mathbf{p} \in \Delta^N}{\arg\min} \{ \langle \boldsymbol{l}_t^k + \mathbf{b}_t^k, \mathbf{p} \rangle + \mathcal{B}^{\psi_2}(\mathbf{p}, \hat{\mathbf{p}}_t^k) \}, \quad (12)$$

where $\boldsymbol{l}_t = (l_t^1, \cdots, l_t^k)$ and $\boldsymbol{l}_t^k = (l_t^{k,1}, \cdots, l_t^{k,N})$ denote losses used by $\mathcal{M}_{\text{top}}$ and $\mathcal{M}_{\text{mid}}^k$, respectively, and $\boldsymbol{a}_t \in \mathbb{R}^K$ and $\mathbf{b}_t^k \in \mathbb{R}^N$ denote the correction terms. At last, we send the gradient $\mathbf{g}_t$ and the regularizer $r(\cdot)$ to each expert. Now, we specify the loss and optimism in (11) and (12). For $\mathcal{M}_{\text{top}}$, we choose

$$l_t^k = \langle \mathbf{g}_t, \mathbf{x}_t^k \rangle + r(\mathbf{x}_t^k) + \gamma_1 \| \mathbf{x}_t^k - \mathbf{x}_{t-1}^k \|_2^2, \ m_t^k = \langle \hat{\mathbf{m}}_t^k, \mathbf{p}_t^k \rangle + r(\mathbf{x}_t^k) + \gamma_1 \| \mathbf{x}_t^k - \mathbf{x}_{t-1}^k \|_2^2, \quad (13)$$

where $\hat{\boldsymbol{m}}_t^k = (\hat{m}_t^{k,1}, \cdots, \hat{m}_t^{k,N})$ with $\hat{m}_t^{k,i} = \langle \mathbf{g}_t, \mathbf{x}_t \rangle - \langle \mathbf{g}_{t-1}, \mathbf{x}_{t-1} - \mathbf{x}_{t-1}^{k,i} \rangle$. For $\mathcal{M}_{\text{mid}}^k$, we choose

$$l_t^{k,i} = \langle \mathbf{g}_t, \mathbf{x}_t^{k,i} \rangle + r(\mathbf{x}_t^{k,i}) + \gamma_2 \| \mathbf{x}_t^{k,i} - \mathbf{x}_{t-1}^{k,i} \|_2^2, \ m_t^{k,i} = \hat{m}_t^{k,i} + r(\mathbf{x}_t^{k,i}) + \gamma_2 \| \mathbf{x}_t^{k,i} - \mathbf{x}_{t-1}^{k,i} \|_2^2. \quad (14)$$

In the above, $\gamma_1$, $\gamma_2$ denote hyperparameters, and $\gamma_1 \| \mathbf{x}_t^k - \mathbf{x}_{t-1}^k \|_2^2$ and $\gamma_2 \| \mathbf{x}_t^{k,i} - \mathbf{x}_{t-1}^{k,i} \|_2^2$ are injected for cancellation during the analysis [Yan et al., 2023]. It should be noticed that we explicitly introduce $r(\cdot)$ in (13) and (14). This is because experts are running over the composite losses and their performance assessment should similarly reflect the composite structure, ensuring an accurate measure for expert tracking. Moreover, the optimism $\boldsymbol{m}_t$ with $r(\cdot)$ helps eliminate the non-smooth component in $\boldsymbol{l}_t$ and thereby enables the meta-algorithm to effectively utilize the smoothness of time-varying functions. To make it clearer, we take the general convex case as an example.

First, the expected regret can be decomposed into the meta-regret and the expert-regret, as below:

$$\mathbb{E}\left[\text{Regret}_T\right] = \mathbb{E}\left[\sum_{t=1}^T \left[f_t(\mathbf{x}_t) + r(\mathbf{x}_t)\right] - \min_{\mathbf{x} \in \mathcal{X}} \sum_{t=1}^T \left[f_t(\mathbf{x}) + r(\mathbf{x})\right]\right]$$

$$\leq \underbrace{\mathbb{E}\left[\sum_{t=1}^T \langle \mathbf{g}_t, \mathbf{x}_t - \mathbf{x}_t^{k^*,i^*} \rangle + r(\mathbf{x}_t) - r(\mathbf{x}_t^{k^*,i^*})\right]}_{:=\text{meta-regret}} + \underbrace{\mathbb{E}\left[\sum_{t=1}^T \langle \mathbf{g}_t, \mathbf{x}_t^{k^*,i^*} - \mathbf{x}^* \rangle + r(\mathbf{x}_t^{k^*,i^*}) - r(\mathbf{x}^*)\right]}_{:=\text{expert-regret}},$$

where $\mathbf{x}_t^{k^*,i^*}$ denotes the decision made by the best expert $E^{k^*,i^*}$. For the expert-regret, we can directly apply the result in Theorem 1. Therefore, the key is how to bound the meta-regret. By the fact that $r(\mathbf{x}_t) \leq \sum_k q_t^k r(\mathbf{x}_t^k)$ and $r(\mathbf{x}_t^k) \leq \sum_i p_t^{k,i} r(\mathbf{x}_t^{k,i})$, the meta-regret is bounded by

$$\text{meta-regret} \leq \sum_{t=1}^T \left[ \langle \boldsymbol{q}_t, \boldsymbol{l}_t \rangle - l_t^{k^*} \right] + \sum_{t=1}^T \left[ \left\langle \boldsymbol{p}_t^{k^*}, \boldsymbol{l}_t^{k^*} \right\rangle - l_t^{k^*,i^*} \right] + \mathcal{O}(1), \quad (15)$$

where the first and second terms denote the regrets for tracking experts of $\mathcal{M}_{\text{top}}$ and $\mathcal{M}_{\text{mid}}^{k^*}$, respectively. While these two terms involve the losses $l_t^k$ and $l_t^{k^*,i}$ that incorporate the regularizer, they can be simultaneously controlled by a favorable bound of $\mathcal{O}(\sum_{t\in[T]}(l_t^{k^*,i^*} - m_t^{k^*,i^*})^2)$, which not only eliminates $r(\cdot)$ by the cancellation between $l_t^{k^*,i^*}$ and $m_t^{k^*,i^*}$, but also suffices to achieve the desirable bound of $\tilde{\mathcal{O}}(\sqrt{\sigma_{1:T}^2} + \sqrt{\Sigma_{1:T}^2})$ for general convex case and $\mathcal{O}(1)$ for other two cases. More details can be found in Appendix B.6.

**Experts.** Each expert is an instance of OptCMD that equips with certain configuration, i.e., (6), (8) or (10). To estimate the unknown curvature $\lambda$ and $\alpha$, we utilize the discretization strategy by Zhang et al. [2022], constructing the candidate sets as

$$\mathcal{P}_{str} = \mathcal{P}_{exp} = \{1/T, 2/T, 2^2/T, \cdots, 2^N/T\},$$

where $N = \lceil \log_2 T \rceil$. Based on the two sets, we create following three types of experts:

- Strongly convex experts, each of which is configured with (8) and a candidate $\lambda^i \in \mathcal{P}_{str}$;
- Exp-concave experts, each of which is configured with (10) and a candidate $\alpha^i \in \mathcal{P}_{exp}$;
- General convex expert, which is configured with (6).

As demonstrated by Zhang et al. [2022], these experts are sufficient to identify the best one that runs for the true function type with the most accurate curvature. For instance, in the strongly convex case, there must exist an expert $E^i$ equipped with $\lambda^i \in \mathcal{P}_{str}$ that satisfies $\lambda^i \leq \lambda \leq 2\lambda^i$. Moreover, instead of directly minimizing the original loss function (1), each expert runs over a *new* composite surrogate loss to avoid high-computational gradient query costs. Specifically, we choose $h_t^c(\mathbf{x}) = \langle \mathbf{g}_t, \mathbf{x}\rangle + r(\mathbf{x})$, $h_{t,i}^{sc}(\mathbf{x}) = \langle \mathbf{g}_t, \mathbf{x}\rangle + r(\mathbf{x}) + \lambda^i \|\mathbf{x} - \mathbf{x}_t\|_2^2/2$ and $h_{t,i}^{exp}(\mathbf{x}) = \langle \mathbf{g}_t, \mathbf{x}\rangle + r(\mathbf{x}) + \beta^i \langle \mathbf{g}_t, \mathbf{x} - \mathbf{x}_t\rangle^2/2$ for general convex, strongly convex, and exp-concave experts, respectively. These surrogate losses not only inherit the composite structure and properties of $f_t(\mathbf{x})$, e.g., the strongly convexity, but they also require only *one* gradient query for $\mathbf{g}_t = \nabla f_t(\mathbf{x}_t)$ in each round. The theoretical guarantees of our universal algorithm are presented below.

**Theorem 6.** *Under Assumptions 1, 2, 3, 4 and 7, with proper hyperparameters, Algorithm 2 ensures the bounds of $\tilde{\mathcal{O}}(\sqrt{\sigma_{1:T}^2} + \sqrt{\Sigma_{1:T}^2})$, $\mathcal{O}((\sigma_{\max}^2 + \Sigma_{\max}^2)\log(\sigma_{1:T}^2 + \Sigma_{1:T}^2))$, and $\mathcal{O}(d\log(\sigma_{1:T}^2 + \Sigma_{1:T}^2))$ for general convex, strongly convex, and exp-concave time-varying functions, respectively.*

**Remark.** Theorem 6 indicates that Algorithm 2 can attain comparable regret bounds to those of OptCMD for the three cases, without knowing the function type and curvature in advance.

# 6 Implications

In the following, we discuss implications of our results for two common intermediate examples.

**Composite adversarially corrupted stochastic data.** We first focus on the composite adversarially corrupted stochastic model, of which a special case without regularizer has been widely investigated in learning with expert advice [Amir et al., 2020], bandits [Zimmert and Seldin, 2021, Ito, 2021], and online optimization [Sachs et al., 2022, Chen et al., 2023]. Specifically, each loss function consists of three terms: $\phi_t(\cdot) = h_t(\cdot) + c_t(\cdot) + r(\cdot)$, where $h_t(\cdot)$ denotes the loss of i.i.d. data from a fixed distribution $\mathcal{D}$ with the variance $\sigma$, and $c_t(\cdot)$ denotes an adversarial perturbation measured by a parameter $C_T > 0$, satisfying $\sum_{t\in[T]} \max_{\mathbf{x}\in\mathcal{X}} \|\nabla c_t(\mathbf{x})\|_2 \leq C_T$, and $r(\cdot)$ denotes the regularizer. In this model, the stochasticity and adversariality come from $h_t(\cdot)$ and $c_t(\cdot)$, respectively. Therefore, $\sigma_{1:T}^2$ and $\Sigma_{1:T}^2$ can be formulated as

$$\sigma_{1:T}^2 = \sigma^2 T, \ \Sigma_{1:T}^2 = \mathbb{E}\left[\sum_{t=1}^T \sup_{\mathbf{x}\in\mathcal{X}} \|\nabla c_t(\mathbf{x}) - \nabla c_{t-1}(\mathbf{x})\|_2^2\right] \leq 4GC_T. \tag{16}$$

With the above specification, our theoretical results can naturally be extended to this scenarios, delivering the following corollary.

**Corollary 1.** *With the specification* (16) *in the scenarios of composite adversarially corrupted stochastic data, we can obtain an $\mathcal{O}(\sigma\sqrt{T} + \sqrt{C_T})$ bound for the general convex case by Theorem 1, an $\mathcal{O}(\log(\sigma^2 T + C_T))$ bound for the strongly convex case by Theorem 3, and an $\mathcal{O}(d\log(\sigma^2 T + C_T))$ bound for the exp-concave case by Theorem 5.*

**Adversarial online learning with limited resources.** We then consider the another common scenarios where loss functions arrive in batches and the available computing resources are insufficient to process them all [Bottou and Cun, 2003, Benczúr et al., 2018, Li et al., 2024, Zhou, 2024]. Specifically, in each round $t$, we receive a group of functions $\mathcal{F}_t = \{f_t(\cdot, i)\}_{i \in [K_t]}$ with the size $K_t$, each of which is selected by the environments adversarially. We denote by $F_t(\cdot) = K_t^{-1} \sum_{i \in [K_t]} f_t(\cdot, i)$ the average function. Due to limited computing resources, we can only sample a subset of functions for gradient estimation, and generate sparse decisions for efficient storage and inference. For this reason, our goal is to minimize $\phi_t(\cdot) = h_t(\cdot) + r(\cdot)$, where $h_t(\cdot) = B_t^{-1} \sum_{i \in [B_t]} \hat{f}_t(\cdot, i)$ denotes the approximation of $F_t(\cdot)$ by i.i.d. sampling $B_t \in [K_t]$ functions $\hat{f}_t(\cdot, i)$ from the group $\mathcal{F}_t$, and $r(\cdot) = \| \cdot \|_1$ denotes the $\ell_1$-norm regularizer. In this scenarios, it can be verified that

$$\sigma_{1:T}^2 = \sum\nolimits_{t=1}^{T} \sup_{\mathbf{x} \in \mathcal{X}} \mathbb{E}_{\hat{f}_t \sim \mathcal{F}_t} \left[ \left\| B_t^{-1} \sum\nolimits_{i=1}^{B_t} \nabla \hat{f}_t(\cdot, i) - \nabla F_t(\mathbf{x}) \right\|_2^2 \right] \leq 4G^2 \sum\nolimits_{t=1}^{T} B_t^{-1} \quad (17)$$

and $\Sigma_{1:T}^2 = \mathbb{E}[\sum_{t=1}^{T} \sup_{\mathbf{x} \in \mathcal{X}} \|\nabla F_t(\mathbf{x}) - \nabla F_{t-1}(\mathbf{x})\|_2^2]$. Applying the above specification into our theorems delivers the following corollary.

**Corollary 2.** *With the specification* (17) *in the scenarios of composite adversarially corrupted stochastic data, we can obtain an* $\mathcal{O}(\sqrt{\sum_{t \in [T]} B_t^{-1}} + \sqrt{\Sigma_{1:T}^2})$ *bound for the general convex case by Theorem 1, an* $\mathcal{O}(\log(\sum_{t \in [T]} B_t^{-1} + \Sigma_{1:T}^2))$ *bound for the strongly convex case by Theorem 3, and an* $\mathcal{O}(d \log(\sum_{t \in [T]} B_t^{-1} + \Sigma_{1:T}^2))$ *bound for the exp-concave case by Theorem 5.*

# 7 Conclusion and future work

In this paper, we investigate the intermediate setting between stochastic and adversarial composite optimization, named composite SEA, and demonstrate that OptCMD is able to attain the regret bounds of $\mathcal{O}(\sqrt{\sigma_{1:T}^2} + \sqrt{\Sigma_{1:T}^2})$, $\mathcal{O}((\sigma_{\max}^2 + \Sigma_{\max}^2) \log(\sigma_{1:T}^2 + \Sigma_{1:T}^2))$ and $\mathcal{O}(d \log(\sigma_{1:T}^2 + \Sigma_{1:T}^2))$ for the general convex, strongly convex and exp-concave cases, respectively. To deal with the unknown function type in real-world problems, we further propose a *novel* universal algorithm in online composite optimization, and show that our universal algorithm is able to achieve the desirable bounds in the three cases, simultaneously. Due to the versatility of $\sigma_{1:T}^2$ and $\Sigma_{1:T}^2$, all our theoretical findings can *recover* previous results in fully stochastic and adversarial composite optimization. Finally, we explore several practical intermediate scenarios to demonstrate the implications of our results. Additionally, we also conduct empirical studies in Appendix A to verify our theoretical results.

There are many valuable directions for future research. First, our methods still rely on the domain and gradient bounded assumptions. We notice that there have been several methods designed for unbounded cases [Orabona, 2014, Orabona and Pál, 2016, Cutkosky and Orabona, 2018, Jacobsen and Cutkosky, 2022], but all of them focus on the setting without the reguralizer. Consequently, developing online algorithms with the unbounded domain and gradient for composite SEA remains an interesting research direction. Second, in the composite SEA model, we implicitly assume that the feedback (i.e., the loss function value and the gradient) is immediately revealed after making the decision, which, however, is not necessarily satisfied in practice [Joulani et al., 2013, Quanrud and Khashabi, 2015, Wan et al., 2022a,c, 2024b]. Therefore, it is also interesting to explore the composite SEA model with the delayed feedback. Thirdly, our proposed universal algorithm currently exhibits a three-layer structure, which presents several challenges in both analysis and practical implementation. Therefore, designing a simpler two-layer universal algorithm in composite SEA is another potential research direction in the future, which may require the advanced techniques in Yan et al. [2024].

## Acknowledgments and Disclosure of Funding

The authors would thank Yu-Hu Yan for helpful discussions, and the anonymous reviewers for their constructive suggestions. This work was partially supported by National Science and Technology Major Project (2022ZD0114801), and NSFC (U23A20382, 62306275).

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

# A Experiments

In this section, we conduct empirical studies to validate our theoretical results.

**Setup.** In this paper, we show that OptCMD with suitable configurations is able to achieve a series of favorable theoretical guarantees for the composite SEA model. Moreover, for the practical scenarios where the prior knowledge of loss functions is unavailable, we propose a novel universal strategy, called USC-SEA, which can still achieve the desired regret bounds for three cases in the composite SEA simultaneously. To verify our theoretical findings, we conduct experiments on the mushroom datasets from the LIBSVM repository [Chang and Lin, 2011], and consider the following online classification problem. Let $T$ denote the number of the total rounds. At each round $t \in [T]$, the learner receives a sampled data $(\mathbf{x}_t, y_t) \in \mathbb{R}^d \times \{-1, 1\}$ with $d = 112$. Then, the learner plays the decision $\mathbf{w}_t$ from the ball $\mathcal{X}$ with the diameter $D = 20$, and suffers a composite loss

$$\phi_t(\mathbf{w}_t; \mathbf{x}_t, y_t) = f_t(\mathbf{w}_t; \mathbf{x}_t, y_t) + \lambda r(\mathbf{w}_t),$$

where we set the hyper-parameter $\lambda = 0.001$. The dataset used in the experiments is considered to be sampled from an unknown distribution, possessing the inherent stochastic property. To simulate the stochastically extended adversarial environments, we perturb the dataset by randomly flipping the labels of $10\%$ data as the adversarial corruptions.

We consider the following three types of loss functions.

- For the general convex case, we choose the smooth and convex cross-entropy function as the time-varying function:

$$f_t(\mathbf{w}_t; \mathbf{x}_t, y_t) = -y_t \log \sigma \left(\mathbf{x}_t^\top \mathbf{w}_t\right) - (1 - y_t) \log \left(1 - \sigma \left(\mathbf{x}_t^\top \mathbf{w}_t\right)\right),$$

  where $\sigma(\cdot)$ denotes the sigmoid function, and utilize the $\ell_1$-norm regularizer $r(\mathbf{w}_t) = \|\mathbf{w}_t\|_1$. Therefore, the composite function takes the form of:

$$\phi(\mathbf{w}_t; \mathbf{x}_t, y_t) = -y_t \log \sigma \left(\mathbf{x}_t^\top \mathbf{w}_t\right) - (1 - y_t) \log \left(1 - \sigma \left(\mathbf{x}_t^\top \mathbf{w}_t\right)\right) + \lambda \|\mathbf{w}_t\|_1.$$

- For the strongly convex case, we employ the cross-entropy functions with the $\ell_2$-norm regularizer as the time-varying function:

$$f_t(\mathbf{w}_t; \mathbf{x}_t, y_t) = -y_t \log \sigma \left(\mathbf{x}_t^\top \mathbf{w}_t\right) - (1 - y_t) \log \left(1 - \sigma \left(\mathbf{x}_t^\top \mathbf{w}_t\right)\right) + \delta \|\mathbf{w}_t\|_2^2,$$

  which is $2\delta$-strongly convex, and still leverage the $\ell_1$-norm regularizer. Hence, the composite loss function is in the form of:

$$\phi(\mathbf{w}_t; \mathbf{x}_t, y_t) = -y_i \log \sigma \left(\mathbf{x}_i^\top \mathbf{w}_t\right) - (1 - y_i) \log \left(1 - \sigma \left(\mathbf{x}_i^\top \mathbf{w}_t\right)\right) + \delta \|\mathbf{w}_t\|_2^2 + \lambda \|\mathbf{w}_t\|_1$$

  where we set the hyper-parameter $\sigma = 0.001$.

- For the exp-concave case, we utilize the logistic function as the time-varying function:

$$f_t(\mathbf{w}_t; \mathbf{x}_t, y_t) = \log \left(1 + \exp \left(-y_t \mathbf{w}_t^\top \mathbf{x}_t\right)\right)$$

  which is exp-concave and smooth [Hazan et al., 2014], and still employ the $\ell_1$-norm regularizer. The composite loss function is shown below:

$$\phi(\mathbf{w}_t; \mathbf{x}_t, y_t) = \log \left(1 + \exp \left(-y_t \mathbf{w}_t^\top \mathbf{x}_t\right)\right) + \lambda \|\mathbf{w}_t\|_1.$$

**Contenders.** For the general convex and strongly convex cases, we compare our methods with OGD [Zinkevich, 2003], COMID [Duchi et al., 2010] and Optimistic-OMD [Chen et al., 2023]. For the exp-concave case, we choose ONS [Hazan et al., 2007], ProxONS [Yang et al., 2024c] and Optimistic-OMD [Chen et al., 2023] as the contenders. All parameters of each method are set according to their theoretical suggestions. For instance, in the general convex case, the learning rate is set as $\eta = ct^{-1/2}$ in OGD, and $\eta = cT^{-1/2}$ in COMID, $\eta_t = D(c + \bar{V}_{t-1})^{-1/2}$ in Optimistic-OMD where $c$ denotes the hyper-parameter selected from $\{10^{-3}, 10^{-2}, \cdots, 10^4\}$.

**Results.** All experiments are repeated ten times, and we report the instantaneous loss, the cumulative loss and the average loss against the number of rounds in Figure 1 for the general convex case, Figure 2 for the strongly convex case and Figure 3 for the exp-concave case. From the experimental results, it is evident that adversarial corruptions cause considerable fluctuations in the instantaneous losses across different methods. Moreover, we also observe that, in the three cases of composite SEA, both OptCMD and USC-SEA suffer lower losses compared to baseline methods, demonstrating better performance. This phenomenon can be attributed to their ability to adapt to the composite SEA environment, and their explicit support for handling the non-smooth component $r(\cdot)$.

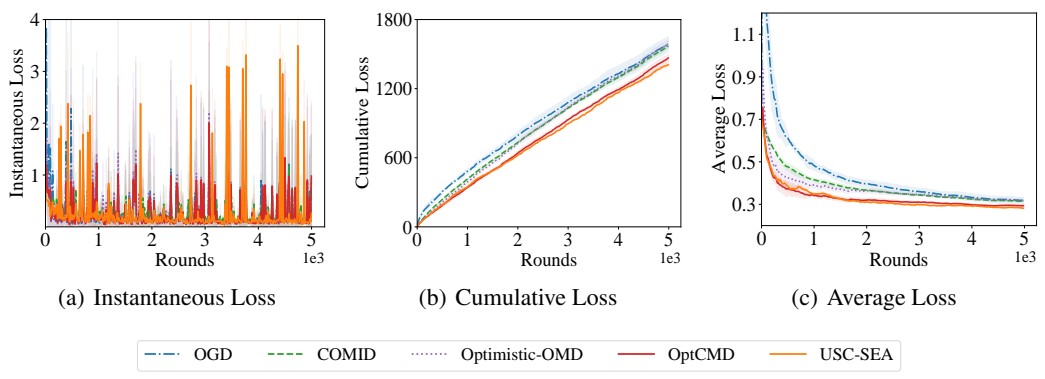

Figure 1: Experimental results for the general convex case.

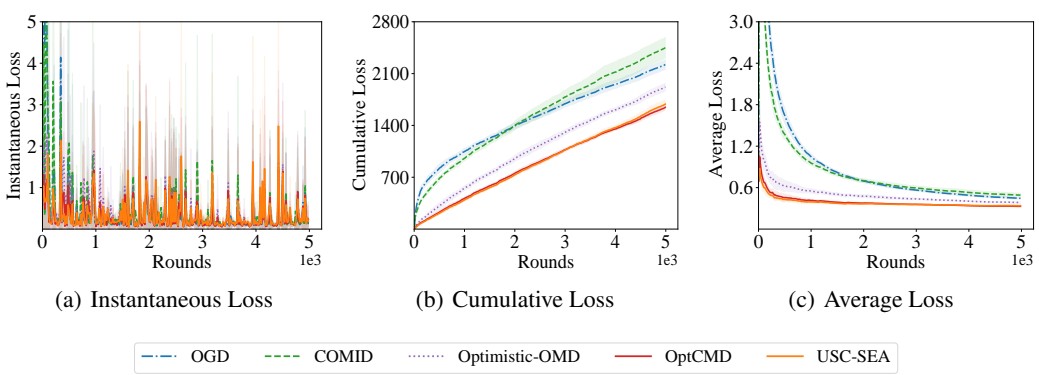

Figure 2: Experimental results for the strongly convex case.

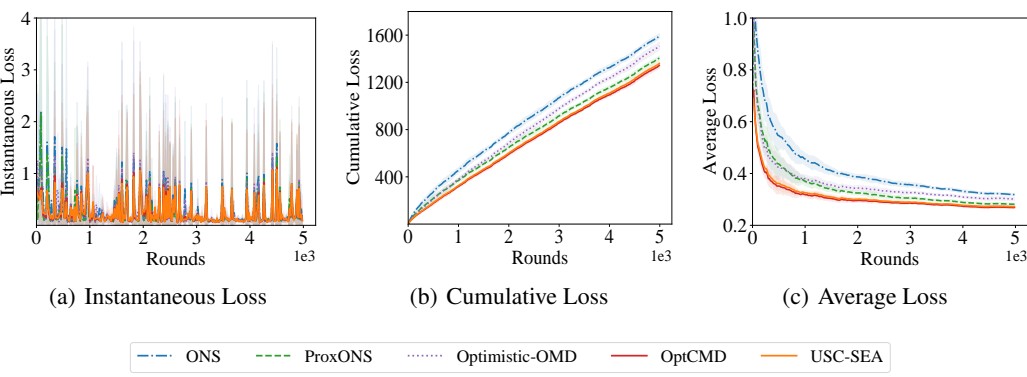

Figure 3: Experimental results for the exp-concave case.

# B   Theoretical analysis

## B.1   Proof of Theorem 1

First, since the expected function $F_t(\cdot)$ is convex in each round $t$, we could decompose the instantaneous regret as below:

$$
\begin{aligned}
\mathbb{E}\left[[f_t(\mathbf{x}_t) + r(\mathbf{x}_t)] - [f_t(\mathbf{x}^*) + r(\mathbf{x}^*)]\right] &= \mathbb{E}\left[[F_t(\mathbf{x}_t) + r(\mathbf{x}_t)] - [F_t(\mathbf{x}^*) + r(\mathbf{x}^*)]\right] \\
&\leq \mathbb{E}\left[\langle\nabla F_t(\mathbf{x}_t), \mathbf{x}_t - \mathbf{x}^*\rangle + r(\mathbf{x}_t) - r(\mathbf{x}^*)\right] = \mathbb{E}\left[\langle\nabla f_t(\mathbf{x}_t), \mathbf{x}_t - \mathbf{x}^*\rangle + r(\mathbf{x}_t) - r(\mathbf{x}^*)\right].
\end{aligned}
\tag{18}
$$

The inequality is due the convexity of $F_t(\cdot)$ and the last equality is due to the interchangeability of differentiation and integration by Leibniz integral rule. In the following, we shed the light on the right side of (18).

Then, we introduce the following lemma for OptCMD.

**Lemma 1.** *Assume $\mathcal{R}_t(\cdot)$ is an $\gamma$-strongly convex function with respect to the norm $\|\cdot\|$ and denote by $\|\cdot\|_*$ the dual norm. According to the updating rule in (4) and (5), for all $\mathbf{x} \in \mathcal{X}$, we have*

$$\langle \nabla f_t(\mathbf{x}_t), \mathbf{x}_t - \mathbf{x} \rangle + r(\mathbf{x}_t) - r(\mathbf{x}) \leq \gamma^{-1}\|M_t - \nabla f_t(\mathbf{x}_t)\|_*^2 + [\mathcal{B}^{\mathcal{R}_t}(\mathbf{x}, \hat{\mathbf{x}}_t) - \mathcal{B}^{\mathcal{R}_t}(\mathbf{x}, \hat{\mathbf{x}}_{t+1})]$$
$$- [\mathcal{B}^{\mathcal{R}_t}(\hat{\mathbf{x}}_{t+1}, \mathbf{x}_t) + \mathcal{B}^{\mathcal{R}_t}(\mathbf{x}_t, \hat{\mathbf{x}}_t)].$$
(19)

In the general convex setting, we choose $\eta_t = \min\{D/\sqrt{1 + \bar{V}_{t-1}}, D/\delta\}$, and $\|\cdot\| = \frac{1}{\sqrt{\eta_t}}\|\cdot\|_2$, and $\|\cdot\|_* = \sqrt{\eta_t}\|\cdot\|_2$. Therefore, the corresponding Bregman divergence becomes $\mathcal{B}^{\mathcal{R}_t}(\mathbf{x}, \mathbf{y}) = \frac{1}{2\eta_t}\|\mathbf{x} - \mathbf{y}\|_2^2$ with respect to the $\eta^{-1}$-strongly convex function $\mathcal{R}_t(\mathbf{x}) = \frac{1}{2\eta_t}\|\mathbf{x}\|_2^2$. Now, we make use of Lemma 1 with $M_t = \nabla f_{t-1}(\mathbf{x}_{t-1})$, and obtain

$$\langle \nabla f_t(\mathbf{x}_t), \mathbf{x}_t - \mathbf{x}^* \rangle + r(\mathbf{x}_t) - r(\mathbf{x}^*) \leq \eta_t\|\nabla f_t(\mathbf{x}_t) - \nabla f_{t-1}(\mathbf{x}_{t-1})\|_2^2$$
$$+ \frac{1}{2\eta_t}\left\{\|\mathbf{x}^* - \hat{\mathbf{x}}_t\|_2^2 - \|\mathbf{x}^* - \hat{\mathbf{x}}_{t+1}\|_2^2\right\} - \frac{1}{2\eta_t}\left\{\|\hat{\mathbf{x}}_{t+1} - \mathbf{x}_t\|_2^2 + \|\mathbf{x}_t - \hat{\mathbf{x}}_t\|_2^2\right\}.$$

Summing the above inequality over $t = 1, \cdots, T$, we have

$$\sum_{t=1}^T \langle \nabla f_t(\mathbf{x}_t), \mathbf{x}_t - \mathbf{x}^* \rangle + r(\mathbf{x}_t) - r(\mathbf{x}^*) \leq \underbrace{\sum_{t=1}^T \eta_t\|\nabla f_t(\mathbf{x}_t) - \nabla f_{t-1}(\mathbf{x}_{t-1})\|_2^2}_{\texttt{term (a)}}$$
$$+ \underbrace{\sum_{t=1}^T \frac{1}{2\eta_t}\left\{\|\mathbf{x}^* - \hat{\mathbf{x}}_t\|_2^2 - \|\mathbf{x}^* - \hat{\mathbf{x}}_{t+1}\|_2^2\right\}}_{\texttt{term (b)}} - \underbrace{\sum_{t=1}^T \frac{1}{2\eta_t}\left\{\|\hat{\mathbf{x}}_{t+1} - \mathbf{x}_t\|_2^2 + \|\mathbf{x}_t - \hat{\mathbf{x}}_t\|_2^2\right\}}_{\texttt{term (c)}}.$$
(20)

Now, we have decomposed the upper bound into three terms and will analyze them separately.

To bound the $\texttt{term (a)}$, we introduce the following lemma.

**Lemma 2.** *[Pogodin and Lattimore, 2019, Lemma 4.8] Let $l_1, \cdots, l_T$ be non-negative real numbers with the $l_t \in [0, B]$. Then, we have*

$$\sum_{t=1}^T \frac{l_t}{\sqrt{1 + \sum_{s=1}^{t-1} l_s}} \leq 4\sqrt{1 + \sum_{t=1}^T l_t} + B,$$

*where for simplicity we define $0/\sqrt{0} = 0$.*

By utilizing Lemma 2 and the fact that $\eta_t \leq D/\sqrt{1 + \bar{V}_{t-1}}$, the $\texttt{term (a)}$ can be upper bounded as follows:

$$\texttt{term (a)} \leq D\sum_{t=1}^T \frac{\|\nabla f_t(\mathbf{x}_t) - \nabla f_{t-1}(\mathbf{x}_{t-1})\|_2^2}{\sqrt{1 + \bar{V}_{t-1}}} \leq 4D\sqrt{1 + \bar{V}_T} + 4DG^2. \quad (21)$$

For the $\texttt{term (b)}$, a simple calculation delivers

$$\texttt{term (b)} = \frac{1}{2\eta_1}\|\mathbf{x}^* - \hat{\mathbf{x}}_1\|_2^2 + \frac{1}{2}\sum_{t=2}^T \left(\frac{1}{\eta_t} - \frac{1}{\eta_{t-1}}\right)\|\mathbf{x}^* - \hat{\mathbf{x}}_t\|_2^2 - \frac{1}{2\eta_T}\|\mathbf{x}^* - \hat{\mathbf{x}}_{T+1}\|_2^2$$
$$\leq \frac{D^2}{2\eta_1} + \frac{D^2}{2}\sum_{t=2}^T \left(\frac{1}{\eta_t} - \frac{1}{\eta_{t-1}}\right) = \frac{D^2}{2\eta_T} = \frac{D}{2}\sqrt{1 + \bar{V}_T} + \frac{D\delta}{2}. \quad (22)$$

For the `term (c)`, we utilize the fact that $\eta_t \leq D/\delta$, and thus obtain

$$
\begin{aligned}
\texttt{term (c)} &= \frac{1}{2}\sum_{t=2}^{T+1}\frac{1}{\eta_t}\|\hat{\mathbf{x}}_t - \mathbf{x}_{t-1}\|_2^2 + \frac{1}{2}\sum_{t=1}^{T}\frac{1}{\eta_t}\|\mathbf{x}_t - \hat{\mathbf{x}}_t\|_2^2 \\
&\geq \frac{\delta}{2D}\sum_{t=2}^{T}\left\{\|\hat{\mathbf{x}}_t - \mathbf{x}_{t-1}\|_2^2 + \|\mathbf{x}_t - \hat{\mathbf{x}}_t\|_2^2\right\} \geq \frac{\delta}{4D}\sum_{t=2}^{T}\|\mathbf{x}_t - \mathbf{x}_{t-1}\|_2^2.
\end{aligned}
\tag{23}
$$

The last step is due to the domain bounded assumption, i.e., Assumption 1. Combining (21), (22) and (23), we obtain that

$$
\sum_{t=1}^{T}\langle \nabla f_t(\mathbf{x}_t), \mathbf{x}_t - \mathbf{x}^* \rangle + r(\mathbf{x}_t) - r(\mathbf{x}^*) \leq \frac{3D}{2}\sqrt{1 + \bar{V}_T} + \frac{D\delta}{2} - \frac{\delta}{4D}\sum_{t=2}^{T}\|\mathbf{x}_t - \mathbf{x}_{t-1}\|_2^2 + 4DG^2.
\tag{24}
$$

To bound the term $\bar{V}_T$, we exploiting the following lemma.

**Lemma 3.** *Under Assumption 2 and 4, we have*

$$
\begin{aligned}
\bar{V}_T \leq& G^2 + 8\sum_{t=1}^{T}\|\nabla f_t(\mathbf{x}_t) - \nabla F_t(\mathbf{x}_t)\|_2^2 \\
&+ 4\sum_{t=2}^{T}\|\nabla F_t(\mathbf{x}_{t-1}) - \nabla F_{t-1}(\mathbf{x}_{t-1})\|_2^2 + 4H^2\sum_{t=2}^{T}\|\mathbf{x}_t - \mathbf{x}_{t-1}\|_2^2.
\end{aligned}
\tag{25}
$$

Plugging (25) into (24) obtains

$$
\begin{aligned}
&\sum_{t=1}^{T}\langle \nabla f_t(\mathbf{x}_t) + \nabla r(\mathbf{x}_t), \mathbf{x}_t - \mathbf{x}^* \rangle \\
&\leq \frac{3(D + GD)}{2} + 4DG^2 + \frac{D\delta}{2} + 6HD\sqrt{\sum_{t=2}^{T}\|\mathbf{x}_t - \mathbf{x}_{t-1}\|_2^2} - \frac{\delta}{4D}\sum_{t=2}^{T}\|\mathbf{x}_t - \mathbf{x}_{t-1}\|_2^2 \\
&+ 12D\sqrt{\sum_{t=1}^{T}\|\nabla f_t(\mathbf{x}_t) - \nabla F_t(\mathbf{x}_t)\|_2^2} + 6D\sqrt{\sum_{t=2}^{T}\|\nabla F_t(\mathbf{x}_{t-1}) - \nabla F_{t-1}(\mathbf{x}_{t-1})\|_2^2}.
\end{aligned}
$$

Then, we apply the AM-GM inequality, i.e., $6HD\sqrt{\sum_{t=2}^{T}\|\mathbf{x}_t - \mathbf{x}_{t-1}\|_2^2} \leq \frac{36H^2D^3}{\delta} + \frac{\delta}{4D}\sum_{t=2}^{T}\|\mathbf{x}_t - \mathbf{x}_{t-1}\|_2^2$, and obtain

$$
\begin{aligned}
&\sum_{t=1}^{T}\langle \nabla f_t(\mathbf{x}_t) + \nabla r(\mathbf{x}_t), \mathbf{x}_t - \mathbf{x}^* \rangle \leq 2(D + GD) + 4DG^2 + \frac{D\delta}{2} + \frac{36H^2D^3}{\delta} \\
&+ 12D\sqrt{\sum_{t=1}^{T}\|\nabla f_t(\mathbf{x}_t) - \nabla F_t(\mathbf{x}_t)\|_2^2} + 6D\sqrt{\sum_{t=2}^{T}\|\nabla F_t(\mathbf{x}_{t-1}) - \nabla F_{t-1}(\mathbf{x}_{t-1})\|_2^2}.
\end{aligned}
$$

By setting $\delta = 6\sqrt{2}HD$, we have

$$
\begin{aligned}
\sum_{t=1}^{T}\langle \nabla f_t(\mathbf{x}_t) + \nabla r(\mathbf{x}_t), \mathbf{x}_t - \mathbf{x}^* \rangle \leq& C_1 + 12D\sqrt{\sum_{t=1}^{T}\|\nabla f_t(\mathbf{x}_t) - \nabla F_t(\mathbf{x}_t)\|_2^2} \\
&+ 6D\sqrt{\sum_{t=2}^{T}\|\nabla F_t(\mathbf{x}_{t-1}) - \nabla F_{t-1}(\mathbf{x}_{t-1})\|_2^2},
\end{aligned}
$$

where $C_1 = 2(D + GD) + 4DG^2 + 6\sqrt{2}HD^2$. Next, according to the definition of $\sigma_{1:T}^2$ and $\Sigma_{1:T}^2$, the expected regret can be upper bound as follows:

$$
\mathbb{E}\left[\text{Regret}_T\right] \leq \mathbb{E}\left[\sum_{t=1}^{T}\langle\nabla f_t(\mathbf{x}_t) + \nabla r(\mathbf{x}_t), \mathbf{x}_t - \mathbf{x}^*\rangle\right]
$$

$$
\leq C_1 + 12D\sqrt{\sigma_{1:T}^2} + 6D\sqrt{\Sigma_{1:T}^2} = \mathcal{O}\left(\sqrt{\sigma_{1:T}^2} + \sqrt{\Sigma_{1:T}^2}\right).
$$

## B.2 Proof of Theorem 2

First, according to (18), we have

$$
\mathbb{E}\left[[f_t(\mathbf{x}_t) + r(\mathbf{x}_t)] - [f_t(\mathbf{x}^*) + r(\mathbf{x}^*)]\right] \leq \mathbb{E}\left[\langle\nabla f_t(\mathbf{x}_t), \mathbf{x}_t - \mathbf{x}^*\rangle\right] + r(\mathbf{x}_t) - r(\mathbf{x}^*). \tag{26}
$$

In this setting, we still choose $\|\cdot\| = \frac{1}{\sqrt{\eta_t}}\|\cdot\|_2$, and $\|\cdot\|_* = \sqrt{\eta_t}\|\cdot\|_2$. Hence, the function $\mathcal{R}_t(\mathbf{x}) = \frac{1}{2\eta_t}\|\mathbf{x}\|_2^2$ is $\eta_t^{-1}$-strongly convex and the Bregman divergence becomes $\mathcal{B}^{\mathcal{R}_t}(\mathbf{x}, \mathbf{y}) = \frac{1}{2\eta_t}\|\mathbf{x} - \mathbf{y}\|_2^2$. Then, applying Lemma 1, we obtain

$$
\langle\nabla f_t(\mathbf{x}_t), \mathbf{x}_t - \mathbf{x}^*\rangle + r(\mathbf{x}_t) - r(\mathbf{x}^*) \leq \eta_t\|M_{t+1} - \nabla f_t(\mathbf{x}_t)\|_2^2 + \frac{1}{2\eta_t}\|\mathbf{x}^* - \hat{\mathbf{x}}_t\|_2^2
$$

$$
- \frac{1}{2\eta_t}\|\mathbf{x}^* - \hat{\mathbf{x}}_{t+1}\|_2^2 - \frac{1}{2\eta_t}\|\hat{\mathbf{x}}_{t+1} - \mathbf{x}_t\|_2^2 - \frac{1}{2\eta_t}\|\mathbf{x}_t - \hat{\mathbf{x}}_t\|_2^2.
$$

Following the configuration in Scroccaro et al. [2023], we set $M_t = \nabla f_{t-1}(\hat{\mathbf{x}}_t)$ and $\eta_t = (4H^2 + \bar{D}_t)^{-1/2}$ where $\bar{D}_t = \sum_{s=1}^{t}\|\nabla f_t(\hat{\mathbf{x}}_t) - \nabla f_{t-1}(\hat{\mathbf{x}}_t)\|_2^2$. According to [Scroccaro et al., 2023, Theorem 2.5], we have

$$
\sum_{t=1}^{T}\langle\nabla f_t(\mathbf{x}_t), \mathbf{x}_t - \mathbf{x}^*\rangle + r(\mathbf{x}_t) - r(\mathbf{x}^*) \leq \left(5 + \frac{3}{2}D^2\right)\sqrt{4H^2 + \bar{D}_T}.
$$

To bound the term $\bar{D}_T$, we introduce the following lemma.

**Lemma 4.** *Under Assumption 2, we have*

$$
\bar{D}_T \leq G^2 + 6\sum_{t=1}^{T}\sup_{\mathbf{x}\in X}\|\nabla f_t(\mathbf{x}) - \nabla F_t(\mathbf{x})\|_2^2 + 4\sum_{t=1}^{T}\sup_{\mathbf{x}\in X}\|\nabla F_t(\mathbf{x}) - \nabla F_{t-1}(\mathbf{x})\|_2^2.
$$

By applying Lemma 4, we obtain

$$
\sum_{t=1}^{T}\langle\nabla f_t(\mathbf{x}_t), \mathbf{x}_t - \mathbf{x}^*\rangle + r(\mathbf{x}_t) - r(\mathbf{x}^*)
$$

$$
\leq \left(5 + \frac{3}{2}D^2\right)\left(\sqrt{4H^2 + G^2} + 3\sqrt{\sum_{t=1}^{T}\sup_{\mathbf{x}\in X}\|\nabla f_t(\mathbf{x}) - \nabla F_t(\mathbf{x})\|_2^2}\right. \tag{27}
$$

$$
\left. + 2\sqrt{\sum_{t=1}^{T}\sup_{\mathbf{x}\in X}\|\nabla F_t(\mathbf{x}) - \nabla F_{t-1}(\mathbf{x})\|_2^2}\right).
$$

Substituting (27) into (26) and taking the expectation on both sides finishes the proof.

## B.3 Proof of Theorem 3

In the beginning, we state the parameter configuration under the strongly convex setting (i.e., $F_t(\cdot)$ is strongly convex and $r(\cdot)$ is general convex). The learning rate is set as $\eta_t = \frac{2}{\delta + \lambda t}$ and the norm is chosen as $\|\cdot\| = \frac{1}{\sqrt{\eta_t}}\|\cdot\|_2$, with the dual norm $\|\cdot\|_* = \sqrt{\eta_t}\|\cdot\|_2$. Therefore, the function becomes $\mathcal{R}_t(\mathbf{x}) = \frac{1}{2\eta_t}\|\mathbf{x}\|_2^2$ and the corresponding Bregman divergence becomes $\mathcal{B}^{\mathcal{R}_t}(\mathbf{x}, \mathbf{y}) = \frac{1}{2\eta_t}\|\mathbf{x} - \mathbf{y}\|_2^2$.

Then, we make use of the strong convexity of $F_t(\cdot)$, and decompose the instantaneous regret as below:

$$\mathbb{E}\left[[f_t(\mathbf{x}_t) + r(\mathbf{x}_t)] - [f_t(\mathbf{x}^*) + r(\mathbf{x}^*)]\right] = \mathbb{E}\left[[F_t(\mathbf{x}_t) + r(\mathbf{x}_t)] - [F_t(\mathbf{x}^*) + r(\mathbf{x}^*)]\right]$$

$$\leq \mathbb{E}\left[\langle \nabla F_t(\mathbf{x}_t), \mathbf{x}_t - \mathbf{x}^*\rangle + r(\mathbf{x}_t) - r(\mathbf{x}^*) - \frac{\lambda}{2}\|\mathbf{x}^* - \mathbf{x}_t\|_2^2\right] \qquad (28)$$

$$= \mathbb{E}\left[\langle \nabla f_t(\mathbf{x}_t), \mathbf{x}_t - \mathbf{x}^*\rangle + r(\mathbf{x}_t) - r(\mathbf{x}^*) - \frac{\lambda}{2}\|\mathbf{x}^* - \mathbf{x}_t\|_2^2\right].$$

Similar to the analysis in Theorem 1, we then focus on the the right side of (28). By utilizing Lemma 1, we have

$$\sum_{t=1}^{T}\langle \nabla f_t(\mathbf{x}_t), \mathbf{x}_t - \mathbf{x}^*\rangle + r(\mathbf{x}_t) - r(\mathbf{x}^*) - \frac{\lambda}{2}\sum_{t=1}^{T}\|\mathbf{x}^* - \mathbf{x}_t\|_2^2 \leq \underbrace{\sum_{t=1}^{T}\eta_t\|\nabla f_t(\mathbf{x}_t) - \nabla f_{t-1}(\mathbf{x}_{t-1})\|_2^2}_{\texttt{term (a)}}$$

$$+ \underbrace{\sum_{t=1}^{T}\left[\frac{1}{2\eta_t}\left\{\|\mathbf{x}^* - \hat{\mathbf{x}}_t\|_2^2 - \|\mathbf{x}^* - \hat{\mathbf{x}}_{t+1}\|_2^2\right\} - \frac{\lambda}{2}\|\mathbf{x}^* - \mathbf{x}_t\|_2^2\right]}_{\texttt{term (b)}} - \underbrace{\sum_{t=1}^{T}\frac{1}{2\eta_t}\left\{\|\hat{\mathbf{x}}_{t+1} - \mathbf{x}_t\|_2^2 + \|\mathbf{x}_t - \hat{\mathbf{x}}_t\|_2^2\right\}}_{\texttt{term (c)}}.$$

Next, we analyze the above three terms separately. For the $\texttt{term (a)}$, we substitute $\eta \leq \frac{2}{\lambda t}$ and obtain

$$\texttt{term (a)} \leq \sum_{t=1}^{T}\frac{2}{\lambda t}\|\nabla f_t(\mathbf{x}_t) - \nabla f_{t-1}(\mathbf{x}_{t-1})\|_2^2. \qquad (29)$$

For the $\texttt{term (b)}$, we exploit one result of Lemma 1 in (52), i.e.,

$$\|\mathbf{x}_t - \hat{\mathbf{x}}_{t+1}\|_2 \leq \eta_t\|\nabla f_t(\mathbf{x}_t) - \nabla f_{t-1}(\mathbf{x}_{t-1})\|_2, \qquad (30)$$

with $\|\cdot\| = \frac{1}{\sqrt{\eta_t}}\|\cdot\|_2$ and $\|\cdot\|_* = \sqrt{\eta_t}\|\cdot\|_2$. By utilizing (30), we can upper bound the $\texttt{term (b)}$ as following:

$$\texttt{term (b)} \leq \frac{D^2}{2\eta_1} + \sum_{t=2}^{T}\left(\frac{1}{2\eta_t} - \frac{1}{2\eta_{t-1}}\right)\|\mathbf{x}^* - \hat{\mathbf{x}}_t\|_2^2 - \frac{\lambda}{2}\sum_{t=1}^{T}\|\mathbf{x}^* - \mathbf{x}_t\|_2^2$$

$$\leq \frac{D^2\lambda}{4} + \frac{\lambda}{4}\sum_{t=1}^{T-1}\left[\|\mathbf{x}^* - \hat{\mathbf{x}}_{t+1}\|_2^2 - 2\|\mathbf{x}^* - \mathbf{x}_t\|_2^2\right]$$

$$\leq \frac{D^2\lambda}{4} + \frac{\lambda}{2}\sum_{t=1}^{T-1}\|\mathbf{x}_t - \hat{\mathbf{x}}_{t+1}\|_2^2 \overset{(30)}{\leq} \frac{D^2\lambda}{4} + \frac{\lambda}{2}\sum_{t=1}^{T-1}\eta_t^2\|\nabla f_t(\mathbf{x}_t) - \nabla f_{t-1}(\mathbf{x}_{t-1})\|_2 \qquad (31)$$

$$\leq \frac{D^2\lambda}{4} + \frac{\lambda\eta_1}{2}\sum_{t=1}^{T-1}\eta_t\|\nabla f_t(\mathbf{x}_t) - \nabla f_{t-1}(\mathbf{x}_{t-1})\|_2 \leq \frac{D^2\lambda}{4} + \texttt{term (a)}.$$

For the $\texttt{term (c)}$, we make use of the non-increasing property of $\eta_t$ and $\eta_t \leq 2/\delta$, and obtain

$$\texttt{term (c)} \geq \sum_{t=2}^{T}\left\{\frac{1}{2\eta_{t-1}}\|\hat{\mathbf{x}}_t - \mathbf{x}_{t-1}\|_2^2 + \frac{1}{2\eta_{t-1}}\|\mathbf{x}_t - \hat{\mathbf{x}}_t\|_2^2\right\}$$

$$\geq \sum_{t=2}^{T}\frac{1}{4\eta_{t-1}}\|\mathbf{x}_t - \mathbf{x}_{t-1}\|_2^2 \geq \frac{\delta}{8}\sum_{t=2}^{T}\|\mathbf{x}_t - \mathbf{x}_{t-1}\|_2^2. \qquad (32)$$

Putting (29), (31) and (32) together, we obtain

$$\sum_{t=1}^{T}\langle \nabla f_t(\mathbf{x}_t), \mathbf{x}_t - \mathbf{x}^*\rangle + r(\mathbf{x}_t) - r(\mathbf{x}^*) - \frac{\lambda}{2}\sum_{t=1}^{T}\|\mathbf{x}^* - \mathbf{x}_t\|_2^2$$

$$\leq \frac{D^2\lambda}{4} + \sum_{t=1}^{T}\frac{4}{\lambda t}\|\nabla f_t(\mathbf{x}_t) - \nabla f_{t-1}(\mathbf{x}_{t-1})\|_2^2 - \frac{\delta}{8}\sum_{t=2}^{T}\|\mathbf{x}_t - \mathbf{x}_{t-1}\|_2^2. \qquad (33)$$

Now, we make use of a byproduct (55) from Lemma 3, i.e.,

$$\|\nabla f_t(\mathbf{x}_t) - \nabla f_{t-1}(\mathbf{x}_{t-1})\|_2^2 \le 4\|\nabla f_t(\mathbf{x}_t) - \nabla F_t(\mathbf{x}_t)\|_2^2 + 4H^2\|\mathbf{x}_t - \mathbf{x}_{t-1}\|_2^2$$
$$+ 4\|\nabla F_t(\mathbf{x}_{t-1}) - \nabla F_{t-1}(\mathbf{x}_{t-1})\|_2^2 + 4\|\nabla F_{t-1}(\mathbf{x}_{t-1}) - \nabla f_{t-1}(\mathbf{x}_{t-1})\|_2^2,$$

and thus (33) becomes

$$\mathbb{E}\left[\sum_{t=1}^{T}\langle\nabla f_t(\mathbf{x}_t), \mathbf{x}_t - \mathbf{x}^*\rangle + r(\mathbf{x}_t) - r(\mathbf{x}^*) - \frac{\lambda}{2}\sum_{t=1}^{T}\|\mathbf{x}^* - \mathbf{x}_t\|_2^2\right]$$
$$\le \sum_{t=1}^{T}\frac{16}{\lambda t}(2\sigma_t^2 + \sup_{\mathbf{x}\in\mathcal{X}}\|\nabla F_t(\mathbf{x}) - \nabla F_{t-1}(\mathbf{x})\|_2^2) \tag{34}$$
$$+ \sum_{t=1}^{T}\frac{16H^2}{\lambda t}\|\mathbf{x}_t - \mathbf{x}_{t-1}\|_2^2 - \frac{\delta}{8}\sum_{t=2}^{T}\|\mathbf{x}_t - \mathbf{x}_{t-1}\|_2^2 + \frac{D^2\lambda}{4}.$$

To bound the first two terms, we introduce the following lemma.

**Lemma 5.** *[Yan et al., 2023, Lemma 9] For a sequence of $\{a_t\}_{t=1}^{T}$ and $b$, where $a_t, b > 0$ for any $t \in [T]$, denoting by $a_{\max} \triangleq \max_t a_t$ and $A \triangleq \lceil b\sum_{t=1}^{T} a_t\rceil$, we have*

$$\sum_{t=1}^{T}\frac{a_t}{bt} \le \frac{a_{\max}}{b}(1 + \ln A) + \frac{1}{b^2}.$$

Let $a_t = 2\sigma_t^2 + \sup_{\mathbf{x}\in\mathcal{X}}\|\nabla F_t(\mathbf{x}) - \nabla F_{t-1}(\mathbf{x})\|_2^2$, $a_{\max} = 2\sigma_{\max}^2 + \Sigma_{\max}^2$, $b = \lambda$ and $A = \lceil\lambda(2\sigma_{1:T}^2 + \Sigma_{1:t}^2)\rceil$. Then, applying Lemma 5, we obtain

$$\sum_{t=1}^{T}\frac{1}{\lambda t}(2\sigma_t^2 + \sup_{\mathbf{x}\in\mathcal{X}}\|\nabla F_t(\mathbf{x}) - \nabla F_{t-1}(\mathbf{x})\|_2^2) \le \frac{2\sigma_{\max}^2 + \Sigma_{\max}^2}{\lambda}\left(1 + \ln(1 + \lambda(2\sigma_{1:T}^2 + \Sigma_{1:T}^2))\right) + \frac{1}{\lambda^2}. \tag{35}$$

Let $a_t = \|\mathbf{x}_t - \mathbf{x}_{t-1}\|_2^2$, $a_{\max} = D^2$, $b = \lambda$ and $A = \lceil\lambda\sum_{t=1}^{T}\|\mathbf{x}_t - \mathbf{x}_{t-1}\|_2^2\rceil$. Applying Lemma 5, we obtain

$$\sum_{t=1}^{T}\frac{1}{\lambda t}\|\mathbf{x}_t - \mathbf{x}_{t-1}\|_2^2 \le \frac{D^2}{\lambda}\left(1 + \ln(1 + \lambda\sum_{t=1}^{T}\|\mathbf{x}_t - \mathbf{x}_{t-1}\|_2^2)\right) + \frac{1}{\lambda^2}$$
$$\le \frac{D^2}{\lambda}\left(1 + \lambda\sum_{t=1}^{T}\|\mathbf{x}_t - \mathbf{x}_{t-1}\|_2^2\right) + \frac{1}{\lambda^2}, \tag{36}$$

where the last step is due to $\ln(1 + x) \le x$ for any $x \ge 0$.

Substituting (35) and (36) into (34), we have

$$\mathbb{E}\left[\sum_{t=1}^{T}\langle\nabla f_t(\mathbf{x}_t), \mathbf{x}_t - \mathbf{x}^*\rangle + r(\mathbf{x}_t) - r(\mathbf{x}^*) - \frac{\lambda}{2}\sum_{t=1}^{T}\|\mathbf{x}^* - \mathbf{x}_t\|_2^2\right]$$
$$\le \mathcal{O}\left(\frac{\sigma_{\max}^2 + \Sigma_{\max}^2}{\lambda}\ln(\sigma_{1:T}^2 + \Sigma_{1:T}^2)\right) + \left(16H^2D^2 - \frac{\delta}{8}\right)\sum_{t=2}^{T}\|\mathbf{x}_t - \mathbf{x}_{t-1}\|_2^2 + \mathcal{O}(1).$$

Setting $\delta = 128H^2D^2$ finishes the proof.

### B.4 Proof of Theorem 4

Similar to (28) in Theorem 3, the instantaneous regret is upper bounded by

$$\mathbb{E}\left[[f_t(\mathbf{x}_t) + r(\mathbf{x}_t)] - [f_t(\mathbf{x}^*) + r(\mathbf{x}^*)]\right] = \mathbb{E}\left[\langle\nabla f_t(\mathbf{x}_t), \mathbf{x}_t - \mathbf{x}^*\rangle + r(\mathbf{x}_t) - r(\mathbf{x}^*) - \frac{\lambda}{2}\|\mathbf{x}^* - \mathbf{x}_t\|_2^2\right].$$

According to [Scroccaro et al., 2023, Theorem 2.9], OptCMD with the configuration in (9) ensures

$$
\sum_{t=1}^{T} \langle \nabla f_t(\mathbf{x}_t), \mathbf{x}_t - \mathbf{x}^* \rangle + r(\mathbf{x}_t) - r(\mathbf{x}^*) - \frac{\lambda}{2} \|\mathbf{x}^* - \mathbf{x}_t\|_2^2
$$

$$
\leq 2HD^2 + \frac{G^2}{H} + \frac{4G^2}{\lambda} \log\left(1 + \frac{\lambda}{4HG^2} \bar{D}_T\right).
$$

(37)

For the term $\log\left(1 + \frac{\lambda}{4HG^2} \bar{D}_T\right)$, we apply Lemma 4 and obtain

$$
\log\left(1 + \frac{\lambda}{4HG^2} \bar{D}_T\right) \leq \log\left(1 + \frac{\lambda}{4H}\right)
$$

$$
+ \log\left(1 + \frac{\lambda}{8HG^2} \left(3 \sum_{t=1}^{T} \sup_{\mathbf{x} \in X} \|\nabla f_t(\mathbf{x}) - \nabla F_t(\mathbf{x})\|_2^2 + 2 \sum_{t=1}^{T} \sup_{\mathbf{x} \in X} \|\nabla F_t(\mathbf{x}) - \nabla F_{t-1}(\mathbf{x})\|_2^2\right)\right).
$$

Taking expectation over both sides of the above inequality delivers

$$
\mathbb{E}\left[\log\left(1 + \frac{\lambda}{4HG^2} \bar{D}_T\right)\right] \leq \log\left(1 + \frac{\lambda}{4H}\right) + \log\left(1 + \frac{\lambda}{8HG^2} \left(3\tilde{\sigma}_{1:T}^2 + 2\Sigma_{1:T}^2\right)\right)
$$

Plug the above result into (37) finishes the proof.

## B.5  Proof of Theorem 5

For the exp-concave setting (i.e., $f_t(\cdot)$ is exp-concave and $r(\cdot)$ is general convex), our parameter configuration is follow Chiang et al. [2012]. To be precise, we define $H_t = \delta I + \frac{\beta}{2} G^2 I + \frac{\beta}{2} \sum_{r=1}^{t-1} h_r$, where $h_r = \nabla f_r(\mathbf{x}_r) \nabla f_r(\mathbf{x}_r)^\top$. The norm is set as $\|\cdot\| = \|\cdot\|_{H_t}$ and $\|\cdot\|_* = \|\cdot\|_{H_t^{-1}}$, and the function becomes $\mathcal{R}_t(\mathbf{x}) = \frac{1}{2}\|\mathbf{x}\|_{H_t}^2$. With this $\mathcal{R}_t(\mathbf{x})$, the corresponding Bregman divergence becomes $\mathcal{B}^{\mathcal{R}_t}(\mathbf{x}, \mathbf{y}) = \frac{1}{2}\|\mathbf{x} - \mathbf{y}\|_{H_t}^2$.

First, we introduce a common property of exp-concave function, as shown below.

**Lemma 6.** *[Hazan et al., 2007, Lemma 3] Under Assumption 1 and 2, if $f(\cdot)$ is exp-concave over $\mathcal{X}$, we have*

$$
f(\mathbf{y}) \geq f(\mathbf{x}) + \langle \nabla f(\mathbf{x}), \mathbf{y} - \mathbf{x} \rangle + \frac{\beta}{2} \langle \nabla f(\mathbf{x}), \mathbf{y} - \mathbf{x} \rangle^2,
$$

*for $\forall \mathbf{x}, \mathbf{y} \in \mathcal{X}$ and $\beta = \frac{1}{2} \min\{\frac{1}{4GD}, \alpha\}$.*

Then, according to the exponential concavity of $f_t(\cdot)$ shown in Lemma 6, we can decompose the instantaneous regret as below:

$$
\mathbb{E}\left[[f_t(\mathbf{x}_t) + r(\mathbf{x}_t)] - [f_t(\mathbf{x}^*) + r(\mathbf{x}^*)]\right]
$$

$$
\leq \mathbb{E}\left[\langle \nabla f_t(\mathbf{x}_t), \mathbf{x}_t - \mathbf{x}^* \rangle + r(\mathbf{x}_t) - r(\mathbf{x}^*) - \frac{\beta}{2} \langle \nabla f_t(\mathbf{x}_t), \mathbf{x}^* - \mathbf{x}_t \rangle^2\right].
$$

Recall that we denote $h_t = \nabla f_t(\mathbf{x}_t) \nabla f_t(\mathbf{x}_t)^\top$. Hence, the last term in (38) can be rewrite as $\langle \nabla f_t(\mathbf{x}_t), \mathbf{x}^* - \mathbf{x}_t \rangle^2 = \|\mathbf{x}^* - \mathbf{x}_t\|_{h_t}^2$.

Now, we focus on the right side of (38). By utilizing Lemma 1, we have

$$
\sum_{t=1}^{T} \langle \nabla f_t(\mathbf{x}_t), \mathbf{x}_t - \mathbf{x}^* \rangle + r(\mathbf{x}_t) - r(\mathbf{x}^*) - \frac{\beta}{2} \sum_{t=1}^{T} \|\mathbf{x}^* - \mathbf{x}_t\|_{h_t}^2 \leq \underbrace{\sum_{t=1}^{T} \|\nabla f_t(\mathbf{x}_t) - \nabla f_{t-1}(\mathbf{x}_{t-1})\|_{H_t^{-1}}^2}_{\texttt{term (a)}}
$$

$$
+ \underbrace{\frac{1}{2} \sum_{t=1}^{T} \left[\|\mathbf{x}^* - \hat{\mathbf{x}}_t\|_{H_t}^2 - \|\mathbf{x}^* - \hat{\mathbf{x}}_{t+1}\|_{H_t}^2 - \beta\|\mathbf{x}^* - \mathbf{x}_t\|_{h_t}^2\right]}_{\texttt{term (b)}} - \underbrace{\frac{1}{2} \sum_{t=1}^{T} \left[\|\hat{\mathbf{x}}_{t+1} - \mathbf{x}_t\|_{H_t}^2 + \|\mathbf{x}_t - \hat{\mathbf{x}}_t\|_{H_t}^2\right]}_{\texttt{term (c)}}.
$$

The `term (a)` can be upper bounded as shown in the following lemma.

**Lemma 7.** *Let $d$ be the dimension. Then, we have*

$$\texttt{term (a)} \leq \frac{8d}{\beta} \ln \left( 1 + \frac{\beta}{8\delta} \sum_{t=1}^{T} \|\nabla f_t(\mathbf{x}_t) - \nabla f_{t-1}(\mathbf{x}_{t-1})\|_2^2 \right). \tag{38}$$

For the $\texttt{term (b)}$, we exploit the fact that $H_{t+1} - H_t = \frac{\beta}{2} h_t$ and obtain

$$
\begin{aligned}
\texttt{term (b)} =& \frac{1}{2} \left[ \|\mathbf{x}^* - \hat{\mathbf{x}}_1\|_{H_1}^2 - \|\mathbf{x}^* - \hat{\mathbf{x}}_{T+1}\|_{H_{T+1}}^2 \right] \\
&+ \frac{1}{2} \sum_{t=1}^{T} \left\{ \|\mathbf{x}^* - \hat{\mathbf{x}}_{t+1}\|_{H_{t+1}}^2 - \|\mathbf{x}^* - \hat{\mathbf{x}}_{t+1}\|_{H_t}^2 - \beta\|\mathbf{x}^* - \mathbf{x}_t\|_{h_t}^2 \right\} \\
=& \frac{1}{2} \left[ \|\mathbf{x}^* - \hat{\mathbf{x}}_1\|_{H_1}^2 - \|\mathbf{x}^* - \hat{\mathbf{x}}_{T+1}\|_{H_{T+1}}^2 \right] + \frac{\beta}{4} \sum_{t=1}^{T} \left\{ \|\mathbf{x}^* - \hat{\mathbf{x}}_{t+1}\|_{h_t}^2 - 2\|\mathbf{x}^* - \mathbf{x}_t\|_{h_t}^2 \right\} \\
\overset{(1)}{\leq}& \left( \frac{\delta}{2} + \frac{\beta}{4} G^2 \right) D^2 + \frac{\beta}{4} \sum_{t=1}^{T} \left\{ \|\mathbf{x}^* - \hat{\mathbf{x}}_{t+1}\|_{h_t}^2 - 2\|\mathbf{x}^* - \mathbf{x}_t\|_{h_t}^2 \right\}. \tag{39}
\end{aligned}
$$

To proceed the proof, we introduce the following lemma.

**Lemma 8.** *[Chiang et al., 2012, Proposition 1] For any $y, z \in \mathbb{R}^N$ and any PSD $H \in \mathbb{R}^{N \times N}$, we have $\|y + z\|_H^2 \leq 2\|y\|_H^2 + 2\|z\|_H^2$.*

Applying Lemma 8 on (39) obtains:

$$
\begin{aligned}
\texttt{term (b)} \leq& \left( \frac{\delta}{2} + \frac{\beta}{4} G^2 \right) D^2 + \frac{\beta}{4} \sum_{t=1}^{T} \left\{ 2\|\mathbf{x}^* - \mathbf{x}_t\|_{h_t}^2 + 2\|\mathbf{x}_t - \hat{\mathbf{x}}_{t+1}\|_{h_t}^2 - 2\|\mathbf{x}^* - \mathbf{x}_t\|_{h_t}^2 \right\} \\
\leq& \left( \frac{\delta}{2} + \frac{\beta}{4} G^2 \right) D^2 + \frac{\beta}{2} \sum_{t=1}^{T} \|\mathbf{x}_t - \hat{\mathbf{x}}_{t+1}\|_{h_t}^2 \leq \left( \frac{1}{2} + \frac{\beta}{4} G^2 \right) D^2 + \sum_{t=1}^{T} \|\mathbf{x}_t - \hat{\mathbf{x}}_{t+1}\|_{H_t}^2 \\
\leq& \left( \frac{\delta}{2} + \frac{\beta}{4} G^2 \right) D^2 + \sum_{t=1}^{T} \|\nabla f_t(\mathbf{x}_t) - \nabla f_{t-1}(\mathbf{x}_{t-1})\|_{H_t^{-1}},
\end{aligned}
$$
$$\tag{40}$$

where the penultimate step is due to $H_t \succeq \frac{\beta}{2} G^2 I \succeq \frac{\beta}{2} h_t$, and the last step is due to

$$\|\mathbf{x}_t - \hat{\mathbf{x}}_{t+1}\|_{H_t} \leq \|\nabla f_t(\mathbf{x}_t) - \nabla f_{t-1}(\mathbf{x}_{t-1})\|_{H_t^{-1}}, \tag{41}$$

which could be obtained from Lemma 1 with $\|\cdot\| = \|\cdot\|_{H_t}$ and $\|\cdot\|_* = \|\cdot\|_{H_t^{-1}}$.

For the $\texttt{term (c)}$, we have

$$
\begin{aligned}
\texttt{term (c)} =& \frac{1}{2} \sum_{t=2}^{T+1} \|\hat{\mathbf{x}}_t - \mathbf{x}_{t-1}\|_{H_{t-1}}^2 + \frac{1}{2} \sum_{t=1}^{T} \|\mathbf{x}_t - \hat{\mathbf{x}}_t\|_{H_t}^2 \\
\geq& \frac{\delta}{2} \sum_{t=2}^{T} \left\{ \|\hat{\mathbf{x}}_t - \mathbf{x}_{t-1}\|_2^2 + \|\mathbf{x}_t - \hat{\mathbf{x}}_t\|_2^2 \right\} \geq \frac{\delta}{4} \sum_{t=2}^{T} \|\mathbf{x}_t - \mathbf{x}_{t-1}\|_2^2,
\end{aligned}
$$
$$\tag{42}$$

where the first inequality is due to $H_t \succeq H_{t-1} \succeq \delta I$, $\|\hat{\mathbf{x}}_{T+1} - \mathbf{x}_T\|_{H_{T+1}}^2 \geq 0$ and $\|\mathbf{x}_1 - \hat{\mathbf{x}}_1\|_{H_1}^2 \geq 0$. Combining (38), (40) and (42) obtains

$$
\begin{aligned}
&\sum_{t=1}^{T} \langle \nabla f_t(\mathbf{x}_t), \mathbf{x}_t - \mathbf{x}^* \rangle + r(\mathbf{x}_t) - r(\mathbf{x}^*) - \frac{\beta}{2} \sum_{t=1}^{T} \|\mathbf{x}^* - \mathbf{x}_t\|_{h_t}^2 \\
&\leq \left( \frac{\delta}{2} + \frac{\beta}{4} G^2 \right) D^2 + \frac{24d}{\beta} \ln \left( 1 + \frac{\beta}{8\delta} \sum_{t=1}^{T} \|\nabla f_t(\mathbf{x}_t) - \nabla f_{t-1}(\mathbf{x}_{t-1})\|_2^2 \right) - \frac{\delta}{4} \sum_{t=2}^{T} \|\mathbf{x}_t - \mathbf{x}_{t-1}\|_2^2.
\end{aligned}
$$

Next, we leverage Lemma 3 and arrive at

$$
\sum_{t=1}^{T} \langle \nabla f_t(\mathbf{x}_t), \mathbf{x}_t - \mathbf{x}^* \rangle + r(\mathbf{x}_t) - r(\mathbf{x}^*) - \frac{\beta}{2} \sum_{t=1}^{T} \|\mathbf{x}^* - \mathbf{x}_t\|_{h_t}^2
$$

$$
\leq \left( \frac{\delta}{2} + \frac{\beta}{4} G^2 \right) D^2 - \frac{\delta}{4} \sum_{t=2}^{T} \|\mathbf{x}_t - \mathbf{x}_{t-1}\|_2^2 + \frac{16d}{\beta} \ln \left( 1 + \frac{\beta}{8\delta} \left[ G^2 + 8 \sum_{t=1}^{T} \|\nabla f_t(\mathbf{x}_t) - \nabla F_t(\mathbf{x}_t)\|_2^2 \right. \right.
$$

$$
+ \left. \left. 4 \sum_{t=2}^{T} \|\nabla F_t(\mathbf{x}_{t-1}) - \nabla F_{t-1}(\mathbf{x}_{t-1})\|_2^2 \right] + \frac{\beta H^2}{2\delta} \sum_{t=2}^{T} \|\mathbf{x}_t - \mathbf{x}_{t-1}\|_2^2 \right)
$$

$$
\leq \left( \frac{\delta}{2} + \frac{\beta}{4} G^2 \right) D^2 - \frac{\delta}{4} \sum_{t=2}^{T} \|\mathbf{x}_t - \mathbf{x}_{t-1}\|_2^2 + \frac{16d}{\beta} \ln \left( 1 + \frac{\beta}{8\delta} \left[ G^2 + 8 \sum_{t=1}^{T} \|\nabla f_t(\mathbf{x}_t) - \nabla F_t(\mathbf{x}_t)\|_2^2 \right. \right.
$$

$$
+ \left. \left. 4 \sum_{t=2}^{T} \|\nabla F_t(\mathbf{x}_{t-1}) - \nabla F_{t-1}(\mathbf{x}_{t-1})\|_2^2 \right] \right) + \frac{16d}{\beta} \ln \left( 1 + \frac{\beta H^2}{2\delta} \sum_{t=2}^{T} \|\mathbf{x}_t - \mathbf{x}_{t-1}\|_2^2 \right),
$$

(43)

where the last step is due to the inequality

$$
\ln(1 + u + v) \leq \ln(1 + u) + \ln(1 + v), \ \forall u, v \geq 0. \tag{44}
$$

To simplify (43), we make use of the following lemma.

**Lemma 9.** *[Chen et al., 2023, Lemma 7] Let $A \geq 0$, $a \geq 0$, $b \geq 0$ and $c > 0$, we have*

$$
a \ln(bA + 1) - cA \leq a \ln \left( \frac{ab}{c} + 1 \right).
$$

From Lemma 9, we have

$$
\frac{16d}{\beta} \ln \left( \frac{\beta H^2}{2\delta} \sum_{t=1}^{T} \|\mathbf{x}_t - \mathbf{x}_{t-1}\|_2^2 + 1 \right) - \frac{\delta}{4} \sum_{t=1}^{T} \|\mathbf{x}_t - \mathbf{x}_{t-1}\|_2^2 \leq \frac{16d}{\beta} \ln \left( 32dH^2 + 1 \right), \tag{45}
$$

where we set $\delta = 1$ in the last step. Combining (43) and (45), we have

$$
\sum_{t=1}^{T} \langle \nabla f_t(\mathbf{x}_t) + \nabla r(\mathbf{x}_t), \mathbf{x}_t - \mathbf{x}^* \rangle - \frac{\beta}{2} \sum_{t=1}^{T} \|\mathbf{x}^* - \mathbf{x}_t\|_{h_t}^2
$$

$$
\leq \left( \frac{1}{2} + \frac{\beta}{4} G^2 \right) D^2 + \frac{16d}{\beta} \ln \left( 32dH^2 + 1 \right) + \frac{16d}{\beta} \ln \left( 1 + \frac{\beta}{8} \left[ G^2 + 8 \sum_{t=1}^{T} \|\nabla f_t(\mathbf{x}_t) - \nabla F_t(\mathbf{x}_t)\|_2^2 \right. \right.
$$

$$
+ \left. \left. 4 \sum_{t=2}^{T} \|\nabla F_t(\mathbf{x}_{t-1}) - \nabla F_{t-1}(\mathbf{x}_{t-1})\|_2^2 \right] \right).
$$

Taking the expectation over both sides delivers

$$
\mathbb{E} \left[ \sum_{t=1}^{T} \langle \nabla f_t(\mathbf{x}_t) + \nabla r(\mathbf{x}_t), \mathbf{x}_t - \mathbf{x}^* \rangle - \frac{\beta}{2} \sum_{t=1}^{T} \|\mathbf{x}^* - \mathbf{x}_t\|_{h_t}^2 \right]
$$

$$
\leq \left( \frac{1}{2} + \frac{\beta}{4} G^2 \right) D^2 + \frac{16d}{\beta} \ln \left( 32dH^2 + 1 \right) + \frac{16d}{\beta} \ln \left( 1 + \frac{\beta}{8} G^2 + \beta \sigma_{1:T}^2 + \frac{\beta}{2} \Sigma_{1:T}^2 \right),
$$

where the last step is due to the Jensen's inequality. Finally, we complete the proof by substitute the above result into (38).

### B.6 Proof of Theorem 6

In this part, we analyze three types of functions separately.

### B.6.1 For convex functions

First, since the expected function $F_t(\cdot)$ is convex in each round $t$, the instantaneous dynamic regret could be upper bounded as below:

$$\mathbb{E}\left[[f_t(\mathbf{x}_t) + r(\mathbf{x}_t)] - [f_t(\mathbf{x}^*) + r(\mathbf{x}^*)]\right] = \mathbb{E}\left[[F_t(\mathbf{x}_t) + r(\mathbf{x}_t)] - [F_t(\mathbf{x}^*) + r(\mathbf{x}^*)]\right]$$
$$\leq \mathbb{E}\left[\langle\nabla F_t(\mathbf{x}_t), \mathbf{x}_t - \mathbf{x}^*\rangle + r(\mathbf{x}_t) - r(\mathbf{x}^*)\right] = \mathbb{E}\left[\langle\nabla f_t(\mathbf{x}_t), \mathbf{x}_t - \mathbf{x}^*\rangle + r(\mathbf{x}_t) - r(\mathbf{x}^*)\right].$$

Then, we decompose the regret into meta-regret and expert-regret:

$$\mathbb{E}\left[\mathrm{Regret}_T\right] \leq \mathbb{E}\left[\sum_{t=1}^{T}\langle\nabla f_t(\mathbf{x}_t), \mathbf{x}_t - \mathbf{x}^*\rangle + r(\mathbf{x}_t) - r(\mathbf{x}^*)\right]$$

$$= \underbrace{\mathbb{E}\left[\sum_{t=1}^{T}\langle\nabla f_t(\mathbf{x}_t), \mathbf{x}_t - \mathbf{x}_t^{k,i}\rangle + r(\mathbf{x}_t) - r(\mathbf{x}_t^{k,i})\right]}_{\text{meta-regret}} + \underbrace{\mathbb{E}\left[\sum_{t=1}^{T}\langle\nabla f_t(\mathbf{x}_t), \mathbf{x}_t^{k,i} - \mathbf{x}^*\rangle + r(\mathbf{x}_t^{k,i}) - r(\mathbf{x}^*)\right]}_{\text{expert-regret}}.$$

**Expert-regret analysis.** According to the above decomposition, we define the surrogate loss functions $h_t^c(\mathbf{x}) = \langle\nabla f_t(\mathbf{x}_t), \mathbf{x}\rangle + r(\mathbf{x})$ for the $i$-th expert. According to (24), we can bound the expert-regret by

$$\text{expert-regret} \leq \mathbb{E}\left[\frac{3D}{2}\sqrt{1 + \bar{V}_T} + \frac{D\delta}{2} - \frac{\delta}{4D}\sum_{t=2}^{T}\|\mathbf{x}_t^{k,i} - \mathbf{x}_{t-1}^{k,i}\|_2^2 + 4DG^2\right]$$

$$\leq \frac{3D}{2}(1 + G) + 4DG^2 + 12D\sqrt{\sigma_{1:T}^2} + 6D\sqrt{\Sigma_{1:T}^2}$$

$$+ 10HD\sum_{t=2}^{T}\|\mathbf{x}_t - \mathbf{x}_{t-1}\|_2^2 - \frac{\delta}{4D}\sum_{t=2}^{T}\|\mathbf{x}_t^{k,i} - \mathbf{x}_{t-1}^{k,i}\|_2^2,$$

where the last step is due to Lemma 3.

**Meta-regret analysis.** We first consider the instantaneous meta-regret:

$$\langle\nabla f_t(\mathbf{x}_t), \mathbf{x}_t - \mathbf{x}_t^{k,i}\rangle + r(\mathbf{x}_t) - r(\mathbf{x}_t^{k,i})$$

$$= \langle\nabla f_t(\mathbf{x}_t), \mathbf{x}_t - \mathbf{x}_t^k\rangle + r(\mathbf{x}_t) - r(\mathbf{x}_t^k) + \langle\nabla f_t(\mathbf{x}_t), \mathbf{x}_t^k - \mathbf{x}_t^{k,i}\rangle + r(\mathbf{x}_t^k) - r(\mathbf{x}_t^{k,i})$$

$$\leq \langle\nabla f_t(\mathbf{x}_t), \mathbf{x}_t - \mathbf{x}_t^k\rangle + \sum_{k=1}^{K} q_t^k r(\mathbf{x}_t^k) - r(\mathbf{x}_t^k) + \langle\nabla f_t(\mathbf{x}_t), \mathbf{x}_t^k - \mathbf{x}_t^{k,i}\rangle + \sum_{i=1}^{N} p_t^{k,i} r(\mathbf{x}_t^{k,i}) - r(\mathbf{x}_t^{k,i})$$

$$\leq \sum_{k=1}^{K} q_t^k l_t^k - l_t^k + \sum_{i=1}^{N} p_t^{k,i} l_t^{k,i} - l_t^{k,i} - \gamma_1\sum_{k=1}^{K} q_t^k\|\mathbf{x}_t^k - \mathbf{x}_{t-1}^k\|_2^2 + \gamma_1\|\mathbf{x}_t^k - \mathbf{x}_{t-1}^k\|_2^2$$

$$- \gamma_2\sum_{i=1}^{N} p_t^{k,i}\|\mathbf{x}_t^{k,i} - \mathbf{x}_{t-1}^{k,i}\|_2^2 + \gamma_2\|\mathbf{x}_t^{k,i} - \mathbf{x}_{t-1}^{k,i}\|_2^2$$

$$= \langle \boldsymbol{q}_t - \mathbf{e}_k, \boldsymbol{l}_t\rangle + \left\langle \boldsymbol{p}_t^k - \mathbf{e}_i, \boldsymbol{l}_t^k\right\rangle - \gamma_1\sum_{k=1}^{K} q_t^k\|\mathbf{x}_t^k - \mathbf{x}_{t-1}^k\|_2^2 + \gamma_1\|\mathbf{x}_t^k - \mathbf{x}_{t-1}^k\|_2^2$$

$$- \gamma_2\sum_{i=1}^{N} p_t^{k,i}\|\mathbf{x}_t^{k,i} - \mathbf{x}_{t-1}^{k,i}\|_2^2 + \gamma_2\|\mathbf{x}_t^{k,i} - \mathbf{x}_{t-1}^{k,i}\|_2^2,$$

$$(46)$$

where the second step is due to Jensen's inequality, i.e., $r(\mathbf{x}_t) = r(\sum_{k=1}^{K} q_t^k \mathbf{x}_t^k) \leq \sum_{k=1}^{K} q_t^k r(\mathbf{x}_t^k)$ and $r(\mathbf{x}_t^k) = r(\sum_{i=1}^{N} p_t^{k,i} \mathbf{x}_t^{k,i}) \leq \sum_{i=1}^{N} p_t^{k,i} r(\mathbf{x}_t^{k,i})$, and the third step is due to the definition of $l_t^k$ and $l_t^{k,i}$ in (13) and (14). For brevity, we denote $\boldsymbol{q}_t \triangleq (q_t^1, \cdots, q_t^K)$, $\boldsymbol{l}_t \triangleq (l_t^1, \cdots, l_t^K)$,

$p_t^k \triangleq (q_t^{k,1}, \cdots, q_t^{k,N})$ and $l_t^k \triangleq (l_t^{k,1}, \cdots, l_t^{k,N})$. By the above results, we have

$$\sum_{t=1}^{T} \langle \nabla f_t(\mathbf{x}_t), \mathbf{x}_t - \mathbf{x}_t^{k,i} \rangle + r(\mathbf{x}_t) - r(\mathbf{x}_t^{k,i})$$

$$\leq \sum_{t=1}^{T} \langle q_t - \mathbf{e}_k, l_t \rangle + \left\langle p_t^k - \mathbf{e}_i, l_t^k \right\rangle - \gamma_1 \sum_{t=1}^{T} \sum_{k=1}^{K} q_t^k \|\mathbf{x}_t^k - \mathbf{x}_{t-1}^k\|_2^2 + \gamma_1 \sum_{t=1}^{T} \|\mathbf{x}_t^k - \mathbf{x}_{t-1}^k\|_2^2 \quad (47)$$

$$- \gamma_2 \sum_{t=1}^{T} \sum_{i=1}^{N} p_t^{k,i} \|\mathbf{x}_t^{k,i} - \mathbf{x}_{t-1}^{k,i}\|_2^2 + \gamma_2 \sum_{t=1}^{T} \|\mathbf{x}_t^{k,i} - \mathbf{x}_{t-1}^{k,i}\|_2^2.$$

Then, we introduce the following lemma, which is similar to Yan et al. [2023, Lemma 3] but with a different composite losses and optimisms in (13) and (14).

**Lemma 10.** *Let* $l_t^k$, $m_t^k$ *be defined in* (13), *and* $l_t^{k,i}$, $m_t^{k,i}$ *be defined in* (14). *Then, we have*

$$\sum_{t=1}^{T} \langle q_t - \mathbf{e}_k, l_t \rangle + \left\langle p_t^k - \mathbf{e}_i, l_t^k \right\rangle \leq \frac{1}{\eta^k} \ln \frac{N}{3(C_0 \eta^k)^2} + 32\eta^k \sum_{t=1}^{T} \left( l_t^{k,i} - m_t^{k,i} \right)^2$$

$$- \frac{C_0}{2} \sum_{t=2}^{T} \|q_t - q_{t-1}\|_1^2 - \frac{C_0}{16} \sum_{t=2}^{T} \|p_t^k - p_{t-1}^k\|_1^2.$$

Note that

$$\sum_{t=1}^{T} \left( l_t^{k,i} - m_t^{k,i} \right)^2 = \sum_{t=1}^{T} \left( \left\langle \nabla f_t(\mathbf{x}_t), \mathbf{x}_t - \mathbf{x}_t^{k,i} \right\rangle - \left\langle \nabla f_{t-1}(\mathbf{x}_{t-1}), \mathbf{x}_{t-1} - \mathbf{x}_{t-1}^{k,i} \right\rangle \right)^2$$

$$\leq 2 \sum_{t=1}^{T} \left\langle \nabla f_t(\mathbf{x}_t) - \nabla f_{t-1}(\mathbf{x}_t), \mathbf{x}_t - \mathbf{x}_t^{k,i} \right\rangle^2 + 2 \sum_{t=1}^{T} \left\langle \nabla f_{t-1}(\mathbf{x}_{t-1}), \mathbf{x}_t - \mathbf{x}_t^{k,i} - \mathbf{x}_{t-1} + \mathbf{x}_{t-1}^{k,i} \right\rangle^2$$

$$\leq 2D^2 \bar{V}_T + 4G^2 \sum_{t=1}^{T} \|\mathbf{x}_t - \mathbf{x}_{t-1}\|_2^2 + 4G^2 \sum_{t=1}^{T} \|\mathbf{x}_t^{k,i} - \mathbf{x}_{t-1}^{k,i}\|_2^2,$$

where the second and third steps are due to the fact that $(a+b)^2 \leq 2a^2 + 2b^2$ for any $a, b \in \mathbb{R}$. Taking the expectation on the both sides, we obtain

$$\mathbb{E}\left[ \sum_{t=1}^{T} \left( l_t^{k,i} - m_t^{k,i} \right)^2 \right] \leq 2D^2 G^2 + 8D^2 (2\sigma_{1:T}^2 + \Sigma_{1:T}^2) + (8D^2 H^2 + 4G^2) \sum_{t=2}^{T} \|\mathbf{x}_t - \mathbf{x}_{t-1}\|_2^2$$

$$+ 4G^2 \sum_{t=1}^{T} \|\mathbf{x}_t^{k,i} - \mathbf{x}_{t-1}^{k,i}\|_2^2,$$

where the inequality is due to Lemma 3. Therefore, combining the above results with (47), Lemma 10 and Lemma 3 delivers

$$\text{meta-regret} = \mathbb{E}\left[ \sum_{t=1}^{T} \langle \nabla f_t(\mathbf{x}_t), \mathbf{x}_t - \mathbf{x}_t^{k,i} \rangle + r(\mathbf{x}_t) - r(\mathbf{x}_t^{k,i}) \right]$$

$$\leq \frac{1}{\eta^k} \ln \frac{N}{3(C_0 \eta^k)^2} + 256\eta^k D^2 (2\sigma_{1:T}^2 + \Sigma_{1:T}^2) + \frac{64}{C_0} \left( 2D^2 H^2 + G^2 - \frac{C_0}{64}\gamma_1 \right) \sum_{t=2}^{T} \|\mathbf{x}_t - \mathbf{x}_{t-1}\|_2^2$$

$$+ \left( \frac{64G^2}{C_0} - \gamma_2 \right) \sum_{t=1}^{T} \|\mathbf{x}_t^{k,i} - \mathbf{x}_{t-1}^{k,i}\|_2^2 - \frac{C_0}{2} \sum_{t=2}^{T} \|q_t - q_{t-1}\|_1^2 - \frac{C_0}{16} \sum_{t=2}^{T} \|p_t^k - p_{t-1}^k\|_1^2 + \frac{32D^2 G^2}{C_0}$$

$$- \gamma_1 \sum_{t=2}^{T} \sum_{k=1}^{K} q_t^k \|\mathbf{x}_t^k - \mathbf{x}_{t-1}^k\|_2^2 - \gamma_2 \sum_{t=2}^{T} \sum_{i=1}^{N} p_t^{k,i} \|\mathbf{x}_t^{k,i} - \mathbf{x}_{t-1}^{k,i}\|_2^2.$$

To bound the first two terms, we exploit the following lemma.

**Lemma 11.** *[Yan et al., 2023, Lemma 7] For the step size pool $\mathcal{H} = \{\eta_1, \cdots, \eta_K\}$ with $\eta_1 = \frac{1}{2C_0} \geq \cdots \geq \eta_T = \frac{1}{2C_0 T}$, if $C_0 \geq \frac{\sqrt{X}}{2T}$, there exists $\eta^k \in \mathcal{H}$ satisfying*

$$\frac{1}{\eta} \ln \frac{Y}{\eta^2} + \eta X \leq 2C_0 \ln\left(4YC_0^2\right) + 4\sqrt{X \ln(4XY)}.$$

Therefore, we have

$$\frac{1}{\eta^k} \ln \frac{N}{3(C_0 \eta^k)^2} + 256 \eta^k D^2 (2\sigma_{1:T}^2 + \Sigma_{1:T}^2)$$

$$\leq 2C_0 \ln(2N) + 64D \sqrt{(2\sigma_{1:T}^2 + \Sigma_{1:T}^2) \ln \left(\frac{2^{10} D^2 N}{C_0^2} (2\sigma_{1:T}^2 + \Sigma_{1:T}^2)\right)}$$

$$\leq 2C_0 \ln(2N) + 64D \sqrt{\ln\left(\frac{2^{10} D^2 N}{C_0^2} (2\sigma_{1:T}^2 + \Sigma_{1:T}^2)\right)} \left(\sqrt{2\sigma_{1:T}^2} + \sqrt{\Sigma_{1:T}^2}\right).$$

Hence, by requiring $C_0 \geq 1$, the meta-regret is bounded as:

meta-regret

$$\leq \tilde{\mathcal{O}}\left(\sqrt{\sigma_{1:T}^2} + \sqrt{\Sigma_{1:T}^2}\right) + \mathcal{O}(1) + 64\left(2D^2 H^2 + G^2 - \frac{1}{64}\gamma_1\right) \sum_{t=2}^{T} \|\mathbf{x}_t - \mathbf{x}_{t-1}\|_2^2$$

$$+ \left(64G^2 - \gamma_2\right) \sum_{t=1}^{T} \|\mathbf{x}_t^{k,i} - \mathbf{x}_{t-1}^{k,i}\|_2^2 - \frac{C_0}{2} \sum_{t=2}^{T} \|\boldsymbol{q}_t - \boldsymbol{q}_{t-1}\|_1^2 - \frac{C_0}{16} \sum_{t=2}^{T} \|\boldsymbol{p}_t^k - \boldsymbol{p}_{t-1}^k\|_1^2$$

$$- \gamma_1 \sum_{t=2}^{T} \sum_{k=1}^{K} q_t^k \|\mathbf{x}_t^k - \mathbf{x}_{t-1}^k\|_2^2 - \gamma_2 \sum_{t=2}^{T} \sum_{i=1}^{N} p_t^{k,i} \|\mathbf{x}_t^{k,i} - \mathbf{x}_{t-1}^{k,i}\|_2^2.$$

Combining the expert-regret and meta-regret, we get

$$\mathbb{E}\left[\text{Regret}_T\right]$$

$$\leq \tilde{\mathcal{O}}\left(\sqrt{\sigma_{1:T}^2} + \sqrt{\Sigma_{1:T}^2}\right) + \mathcal{O}(1) + \left(128D^2 H^2 + 64G^2 + 10DH - \gamma_1\right) \sum_{t=2}^{T} \|\mathbf{x}_t - \mathbf{x}_{t-1}\|_2^2$$

$$+ \left(64G^2 - \gamma_2 - \frac{\delta}{4D}\right) \sum_{t=1}^{T} \|\mathbf{x}_t^{k,i} - \mathbf{x}_{t-1}^{k,i}\|_2^2 - \frac{C_0}{2} \sum_{t=2}^{T} \|\boldsymbol{q}_t - \boldsymbol{q}_{t-1}\|_1^2 - \frac{C_0}{16} \sum_{t=2}^{T} \|\boldsymbol{p}_t^k - \boldsymbol{p}_{t-1}^k\|_1^2$$

$$\leq \tilde{\mathcal{O}}\left(\sqrt{\sigma_{1:T}^2} + \sqrt{\Sigma_{1:T}^2}\right) + \mathcal{O}(1) + (2C_1 - 2\gamma_1) \sum_{t=2}^{T} q_t^k \|\mathbf{x}_t^k - \mathbf{x}_{t-1}^k\|_2^2$$

$$+ \left(2D^2 C_1 - 2D^2 \gamma_1 - \frac{C_0}{2}\right) \sum_{t=2}^{T} \|\boldsymbol{q}_t - \boldsymbol{q}_{t-1}\|_1^2 + \left(64G^2 - \gamma_2 - \frac{\delta}{4D}\right) \sum_{t=1}^{T} \|\mathbf{x}_t^{k,i} - \mathbf{x}_{t-1}^{k,i}\|_2^2,$$

where $C_1 = 128D^2 H^2 + 64G^2 + 10DH$ and the last step is due to the following lemma.

**Lemma 12.** *Let $\mathbf{x}_t = \sum_{k=1}^{K} q_t^k \mathbf{x}_t^k$. Then, we have*

$$\|\mathbf{x}_t - \mathbf{x}_{t-1}\|_2^2 \leq 2 \sum_{k=1}^{K} q_t^k \|\mathbf{x}_t^k - \mathbf{x}_{t-1}^k\|_2^2 + 2D^2 \|\boldsymbol{q}_t - \boldsymbol{q}_{t-1}\|_1^2. \tag{48}$$

By setting $C_0 \geq 1$, $C_0 \geq 8D^2 C_1$, $\gamma_1 \geq C_1$ and $\delta \geq 256G^2 D + 4D\gamma_2$, we finish the proof.

### B.6.2 For strongly convex functions

According to (28), we could decompose the regret as below:

$$\mathbb{E}\left[\text{Regret}_T\right] \leq \mathbb{E}\left[\sum_{t=1}^{T}\langle\nabla f_t(\mathbf{x}_t),\mathbf{x}_t - \mathbf{x}^*\rangle + r(\mathbf{x}_t) - r(\mathbf{x}^*) - \frac{\lambda}{2}\sum_{t=1}^{T}\|\mathbf{x}^* - \mathbf{x}_t\|_2^2\right]$$

$$\leq\mathbb{E}\left[\sum_{t=1}^{T}\langle\nabla f_t(\mathbf{x}_t),\mathbf{x}_t - \mathbf{x}^*\rangle + r(\mathbf{x}_t) - r(\mathbf{x}^*) - \frac{\lambda^i}{2}\sum_{t=1}^{T}\|\mathbf{x}^* - \mathbf{x}_t\|_2^2\right]$$

$$=\mathbb{E}\underbrace{\left[\sum_{t=1}^{T}\langle\nabla f_t(\mathbf{x}_t),\mathbf{x}_t - \mathbf{x}_t^{k,i}\rangle + r(\mathbf{x}_t) - r(\mathbf{x}_t^{k,i}) - \frac{\lambda^i}{2}\sum_{t=1}^{T}\|\mathbf{x}_t^{k,i} - \mathbf{x}_t\|_2^2\right]}_{\text{meta-regret}}$$

$$+\mathbb{E}\underbrace{\left[\sum_{t=1}^{T}\langle\nabla f_t(\mathbf{x}_t),\mathbf{x}_t^{k,i} - \mathbf{x}^*\rangle + r(\mathbf{x}_t^{k,i}) - r(\mathbf{x}^*) + \frac{\lambda^i}{2}\sum_{t=1}^{T}\|\mathbf{x}_t^{k,i} - \mathbf{x}_t\|_2^2 - \frac{\lambda^i}{2}\sum_{t=1}^{T}\|\mathbf{x}^* - \mathbf{x}_t\|_2^2\right]}_{\text{expert-regret}},$$

where the first step is due to the fact that there exists an $\lambda^i \in \mathcal{H}$ satisfying $\lambda^i \leq \lambda \leq 2\lambda^i$.

**Expert-regret analysis.** We define the surrogate loss functions $h_{t,i}^{sc}(\mathbf{x}) = \langle\nabla f_t(\mathbf{x}_t),\mathbf{x}\rangle + r(\mathbf{x}) + \frac{\lambda^i}{2}\sum_{t=1}^{T}\|\mathbf{x} - \mathbf{x}_t\|_2^2$ for the $i$-th expert. According to Theorem 3, we can bound the expert-regret by

expert-regret

$$\leq\mathcal{O}\left(\frac{1}{\lambda}\left(\sigma_{\max}^2 + \Sigma_{\max}^2\right)\ln\left(\sigma_{1:T}^2 + \Sigma_{1:T}^2\right)\right) + 16D^2\left(H^2 + 1\right)^2\sum_{t=2}^{T}\|\mathbf{x}_t - \mathbf{x}_{t-1}\|^2$$

$$+\left(8D^2\left(H^2 + 1\right) - \frac{\delta}{8}\right)\sum_{t=2}^{T}\left\|\mathbf{x}_t^{k,i} - \mathbf{x}_{t-1}^{k,i}\right\|^2 + \frac{1}{4}\delta D^2 + \frac{20G^2}{\lambda} + \frac{1}{\lambda^2} + \frac{\lambda D^2}{4}.$$

**Meta-regret analysis.** Similar to (46), the instantaneous meta-regret can be bounded by

$$\langle\nabla f_t(\mathbf{x}_t),\mathbf{x}_t - \mathbf{x}_t^{k,i}\rangle + r(\mathbf{x}_t) - r(\mathbf{x}_t^{k,i}) - \frac{\lambda^i}{2}\|\mathbf{x}_t^{k,i} - \mathbf{x}_t\|_2^2$$

$$\leq\langle\boldsymbol{q}_t - \mathbf{e}_k,\boldsymbol{l}_t\rangle + \left\langle\boldsymbol{p}_t^k - \mathbf{e}_i,\boldsymbol{l}_t^k\right\rangle - \gamma_1\sum_{k=1}^{K}q_t^k\|\mathbf{x}_t^k - \mathbf{x}_{t-1}^k\|_2^2 + \gamma_1\|\mathbf{x}_t^k - \mathbf{x}_{t-1}^k\|_2^2 \qquad (49)$$

$$-\gamma_2\sum_{i=1}^{N}p_t^{k,i}\|\mathbf{x}_t^{k,i} - \mathbf{x}_{t-1}^{k,i}\|_2^2 + \gamma_2\|\mathbf{x}_t^{k,i} - \mathbf{x}_{t-1}^{k,i}\|_2^2 - \frac{\lambda^i}{2}\|\mathbf{x}_t^{k,i} - \mathbf{x}_t\|_2^2.$$

For the first two terms, we exploit Lemma 10 and obtain

$$\sum_{t=1}^{T}\langle\boldsymbol{q}_t - \mathbf{e}_k,\boldsymbol{l}_t\rangle + \left\langle\boldsymbol{p}_t^k - \mathbf{e}_i,\boldsymbol{l}_t^k\right\rangle \leq\frac{1}{\eta^k}\ln\frac{N}{3(C_0\eta^k)^2} + 32\eta^k\sum_{t=1}^{T}\left(l_t^{k,i} - m_t^{k,i}\right)^2$$

$$-\frac{C_0}{2}\sum_{t=2}^{T}\|\boldsymbol{q}_t - \boldsymbol{q}_{t-1}\|_1^2 - \frac{C_0}{16}\sum_{t=2}^{T}\|\boldsymbol{p}_t^k - \boldsymbol{p}_{t-1}^k\|_1^2. \qquad (50)$$

Note that

$$\sum_{t=1}^{T}\left(l_t^{k,i} - m_t^{k,i}\right)^2 = \sum_{t=1}^{T}\left(\left\langle\nabla f_t(\mathbf{x}_t),\mathbf{x}_t - \mathbf{x}_t^{k,i}\right\rangle - \left\langle\nabla f_{t-1}(\mathbf{x}_{t-1}),\mathbf{x}_{t-1} - \mathbf{x}_{t-1}^{k,i}\right\rangle\right)^2$$

$$\leq2\sum_{t=1}^{T}\left\langle\nabla f_t(\mathbf{x}_t),\mathbf{x}_t - \mathbf{x}_t^{k,i}\right\rangle^2 + 2\sum_{t=1}^{T}\left\langle\nabla f_{t-1}(\mathbf{x}_{t-1}),\mathbf{x}_{t-1} - \mathbf{x}_{t-1}^{k,i}\right\rangle^2 \qquad (51)$$

$$\leq4\sum_{t=1}^{T}\left\langle\nabla f_t(\mathbf{x}_t),\mathbf{x}_t - \mathbf{x}_t^{k,i}\right\rangle^2 + 2G^2D^2 \leq 4G^2\sum_{t=1}^{T}\left\|\mathbf{x}_t - \mathbf{x}_t^{k,i}\right\|^2 + 2G^2D^2,$$

where the first step is due to the fact that $(a+b)^2 \leq 2a^2 + 2b^2$ for any $a, b \in \mathbb{R}$. Substituting (50) and (51) into (49) obtains

meta-regret

$$
\begin{aligned}
\leq & \frac{1}{\eta^k} \ln \frac{N}{3(C_0\eta^k)^2} + \left(128\eta^k G^2 - \frac{\lambda^i}{2}\right) \sum_{t=1}^{T} \left\| \mathbf{x}_t - \mathbf{x}_t^{k,i} \right\|^2 - \frac{C_0}{2} \sum_{t=2}^{T} \|\boldsymbol{q}_t - \boldsymbol{q}_{t-1}\|_1^2 \\
& - \frac{C_0}{16} \sum_{t=2}^{T} \|\boldsymbol{p}_t^k - \boldsymbol{p}_{t-1}^k\|_1^2 - \gamma_1 \sum_{k=1}^{K} q_t^k \|\mathbf{x}_t^k - \mathbf{x}_{t-1}^k\|_2^2 + \gamma_1 \|\mathbf{x}_t^k - \mathbf{x}_{t-1}^k\|_2^2 \\
& - \gamma_2 \sum_{i=1}^{N} p_t^{k,i} \|\mathbf{x}_t^{k,i} - \mathbf{x}_{t-1}^{k,i}\|_2^2 + \gamma_2 \|\mathbf{x}_t^{k,i} - \mathbf{x}_{t-1}^{k,i}\|_2^2 + 64\eta^k G^2 D^2 \\
\leq & \frac{1}{\eta^k} \ln \frac{N}{(\eta^k)^2} + \left(128\eta^k G^2 - \frac{\lambda^i}{2}\right) \sum_{t=1}^{T} \left\| \mathbf{x}_t - \mathbf{x}_t^{k,i} \right\|^2 - \frac{C_0}{2} \sum_{t=2}^{T} \|\boldsymbol{q}_t - \boldsymbol{q}_{t-1}\|_1^2 \\
& + \left(2D^2\gamma_1 - \frac{C_0}{16}\right) \sum_{t=2}^{T} \|\boldsymbol{p}_t^k - \boldsymbol{p}_{t-1}^k\|_1^2 - \gamma_1 \sum_{k=1}^{K} q_t^k \|\mathbf{x}_t^k - \mathbf{x}_{t-1}^k\|_2^2 \\
& + (2\gamma_1 - \gamma_2) \sum_{i=1}^{N} p_t^{k,i} \|\mathbf{x}_t^{k,i} - \mathbf{x}_{t-1}^{k,i}\|_2^2 + \gamma_2 \|\mathbf{x}_t^{k,i} - \mathbf{x}_{t-1}^{k,i}\|_2^2 + 64\eta^k G^2 D^2,
\end{aligned}
$$

where the last step is due to Lemma 12 and $C_0 \geq 1$. To bound the first term in the above result, we employ the following lemma.

**Lemma 13.** *[Yan et al., 2023, Lemma 8] For the step size pool $\mathcal{H} = \{\eta_1, \cdots, \eta_K\}$ with $\eta_1 = \frac{1}{2C_0} \geq \cdots \geq \eta_T = \frac{1}{2C_0 T}$, if $C_0 \geq \frac{1}{2\eta^* T}$ where $\eta^*$ is the optimal step size, there exists $\eta^k \in \mathcal{H}$ satisfying*

$$
\frac{1}{\eta} \ln \frac{Y}{\eta^2} \leq 2C_0 \ln \left(4YC_0^2\right) + \frac{1}{\eta^*} \ln \frac{4Y}{(\eta^*)^2}.
$$

Applying the above lemma with $\eta^* = \frac{\lambda^i}{256G^2}$ delivers

$$
\begin{aligned}
\text{meta-regret} \leq & 2C_0 \ln\left(4NC_0^2\right) + \frac{512G^2}{\lambda} \ln \frac{2048NG^2}{\lambda^2} - \frac{C_0}{2} \sum_{t=2}^{T} \|\boldsymbol{q}_t - \boldsymbol{q}_{t-1}\|_1^2 \\
& + \left(2D^2\gamma_1 - \frac{C_0}{16}\right) \sum_{t=2}^{T} \|\boldsymbol{p}_t^k - \boldsymbol{p}_{t-1}^k\|_1^2 - \gamma_1 \sum_{k=1}^{K} q_t^k \|\mathbf{x}_t^k - \mathbf{x}_{t-1}^k\|_2^2 \\
& + (2\gamma_1 - \gamma_2) \sum_{i=1}^{N} p_t^{k,i} \|\mathbf{x}_t^{k,i} - \mathbf{x}_{t-1}^{k,i}\|_2^2 + \gamma_2 \|\mathbf{x}_t^{k,i} - \mathbf{x}_{t-1}^{k,i}\|_2^2 + \frac{\lambda D^2}{4}.
\end{aligned}
$$

Then, we combine the expert-regret and meta-regret:

$$
\begin{aligned}
\mathbb{E}\left[\text{Regret}_T\right] \leq & \mathcal{O}\left(\frac{1}{\lambda}\left(\sigma_{\max}^2 + \Sigma_{\max}^2\right) \ln\left(\sigma_{1:T}^2 + \Sigma_{1:T}^2\right)\right) \\
& + \left(8D^2\left(H^2 + 1\right) - \frac{\delta}{8} + \gamma_2\right) \sum_{t=2}^{T} \left\|\mathbf{x}_t^{k,i} - \mathbf{x}_{t-1}^{k,i}\right\|^2 + \left(2D^2 C_2 - \frac{C_0}{2}\right) \sum_{t=2}^{T} \|\boldsymbol{q}_t - \boldsymbol{q}_{t-1}\|_1^2 \\
& + \left(2D^2\gamma_1 - \frac{C_0}{16}\right) \sum_{t=2}^{T} \|\boldsymbol{p}_t^k - \boldsymbol{p}_{t-1}^k\|_1^2 + (2C_2 - \gamma_1) \sum_{k=1}^{K} q_t^k \|\mathbf{x}_t^k - \mathbf{x}_{t-1}^k\|_2^2 \\
& + (2\gamma_1 - \gamma_2) \sum_{i=1}^{N} p_t^{k,i} \|\mathbf{x}_t^{k,i} - \mathbf{x}_{t-1}^{k,i}\|_2^2 + \mathcal{O}(1).
\end{aligned}
$$

where $C_2 = 16D^2\left(H^2 + 1\right)^2$. Setting $C_0 \geq 4D^2 C_2$, $C_0 \geq 32D^2\gamma_1$, $\delta \geq 64D^2(H^2 + 1) + 8\gamma_2$, $\gamma_1 \geq 2C_2$ and $\gamma_2 \geq \gamma_1$ completes the proof.

### B.6.3 For exp-concave functions

According to (38), we could decompose the regret as below:

$$\mathbb{E}\left[\text{Regret}_T\right] \leq \mathbb{E}\left[\sum_{t=1}^{T}\langle\nabla f_t(\mathbf{x}_t),\mathbf{x}_t-\mathbf{x}^*\rangle + r(\mathbf{x}_t) - r(\mathbf{x}^*) - \frac{\beta}{2}\sum_{t=1}^{T}\langle\nabla f_t(\mathbf{x}_t),\mathbf{x}^*-\mathbf{x}_t\rangle^2\right]$$

$$\leq\mathbb{E}\left[\sum_{t=1}^{T}\langle\nabla f_t(\mathbf{x}_t),\mathbf{x}_t-\mathbf{x}^*\rangle + r(\mathbf{x}_t) - r(\mathbf{x}^*) - \frac{\beta^i}{2}\sum_{t=1}^{T}\langle\nabla f_t(\mathbf{x}_t),\mathbf{x}^*-\mathbf{x}_t\rangle^2\right]$$

$$=\mathbb{E}\underbrace{\left[\sum_{t=1}^{T}\langle\mathbf{g}_t,\mathbf{x}_t-\mathbf{x}_t^{k,i}\rangle + r(\mathbf{x}_t) - r(\mathbf{x}_t^{k,i}) - \frac{\beta^i}{2}\sum_{t=1}^{T}\left\langle\mathbf{g}_t,\mathbf{x}_t^{k,i}-\mathbf{x}_t\right\rangle^2\right]}_{\text{meta-regret}}$$

$$+\mathbb{E}\underbrace{\left[\sum_{t=1}^{T}\langle\mathbf{g}_t,\mathbf{x}_t^{k,i}-\mathbf{x}^*\rangle + r(\mathbf{x}_t^{k,i}) - r(\mathbf{x}^*) + \frac{\beta^i}{2}\sum_{t=1}^{T}\left\langle\mathbf{g}_t,\mathbf{x}_t^{k,i}-\mathbf{x}_t\right\rangle^2 - \frac{\beta^i}{2}\sum_{t=1}^{T}\langle\mathbf{g}_t,\mathbf{x}^*-\mathbf{x}_t\rangle^2\right]}_{\text{expert-regret}},$$

where $\mathbf{g}_t = \nabla f_t(\mathbf{x}_t)$, $\beta^i = \frac{1}{2}\min\{\frac{1}{4GD},\alpha^i\}$, and the first step is due to the fact that there exists an $\alpha^i \in \mathcal{H}$ satisfying $\alpha^i \leq \alpha \leq 2\alpha^i$.

**Expert-regret analysis.** We define the surrogate loss functions $h_{t,i}^{exp}(\mathbf{x}) = \langle\mathbf{g}_t,\mathbf{x}\rangle + r(\mathbf{x}) + \frac{\beta^i}{2}\langle\mathbf{g}_t,\mathbf{x}-\mathbf{x}_t\rangle^2$ for the $i$-th expert. According to Theorem 5, we can bound the expert-regret by

expert-regret

$$\leq\mathcal{O}\left(\frac{d}{\alpha}\ln\left(\sigma_{1:T}^2+\Sigma_{1:T}^2\right)\right) + \frac{24d}{\beta}\ln\left(1+\frac{C_3}{\delta}\sum_{t=2}^{T}\|\mathbf{x}_t-\mathbf{x}_{t-1}\|_2^2\right)$$

$$+\frac{24d}{\beta}\ln\left(1+\frac{\beta G^4}{\delta}\sum_{t=2}^{T}\|\mathbf{x}_t^{k,i}-\mathbf{x}_{t-1}^{k,i}\|_2^2\right) - \frac{\delta}{4}\sum_{t=2}^{T}\|\mathbf{x}_t^{k,i}-\mathbf{x}_{t-1}^{k,i}\|_2^2 + \left(\frac{\delta}{2}+\frac{\beta}{4}G^2\right)D^2$$

$$\leq\mathcal{O}\left(\frac{d}{\alpha}\ln\left(\sigma_{1:T}^2+\Sigma_{1:T}^2\right)\right) + \frac{24dC_3}{\delta}\sum_{t=2}^{T}\|\mathbf{x}_t-\mathbf{x}_{t-1}\|_2^2 + \left(\frac{24dG^4}{\delta}-\frac{\delta}{4}\right)\sum_{t=2}^{T}\|\mathbf{x}_t^{k,i}-\mathbf{x}_{t-1}^{k,i}\|_2^2$$

$$+\left(\frac{\delta}{2}+\frac{\beta}{4}G^2\right)D^2,$$

where $C_3 = 8H^2 + 64D^2G^2H^2 + 8G^4$.

**Meta-regret analysis.** Similar to (46), the instantaneous meta-regret can be bounded by

$$\langle\nabla f_t(\mathbf{x}_t),\mathbf{x}_t-\mathbf{x}_t^{k,i}\rangle + r(\mathbf{x}_t) - r(\mathbf{x}_t^{k,i}) - \frac{\beta^i}{2}\left\langle\nabla f_t(\mathbf{x}_t),\mathbf{x}_t^{k,i}-\mathbf{x}_t\right\rangle^2$$

$$\leq\langle\boldsymbol{q}_t-\mathbf{e}_k,\boldsymbol{l}_t\rangle + \left\langle\boldsymbol{p}_t^k-\mathbf{e}_i,\boldsymbol{l}_t^k\right\rangle - \gamma_1\sum_{k=1}^{K}q_t^k\|\mathbf{x}_t^k-\mathbf{x}_{t-1}^k\|_2^2 + \gamma_1\|\mathbf{x}_t^k-\mathbf{x}_{t-1}^k\|_2^2$$

$$-\gamma_2\sum_{i=1}^{N}p_t^{k,i}\|\mathbf{x}_t^{k,i}-\mathbf{x}_{t-1}^{k,i}\|_2^2 + \gamma_2\|\mathbf{x}_t^{k,i}-\mathbf{x}_{t-1}^{k,i}\|_2^2 - \frac{\beta^i}{2}\left\langle\nabla f_t(\mathbf{x}_t),\mathbf{x}_t^{k,i}-\mathbf{x}_t\right\rangle^2.$$

For the first two terms, we exploit Lemma 10 and obtain

$$\sum_{t=1}^{T}\langle\boldsymbol{q}_t-\mathbf{e}_k,\boldsymbol{l}_t\rangle + \left\langle\boldsymbol{p}_t^k-\mathbf{e}_i,\boldsymbol{l}_t^k\right\rangle \leq\frac{1}{\eta^k}\ln\frac{N}{3(C_0\eta^k)^2} + 32\eta^k\sum_{t=1}^{T}\left(l_t^{k,i}-m_t^{k,i}\right)^2$$

$$-\frac{C_0}{2}\sum_{t=2}^{T}\|\boldsymbol{q}_t-\boldsymbol{q}_{t-1}\|_1^2 - \frac{C_0}{16}\sum_{t=2}^{T}\|\boldsymbol{p}_t^k-\boldsymbol{p}_{t-1}^k\|_1^2.$$

Note that

$$\sum_{t=1}^{T} \left( l_t^{k,i} - m_t^{k,i} \right)^2 = \sum_{t=1}^{T} \left( \left\langle \nabla f_t(\mathbf{x}_t), \mathbf{x}_t - \mathbf{x}_t^{k,i} \right\rangle - \left\langle \nabla f_{t-1}(\mathbf{x}_{t-1}), \mathbf{x}_{t-1} - \mathbf{x}_{t-1}^{k,i} \right\rangle \right)^2$$

$$\leq 2 \sum_{t=1}^{T} \left\langle \nabla f_t(\mathbf{x}_t), \mathbf{x}_t - \mathbf{x}_t^{k,i} \right\rangle^2 + 2 \sum_{t=1}^{T} \left\langle \nabla f_{t-1}(\mathbf{x}_{t-1}), \mathbf{x}_{t-1} - \mathbf{x}_{t-1}^{k,i} \right\rangle^2$$

$$\leq 4 \sum_{t=1}^{T} \left\langle \nabla f_t(\mathbf{x}_t), \mathbf{x}_t - \mathbf{x}_t^{k,i} \right\rangle^2 + 2G^2 D^2.$$

Therefore, the meta-regret is bounded by

meta-regret

$$\leq \frac{1}{\eta^k} \ln \frac{N}{3(C_0 \eta^k)^2} + \left( 128\eta^k - \frac{\beta^i}{2} \right) \sum_{t=1}^{T} \left\langle \nabla f_t(\mathbf{x}_t), \mathbf{x}_t - \mathbf{x}_t^{k,i} \right\rangle^2 + 256 G^2 D^2 \eta^k$$

$$- \frac{C_0}{2} \sum_{t=2}^{T} \|\boldsymbol{q}_t - \boldsymbol{q}_{t-1}\|_1^2 - \frac{C_0}{16} \sum_{t=2}^{T} \|\boldsymbol{p}_t^k - \boldsymbol{p}_{t-1}^k\|_1^2 - \gamma_1 \sum_{k=1}^{K} q_t^k \|\mathbf{x}_t^k - \mathbf{x}_{t-1}^k\|_2^2 + \gamma_1 \|\mathbf{x}_t^k - \mathbf{x}_{t-1}^k\|_2^2$$

$$- \gamma_2 \sum_{i=1}^{N} p_t^{k,i} \|\mathbf{x}_t^{k,i} - \mathbf{x}_{t-1}^{k,i}\|_2^2 + \gamma_2 \|\mathbf{x}_t^{k,i} - \mathbf{x}_{t-1}^{k,i}\|_2^2.$$

Applying Lemma 13 with $\eta^* = \frac{\beta^i}{256}$ delivers

$$\text{meta-regret} \leq 2C_0 \ln \left( 4NC_0^2 \right) + \frac{512}{\beta} \ln \frac{2048N}{\beta^2} + G^2 D^2 \beta - \frac{C_0}{2} \sum_{t=2}^{T} \|\boldsymbol{q}_t - \boldsymbol{q}_{t-1}\|_1^2$$

$$+ \left( 2D^2 \gamma_1 - \frac{C_0}{16} \right) \sum_{t=2}^{T} \|\boldsymbol{p}_t^k - \boldsymbol{p}_{t-1}^k\|_1^2 - \gamma_1 \sum_{k=1}^{K} q_t^k \|\mathbf{x}_t^k - \mathbf{x}_{t-1}^k\|_2^2$$

$$+ (2\gamma_1 - \gamma_2) \sum_{i=1}^{N} p_t^{k,i} \|\mathbf{x}_t^{k,i} - \mathbf{x}_{t-1}^{k,i}\|_2^2 + \gamma_2 \|\mathbf{x}_t^{k,i} - \mathbf{x}_{t-1}^{k,i}\|_2^2.$$

Then, we combine the expert-regret and meta-regret:

$$\mathbb{E}\left[ \text{Regret}_T \right] \leq \mathcal{O} \left( \frac{d}{\alpha} \ln \left( \sigma_{1:T}^2 + \Sigma_{1:T}^2 \right) \right) + \left( \frac{48dDC_3}{\delta} - \frac{C_0}{2} \right) \sum_{t=2}^{T} \|\boldsymbol{q}_t - \boldsymbol{q}_{t-1}\|_1^2$$

$$+ \left( 2D^2 \gamma_1 - \frac{C_0}{16} \right) \sum_{t=2}^{T} \|\boldsymbol{p}_t^k - \boldsymbol{p}_{t-1}^k\|_1^2 + \left( \frac{48dC_3}{\delta} - \gamma_1 \right) \sum_{k=1}^{K} q_t^k \|\mathbf{x}_t^k - \mathbf{x}_{t-1}^k\|_2^2$$

$$+ (2\gamma_1 - \gamma_2) \sum_{i=1}^{N} p_t^{k,i} \|\mathbf{x}_t^{k,i} - \mathbf{x}_{t-1}^{k,i}\|_2^2 + \left( \frac{24dG^4}{\delta} - \frac{\delta}{4} + \gamma_2 \right) \|\mathbf{x}_t^{k,i} - \mathbf{x}_{t-1}^{k,i}\|_2^2 + \mathcal{O}(1).$$

where $C_3 = 8H^2 + 64D^2 G^2 H^2 + 8G^4$. Setting $C_0 \geq 96dDC_3/\delta$, $C_0 \geq 32D^2 \gamma_1$, $\delta \geq 4\sqrt{6d(G^4 + 4C_3)}$, $\gamma_1 \geq 48dC_3/\delta$ and $\gamma_2 \geq 2\gamma_1$ completes the proof.

## C    Supporting lemmas

### C.1    Proof of Lemma 1

First, we introduce two lemmas that will be used in the following proof.

**Lemma 14.** *[Scroccaro et al., 2023, Lemma 3.1] Let $\mathcal{X}$ be a convex set, $\varphi(\cdot) : \mathcal{X} \to \mathbb{R}$ be a convex function and $\eta > 0$. Define*

$$\mathbf{u} = \arg \min_{\mathbf{x} \in \mathcal{X}} \left\{ \eta \varphi(\mathbf{x}) + \mathcal{B}^{\mathcal{R}}(\mathbf{x}, \mathbf{v}) \right\},$$

*Then, for any $\mathbf{z} \in \mathcal{X}$, we have*

$$\eta \langle \mathbf{g}(\mathbf{u}), \mathbf{u} - \mathbf{z} \rangle \leq \mathcal{B}^{\mathcal{R}}(\mathbf{z}, \mathbf{v}) - \mathcal{B}^{\mathcal{R}}(\mathbf{z}, \mathbf{u}) - \mathcal{B}^{\mathcal{R}}(\mathbf{u}, \mathbf{v})$$

*where $\mathbf{g}(\mathbf{u}) \in \partial \varphi(\mathbf{u})$.*

**Lemma 15.** *[Scroccaro et al., 2023, Lemma 3.2] Let $\mathcal{X}$ be a convex set, $\mathcal{R} : \mathcal{X} \to \mathbb{R}$ be an $\alpha$-strongly convex function with respect to the norm $\|\cdot\|$. Define*

$$\mathbf{u}_1 = \arg \min_{\mathbf{x}_1 \in \mathcal{X}} \left\{ \langle \mathbf{w}_1, \mathbf{x}_1 \rangle + r(\mathbf{x}_1) + \mathcal{B}^{\mathcal{R}}(\mathbf{x}_1, \mathbf{v}) \right\}$$
$$\mathbf{u}_2 = \arg \min_{\mathbf{x}_2 \in \mathcal{X}} \left\{ \langle \mathbf{w}_2, \mathbf{x}_2 \rangle + r(\mathbf{x}_2) + \mathcal{B}^{\mathcal{R}}(\mathbf{x}_2, \mathbf{v}) \right\}.$$

*Then, we have*

$$\|\mathbf{u}_1 - \mathbf{u}_2\| \leq \alpha^{-1} \|\mathbf{w}_1 - \mathbf{w}_2\|_*.$$

Then, by exploiting the convexity of $r(\cdot)$, we have

$$\langle \nabla f_t(\mathbf{x}_t), \mathbf{x}_t - \mathbf{x} \rangle + r(\mathbf{x}_t) - r(\mathbf{x}) = \langle \nabla f_t(\mathbf{x}_t), \mathbf{x}_t - \mathbf{x} \rangle + r(\mathbf{x}_t) - r(\hat{\mathbf{x}}_{t+1}) + r(\hat{\mathbf{x}}_{t+1}) - r(\mathbf{x})$$
$$\leq \langle \nabla f_t(\mathbf{x}_t), \mathbf{x}_t - \mathbf{x} \rangle + \langle \nabla r(\mathbf{x}_t), \mathbf{x}_t - \hat{\mathbf{x}}_{t+1} \rangle + \langle \nabla r(\hat{\mathbf{x}}_{t+1}), \hat{\mathbf{x}}_{t+1} - \mathbf{x} \rangle$$
$$= \underbrace{\langle \nabla f_t(\mathbf{x}_t) - M_t, \mathbf{x}_t - \hat{\mathbf{x}}_{t+1} \rangle}_{\texttt{term (a)}} + \underbrace{\langle M_t + \nabla r(\mathbf{x}_t), \mathbf{x}_t - \hat{\mathbf{x}}_{t+1} \rangle}_{\texttt{term (b)}} + \underbrace{\langle \nabla f_t(\mathbf{x}_t) + \nabla r(\hat{\mathbf{x}}_{t+1}), \hat{\mathbf{x}}_{t+1} - \mathbf{x} \rangle}_{\texttt{term (c)}}.$$

To upper bound the $\texttt{term (a)}$, we exploit Lemma 15 and obtain

$$\texttt{term (a)} \leq \|\nabla f_t(\mathbf{x}_t) - M_t\|_* \|\mathbf{x}_t - \hat{\mathbf{x}}_{t+1}\| \leq \alpha^{-1} \|\nabla f_t(\mathbf{x}_t) - M_t\|_*^2. \tag{52}$$

Next, we apply Lemma 14 with the update (4) and (5), and obtain

$$\texttt{term (b)} \leq \mathcal{B}^{\mathcal{R}_t}(\hat{\mathbf{x}}_{t+1}, \hat{\mathbf{x}}_t) - \mathcal{B}^{\mathcal{R}_t}(\hat{\mathbf{x}}_{t+1}, \mathbf{x}_t) - \mathcal{B}^{\mathcal{R}_t}(\mathbf{x}_t, \hat{\mathbf{x}}_t) \tag{53}$$
$$\texttt{term (c)} \leq \mathcal{B}^{\mathcal{R}_t}(\mathbf{x}, \hat{\mathbf{x}}_t) - \mathcal{B}^{\mathcal{R}_t}(\mathbf{x}, \hat{\mathbf{x}}_{t+1}) - \mathcal{B}^{\mathcal{R}_t}(\hat{\mathbf{x}}_{t+1}, \hat{\mathbf{x}}_t) \tag{54}$$

Combining (52), (53) and (54) finishes the proof.

### C.2    Proof of Lemma 3

First, we consider the case $t = 1$. Under the Assumption 2, we obtain

$$\|\nabla f_1(\mathbf{x}_1) - \nabla f_0(\mathbf{x}_0)\|_2^2 = \|\nabla f_1(\mathbf{x}_1)\|_2^2 \leq G^2$$

In the case $t \geq 2$, according to the inequality $\|\mathbf{a} + \mathbf{b}\|_2^2 \leq 2\|\mathbf{a}\|_2^2 + 2\|\mathbf{b}\|_2^2$, we have

$$\|\nabla f_t(\mathbf{x}_t) - \nabla f_{t-1}(\mathbf{x}_{t-1})\|_2^2$$
$$\leq 2\|\nabla f_t(\mathbf{x}_t) - \nabla F_t(\mathbf{x}_{t-1})\|_2^2 + 2\|\nabla F_t(\mathbf{x}_{t-1}) - \nabla f_{t-1}(\mathbf{x}_{t-1})\|_2^2$$
$$\leq 4\|\nabla f_t(\mathbf{x}_t) - \nabla F_t(\mathbf{x}_t)\|_2^2 + 4\|\nabla F_t(\mathbf{x}_t) - \nabla F_t(\mathbf{x}_{t-1})\|_2^2$$
$$\quad + 4\|\nabla F_t(\mathbf{x}_{t-1}) - \nabla F_{t-1}(\mathbf{x}_{t-1})\|_2^2 + 4\|\nabla F_{t-1}(\mathbf{x}_{t-1}) - \nabla f_{t-1}(\mathbf{x}_{t-1})\|_2^2.$$

Under the Assumption 4, the above result becomes

$$\|\nabla f_t(\mathbf{x}_t) - \nabla f_{t-1}(\mathbf{x}_{t-1})\|_2^2$$
$$\leq 4\|\nabla f_t(\mathbf{x}_t) - \nabla F_t(\mathbf{x}_t)\|_2^2 + 4H^2\|\mathbf{x}_t - \mathbf{x}_{t-1}\|_2^2 \tag{55}$$
$$\quad + 4\|\nabla F_t(\mathbf{x}_{t-1}) - \nabla F_{t-1}(\mathbf{x}_{t-1})\|_2^2 + 4\|\nabla F_{t-1}(\mathbf{x}_{t-1}) - \nabla f_{t-1}(\mathbf{x}_{t-1})\|_2^2.$$

Combining both cases, we reach at

$$\|\nabla f_t(\mathbf{x}_t) - \nabla f_{t-1}(\mathbf{x}_{t-1})\|_2^2 \le G^2 + 4\|\nabla f_t(\mathbf{x}_t) - \nabla F_t(\mathbf{x}_t)\|_2^2 + 4\|\nabla F_{t-1}(\mathbf{x}_{t-1}) - \nabla F_{t-1}(\mathbf{x}_{t-1})\|_2^2$$
$$+ 4H^2\|\mathbf{x}_t - \mathbf{x}_{t-1}\|_2^2 + 4\|\nabla F_{t-1}(\mathbf{x}_{t-1}) - \nabla f_{t-1}(\mathbf{x}_{t-1})\|_2^2.$$

Summing up both sides of the above inequality over $t = 1, \cdots, T$ obtains

$$\bar{V}_T = G^2 + 4\sum_{t=2}^{T}\|\nabla f_t(\mathbf{x}_t) - \nabla F_t(\mathbf{x}_t)\|_2^2 + 4\sum_{t=2}^{T}\|\nabla F_t(\mathbf{x}_{t-1}) - \nabla F_{t-1}(\mathbf{x}_{t-1})\|_2^2$$

$$+ 4H^2\sum_{t=2}^{T}\|\mathbf{x}_t - \mathbf{x}_{t-1}\|_2^2 + 4\sum_{t=2}^{T}\|\nabla F_{t-1}(\mathbf{x}_{t-1}) - \nabla f_{t-1}(\mathbf{x}_{t-1})\|_2^2$$

$$= G^2 + 8\sum_{t=1}^{T}\|\nabla f_t(\mathbf{x}_t) - \nabla F_t(\mathbf{x}_t)\|_2^2 + 4\sum_{t=2}^{T}\|\nabla F_t(\mathbf{x}_{t-1}) - \nabla F_{t-1}(\mathbf{x}_{t-1})\|_2^2$$

$$+ 4H^2\sum_{t=2}^{T}\|\mathbf{x}_t - \mathbf{x}_{t-1}\|_2^2,$$

which completes the proof.

## C.3 Proof of Lemma 4

Similar to the analysis in Lemma 3, we first consider the case $t = 1$.

$$\|\nabla f_1(\hat{\mathbf{x}}_1) - \nabla f_0(\hat{\mathbf{x}}_0)\|_2^2 = \|\nabla f_1(\hat{\mathbf{x}}_1)\|_2^2 \le G^2.$$

Then, for the case $t \ge 2$, we have

$$\|\nabla f_t(\hat{\mathbf{x}}_t) - \nabla f_{t-1}(\hat{\mathbf{x}}_t)\|_2^2 \le \sup_{\mathbf{x}\in\mathcal{X}}\|\nabla f_t(\mathbf{x}) - \nabla f_{t-1}(\mathbf{x})\|_2^2$$

$$\le \sup_{\mathbf{x}\in\mathcal{X}} 2\|\nabla f_t(\mathbf{x}) - \nabla F_t(\mathbf{x})\|_2^2 + \sup_{\mathbf{x}\in\mathcal{X}} 2\|\nabla F_t(\mathbf{x}) - \nabla f_{t-1}(\mathbf{x})\|_2^2$$

$$\le \sup_{\mathbf{x}\in\mathcal{X}} 2\|\nabla f_t(\mathbf{x}) - \nabla F_t(\mathbf{x})\|_2^2 + \sup_{\mathbf{x}\in\mathcal{X}} 2\|\nabla F_t(\mathbf{x}) - \nabla F_{t-1}(\mathbf{x})\|_2^2 + \sup_{\mathbf{x}\in\mathcal{X}} 2\|\nabla F_{t-1}(\mathbf{x}) - \nabla f_{t-1}(\mathbf{x})\|_2^2.$$

Summing up both sides of the above inequality over $t = 1, \cdots, T$ completes the proof.

## C.4 Proof of Lemma 7

The analysis is based on Lemma 19 in Chiang et al. [2012]. Recall the definition $H_t = I + \frac{\beta}{2}G^2\delta I + \frac{\beta}{2}\sum_{r=1}^{t-1}h_r$ with $h_r = \nabla f_r(\mathbf{x}_r)\nabla f_r(\mathbf{x}_r)^\top$. Then, we can proof that

$$H_t \succeq \delta I + \frac{\beta}{2}\sum_{r=1}^{t}h_r \succeq \delta I + \frac{\beta}{4}\sum_{r=1}^{t}(h_r + h_{r-1}). \tag{56}$$

The first step is due to Assumption 2, and the last step is due to $\nabla f_0(\mathbf{x}_0)$ the all-0 vector. Next, according to the fact that

$$\frac{1}{2}h_r + \frac{1}{2}h_{r-1} + \frac{1}{2}\nabla f_{r-1}(\mathbf{x}_{r-1})\nabla f_r(\mathbf{x}_r)^\top + \frac{1}{2}\nabla f_r(\mathbf{x}_r)\nabla f_{r-1}(\mathbf{x}_{r-1})^\top$$

$$= \frac{1}{2}\left(\nabla f_r(\mathbf{x}_r) + \nabla f_{r-1}(\mathbf{x}_{r-1})\right)\left(\nabla f_r(\mathbf{x}_r) + \nabla f_{r-1}(\mathbf{x}_{r-1})\right)^\top \succeq 0,$$

we have

$$h_r + h_{r-1} \succeq \frac{1}{2}\left(\nabla f_r(\mathbf{x}_r) - \nabla f_{r-1}(\mathbf{x}_{r-1})\right)\left(\nabla f_r(\mathbf{x}_r) - \nabla f_{r-1}(\mathbf{x}_{r-1})\right)^\top. \tag{57}$$

Substituting (57) into (56) delivers

$$H_t \succeq \delta I + \frac{\beta}{8}\sum_{r=1}^{t}\left(\nabla f_r(\mathbf{x}_r) - \nabla f_{r-1}(\mathbf{x}_{r-1})\right)\left(\nabla f_r(\mathbf{x}_r) - \nabla f_{r-1}(\mathbf{x}_{r-1})\right)^\top. \tag{58}$$

For brevity, we denote $K_t = \delta I + \frac{\beta}{8} \sum_{r=1}^{t} (\nabla f_r(\mathbf{x}_r) - \nabla f_{r-1}(\mathbf{x}_{r-1})) (\nabla f_r(\mathbf{x}_r) - \nabla f_{r-1}(\mathbf{x}_{r-1}))^\top$. From (58), we obtain that $H_t^{-1} \preceq K_t^{-1}$, which implies

$$\sum_{t=1}^{T} \|\nabla f_t(\mathbf{x}_t) - \nabla f_{t-1}(\mathbf{x}_{t-1})\|_{H_t^{-1}}^2 \leq \sum_{t=1}^{T} \|\nabla f_t(\mathbf{x}_t) - \nabla f_{t-1}(\mathbf{x}_{t-1})\|_{K_t^{-1}}^2$$
$$= \frac{8}{\beta} \sum_{t=1}^{T} \left\| \sqrt{\frac{\beta}{8}} (\nabla f_t(\mathbf{x}_t) - \nabla f_{t-1}(\mathbf{x}_{t-1})) \right\|_{K_t^{-1}}^2 \tag{59}$$

Then, we introduce the following lemma:

**Lemma 16.** *[Hazan et al., 2007, Lemma 11] Let $\mathbf{u}_1, \cdots, \mathbf{u}_T \in \mathbb{R}^d$ be any sequence vectors and $\epsilon > 0$ be a positive real number. Denote $V_t = \epsilon I + \sum_{i=1}^{t} \mathbf{u}_i \mathbf{u}_i^\top$, then we have*

$$\sum_{t=1}^{T} \mathbf{u}_t^\top V_t^{-1} \mathbf{u}_t \leq d \log \left( 1 + \frac{1}{\epsilon} \sum_{t=1}^{T} \|\mathbf{u}_t\|_2^2 \right).$$

By setting $\epsilon = \delta$ and $\mathbf{u}_t = \sqrt{\frac{\beta}{8}} (\nabla f_t(\mathbf{x}_t) - \nabla f_{t-1}(\mathbf{x}_{t-1}))$, we have

$$\sum_{t=1}^{T} \left\| \sqrt{\frac{\beta}{8}} (\nabla f_t(\mathbf{x}_t) - \nabla f_{t-1}(\mathbf{x}_{t-1})) \right\|_{K_t^{-1}}^2 \leq d \log \left( 1 + \frac{\beta}{8\delta} \sum_{t=1}^{T} \|\nabla f_t(\mathbf{x}_t) - \nabla f_{t-1}(\mathbf{x}_{t-1})\|_2^2 \right). \tag{60}$$

We complete the proof by combining (59) and (60).

### C.5 Proof of Lemma 10

Our proof starts from the following lemma for single-layer MsMwC.

**Lemma 17.** *[Yan et al., 2023, Lemma 2] If $\max_{t \in [T], i \in [N]} |l_t^i|, |m_t^i| \leq 1$, then MsMwC enjoys*

$$\sum_{t=1}^{T} \langle \boldsymbol{l}_t, \boldsymbol{p}_t \rangle - \sum_{t=1}^{T} l_t^i \leq \frac{1}{\eta^i} \ln \frac{1}{\hat{p}_1^i} + \sum_{i=1}^{N} \frac{\hat{p}_1^i}{\eta^i} - 8 \sum_{t=1}^{T} \sum_{i=1}^{N} \eta^i p_t^i \left( l_t^i - m_t^i \right)^2$$
$$+ 16 \eta^i \sum_{t=1}^{T} \left( l_t^i - m_t^i \right)^2 - \min_{k \in [K]} \frac{1}{4\eta^k} \sum_{t=2}^{T} \|\boldsymbol{p}_t - \boldsymbol{p}_{t-1}\|_1^2.$$

It can be verified that under Assumption 1, 2 and 3, all our composite losses and optimisms defined in (13) and (14) are bounded

$$|l_t^k| \leq GD + C + \gamma_1 D, \; |m_t^k| \leq GD + C + \gamma_1 D$$
$$|l_t^{k,i}| \leq GD + C + \gamma_2 D, \; |m_t^{k,i}| \leq GD + C + \gamma_2 D,$$

and we can rescale them to $[-1, 1]$ with only constant multiplicative factors in the constant hyperparameter $\gamma_1$ and $\gamma_2$. Therefore, applying Lemma 17 with the first layer(i.e., with $l_t^k$ as the surrogate losses), we have

$$\sum_{t=1}^{T} \langle \boldsymbol{l}_t, \boldsymbol{q}_t \rangle - \sum_{t=1}^{T} l_t^k \leq \frac{1}{\eta^k} \ln \frac{1}{\hat{q}_1^k} + \sum_{k=1}^{K} \frac{\hat{q}_1^k}{\eta^k} - 8 \sum_{t=1}^{T} \sum_{k=1}^{K} \eta^k q_t^k \left( l_t^k - m_t^k \right)^2$$
$$+ 16 \eta^k \sum_{t=1}^{T} \left( l_t^k - m_t^k \right)^2 - \min_{k \in [K]} \frac{1}{4\eta^k} \sum_{t=2}^{T} \|\boldsymbol{q}_t - \boldsymbol{q}_{t-1}\|_1^2. \tag{61}$$

For the first two terms, the initialization $\hat{q}_1^k = (\eta^k)^2 / \sum_{k=1}^{K} (\eta^k)^2$ and $\eta^k = 1/(C_0 2^k)$ imply

$$\frac{1}{\eta^k} \ln \frac{1}{\hat{q}_1^k} + \sum_{k=1}^{K} \frac{\hat{q}_1^k}{\eta^k} = \frac{1}{\eta^k} \ln \frac{\sum_{k=1}^{K} (\eta^k)^2}{(\eta^k)^2} + \sum_{k=1}^{K} \frac{\eta^k}{\sum_{k=1}^{K} (\eta^k)^2} \leq \frac{1}{\eta^k} \ln \frac{1}{3(C_0 \eta^k)^2} + 4C_0. \tag{62}$$

For the last term, due to $\eta^k = 1/(C_0 2^k) \leq 2/C_0$, we have

$$\min_{k \in [K]} \frac{1}{4\eta^k} \sum_{t=2}^{T} \|\boldsymbol{q}_t - \boldsymbol{q}_{t-1}\|_1^2 \geq \frac{C_0}{2} \sum_{t=2}^{T} \|\boldsymbol{q}_t - \boldsymbol{q}_{t-1}\|_1^2. \tag{63}$$

Substitute (62) and (63) into (61) delivers

$$\sum_{t=1}^{T} \langle \boldsymbol{l}_t, \boldsymbol{q}_t \rangle - \sum_{t=1}^{T} l_t^k \leq \frac{1}{\eta^k} \ln \frac{1}{3(C_0 \eta^k)^2} + 16\eta^k \sum_{t=1}^{T} \left( l_t^k - m_t^k \right)^2 - \frac{C_0}{2} \sum_{t=2}^{T} \|\boldsymbol{q}_t - \boldsymbol{q}_{t-1}\|_1^2. \tag{64}$$

Then, we apply Lemma 17 again with $l_t^{k,i}$ and $m_t^{k,i}$.

$$\sum_{t=1}^{T} \left\langle \boldsymbol{l}_t^k, \boldsymbol{p}_t^k \right\rangle - \sum_{t=1}^{T} l_t^{k,i} \leq \frac{1}{\eta^{k,i}} \ln \frac{1}{\hat{p}_1^{k,i}} + \sum_{i=1}^{N} \frac{\hat{p}_1^{k,i}}{\eta^{k,i}} - 8 \sum_{t=1}^{T} \sum_{i=1}^{N} \eta^{k,i} p_t^{k,i} \left( l_t^{k,i} - m_t^{k,i} \right)^2$$
$$+ 16\eta^{k,i} \sum_{t=1}^{T} \left( l_t^{k,i} - m_t^{k,i} \right)^2 - \min_{i \in [N]} \frac{1}{4\eta^{k,i}} \sum_{t=2}^{T} \|\boldsymbol{p}_t^k - \boldsymbol{p}_{t-1}^k\|_1^2. \tag{65}$$

With the initialization $\hat{p}_1^{k,i} = 1/N$ and $\eta^{k,i} = 2\eta^k$, we obtain

$$\frac{1}{\eta^{k,i}} \ln \frac{1}{\hat{p}_1^{k,i}} + \sum_{i=1}^{N} \frac{\hat{p}_1^{k,i}}{\eta^{k,i}} = \frac{\ln N + 1}{2\eta^k} \leq C_0 (\ln N + 1) = \mathcal{O}(1).$$

Therefore, (65) becomes

$$\sum_{t=1}^{T} \left\langle \boldsymbol{l}_t^k, \boldsymbol{p}_t^k \right\rangle - \sum_{t=1}^{T} l_t^{k,i} \leq \frac{\ln N + 1}{2\eta^k} - 16\eta^k \sum_{t=1}^{T} \sum_{i=1}^{N} p_t^{k,i} \left( l_t^{k,i} - m_t^{k,i} \right)^2$$
$$+ 32\eta^k \sum_{t=1}^{T} \left( l_t^{k,i} - m_t^{k,i} \right)^2 - \frac{C_0}{16} \sum_{t=2}^{T} \|\boldsymbol{p}_t^k - \boldsymbol{p}_{t-1}^k\|_1^2. \tag{66}$$

Combining (64) and (66), we obtain

$$\sum_{t=1}^{T} \langle \boldsymbol{q}_t - \mathbf{e}_k, \boldsymbol{l}_t \rangle + \left\langle \boldsymbol{p}_t^k - \mathbf{e}_i, \boldsymbol{l}_t^k \right\rangle$$

$$\leq \frac{1}{\eta^k} \ln \frac{1}{3(C_0 \eta^k)^2} + 32\eta^k \sum_{t=1}^{T} \left( l_t^{k,i} - m_t^{k,i} \right)^2 - \frac{C_0}{16} \sum_{t=2}^{T} \|\boldsymbol{p}_t^k - \boldsymbol{p}_{t-1}^k\|_1^2 - \frac{C_0}{2} \sum_{t=2}^{T} \|\boldsymbol{q}_t - \boldsymbol{q}_{t-1}\|_1^2$$
$$+ 16\eta^k \sum_{t=1}^{T} \left( \left( l_t^k - m_t^k \right)^2 - \sum_{i=1}^{N} p_t^{k,i} \left( l_t^{k,i} - m_t^{k,i} \right)^2 \right). \tag{67}$$

According to (13) and (14), the last term is bounded by

$$\left( l_t^k - m_t^k \right)^2 - \sum_{i=1}^{N} p_t^{k,i} \left( l_t^{k,i} - m_t^{k,i} \right)^2$$

$$= \left( \langle \nabla f_t(\mathbf{x}_t), \mathbf{x}_t^k \rangle - \left\langle \hat{\boldsymbol{m}}_t^k, \boldsymbol{p}_t^k \right\rangle \right)^2 - \sum_{i=1}^{N} p_t^{k,i} \left( l_t^{k,i} - m_t^{k,i} \right)^2 \tag{68}$$

$$= \left\langle \hat{\boldsymbol{l}}_t^k - \hat{\boldsymbol{m}}_t^k, \boldsymbol{p}_t^k \right\rangle^2 - \sum_{i=1}^{N} p_t^{k,i} \left( l_t^{k,i} - m_t^{k,i} \right)^2 \leq 0$$

where $\hat{\boldsymbol{l}}_t^k = \left( \langle \nabla f_t(\mathbf{x}_t), \mathbf{x}_t^{k,1} \rangle, \cdots, \langle \nabla f_t(\mathbf{x}_t), \mathbf{x}_t^{k,N} \rangle \right)$ and the last step is due to Cauchy-Schwarz inequality. Combining (67) and (68) completes the proof.

## C.6  Proof of Lemma 12

$$\|\mathbf{x}_t - \mathbf{x}_{t-1}\|_2^2 = \left\|\sum_{k=1}^{K}\left(q_t^k \mathbf{x}_t^k - q_{t-1}^k \mathbf{x}_{t-1}^k\right)\right\|_2^2 = \left\|\sum_{k=1}^{K} q_t^k \left(\mathbf{x}_t^k - \mathbf{x}_{t-1}^k\right) + \sum_{k=1}^{K} \mathbf{x}_{t-1}^k \left(q_t^k - q_{t-1}^k\right)\right\|_2^2$$

$$\leq 2\left\|\sum_{k=1}^{K} q_t^k \left(\mathbf{x}_t^k - \mathbf{x}_{t-1}^k\right)\right\|_2^2 + 2\left\|\sum_{k=1}^{K} \mathbf{x}_{t-1}^k \left(q_t^k - q_{t-1}^k\right)\right\|_2^2$$

$$\leq 2\sum_{k=1}^{K} q_t^k \left\|\mathbf{x}_t^k - \mathbf{x}_t^k\right\|_2^2 + 2D^2\|\mathbf{q}_t - \mathbf{q}_{t-1}\|_1^2,$$

where the second step is due to $(a+b)^2 \leq 2a^2 + 2b^2$ and the last step is due to the triangle inequality.

