# OpenReview forum: "Online Composite Optimization Between Stochastic and Adversarial Environments"
_NeurIPS.cc/2024/Conference — NeurIPS 2024 poster_

### Official Review · Reviewer_wJwu · 2024-06-28

**Soundness:** 3
**Presentation:** 4
**Contribution:** 4
**Rating:** 8
**Confidence:** 4

**Summary:**

This paper studies the problem of online composite optimization under the Stochastically Extended Adversarial (SEA) model, where the environments select loss functions in a manner that interpolates between fully stochastic and fully adversarial scenarios. Specifically, the authors establish regret bounds for three kinds of loss functions and introduce a multi-level universal algorithm that can achieve these bounds simultaneously, without prior knowledge of the loss functions. These theoretical guarantees are interesting as they can not only recover previous findings for online composite optimization in both fully stochastic and adversarial environments but also lead to new results, including the first problem-dependent bound for exp-concave cases in the full adversarial setting.

**Strengths:**

1. The presentation is excellent, and the proof is readily comprehensible.

2. The investigated problem of online composite optimization between stochastic and adversarial environments is significant, as it not only bridges two previously separate fields in online composite optimization but also extends the special case without the regularizer.

3. The theoretical results are comprehensive and generalize previous findings in two extreme settings. Moreover, the authors clearly discuss the inadequacies of directly applying the existing work by Scroccaro et al. [2023] in the intermediate setting.

**Weaknesses:**

My only concern is that this article appears to be purely theoretical at present and lacks empirical validation. I believe that some experimental studies could further strengthen the theoretical results.

**Questions:**

For questions please refer to the weaknesses section.

some suggestions:

1. I recommend relocating the Implications Section to follow the Universal Strategy Section. Such reorganization will enhance the logical flow and ensure the continuity of the narrative.

2. Currently, this paper primarily focuses on static regret minimization. I am curious whether we can establish the more general dynamic regret bounds in the investigated setting. I believe that theoretical insights into dynamic regret would also be significant for the field.

**Limitations:**

The authors discuss some limitations & future works in the Conclusion. And there is not potential negative societal impact, since this paper is purely theoretical.

---

> ### Author Rebuttal · Authors · 2024-08-04
>
> **Many thanks for the constructive reviews!**
>
> ---
> **Q1**: I believe that some experimental studies could further strengthen the theoretical results.
>
> **A1**: Thanks for the helpful suggestion! We have conducted empirical studies to verify our theoretical findings. Detailed descriptions can be found in the **General Response**.
>
> ---
> **Q2**: I recommend relocating the Implications section to follow the Universal strategy section.
>
> **A2**: We truly appreciate your advice, and will relocate the Implications section in the revised version to make it more logically fluent.
>
> ---
> **Q3**: I am curious about whether we can establish the more general dynamic regret bounds in the investigated setting.
>
> **A3**: Thanks for the insightful question! Compared to the static regret minimization, minimizing the dynamic regret is more challenging since it needs to build a universal guarantee over any comparators. In the composite SEA, we have also attempted to optimize the dynamic regret. Theoretically, we can achieve a provable dynamic regret bound of $O((\sqrt{\sigma_{1:T}^2} + \sqrt{ \Sigma_{1:T}^2})(P_T+1))$, where $P_T$ denotes the path-length of the comparator sequence, and a tighter $O(P_T + (\sqrt{\sigma_{1:T}^2} + \sqrt{ \Sigma_{1:T}^2})\sqrt{P_T+1})$ dynamic regret bound based on the meta-expert framework. Both two results can also reduce to the purely adversarial and stochastic setting in online composite optimization with the specifications on $\sigma_{1:T}^2$ and $\Sigma_{1:T}^2$. We believe the two results can further broaden the field of composite SEA, and will add them in the extension version.
>
> ---
> We hope that our responses can address your concerns, and we are also happy to provide further clarifications during the reviewer-author discussions if necessary.

---

> > ### Comment · Reviewer_wJwu · 2024-08-10
> >
> > I appreciate the author's rebuttal, and it has fully addressed my concerns. I believe this paper should be accepted, and therefore, I raise my score to 8.

---

> > > ### Author Response · Authors · 2024-08-10
> > >
> > > Many thanks for your kind response and supports! We will revise our paper according to your constructive suggestions.

---

### Official Review · Reviewer_tdoW · 2024-07-10

**Soundness:** 4
**Presentation:** 4
**Contribution:** 3
**Rating:** 7
**Confidence:** 5

**Summary:**

This paper investigates online composite optimization in the regime between stochastic and adversarial environments, establishing theoretical guarantees for three types of time-varying loss functions: smooth and general convex, smooth and strongly convex, and smooth and exp-concave. These bounds not only extend previous results in online composite optimization but also recover those in the special setting without the regularizer. Furthermore, the paper introduces a universal algorithm for online composite optimization that is agnostic to the type of functions and attains optimal bounds for all three cases simultaneously.

**Strengths:**

- The paper is well-organized, with a clear presentation of the background, motivations, and methods. The proofs are also easy to follow.
- The investigation of the intermediate setting between fully stochastic and adversarial environments is an important topic in the community. This paper enriches this line of work, especially in online composite optimization. The regret bounds relate to both stochastic and adversarial aspects of the environments (i.e., $\sigma_{1:T}^2$ and $\Sigma_{1:T}^2$), allowing adaptation to different environments automatically.
- The authors provide intermediate examples and explain the implications of their bounds in these cases, further highlighting the significance of their findings.

**Weaknesses:**

- Although the theoretical results are substantial, the paper lacks experimental validation. It would be more compelling if the authors conducted experiments to validate their theoretical results.
- This paper appears to be an extension of [Sachs et al., 2022; Chen et al., 2023]. The authors should highlight their technical contributions more clearly.

**Questions:**

- The proposed algorithms in this paper are based on optimistic OMD. Is it possible to achieve similar conclusions using optimistic FTRL?
- When the function $f_t(\cdot)$ is non-smooth, can we still apply the proposed algorithms?

**Limitations:**

Yes.

---

> ### Author Rebuttal · Authors · 2024-08-04
>
> **Many thanks for the constructive reviews!**
>
> ---
> **Q1**: Although the theoretical results are substantial, the paper lacks experimental validation.
>
> **A1**: Thanks for the helpful suggestion! We have conducted empirical studies to verify our theoretical findings. Detailed descriptions can be found in the **General Response**.
>
> ---
> **Q2**: The authors should highlight their technical contributions more clearly.
>
> **A2**: Thanks for the valuable suggestion! We have summarized our technical contributions in A1 to Reviewer qCfX. Additionally, we would like to emphasize that although some employed techniques are not entirely new, introducing them to the composite SEA model and modifying them to deliver favorable results remain highly non-trivial.
>
> ---
> **Q3**: Is it possible to achieve similar conclusions using optimistic FTRL?
>
> **A3**: Thanks for the insightful question! We believe that **in the static regret minimization, it is possible to achieve similar theoretical guarantees using optimistic FTRL under the composite SEA model**. The reason lies in that optimistic online learning methods, including optimistic OMD and optimistic FTRL, can yield gradient variation bounds, from which we are able to capture the stochastic and adversarial aspects of the environments with careful analysis. In the literature, optimistic FTRL-based methods, such as CAO-FTRL [Mohri and Yang, 2016], have been shown ensuring the gradient variation bounds in general convex and strongly convex cases of online composite optimization. In the exp-concave case, although no optimistic FTRL-based method currently secures the gradient variation bound, we believe achieving such a guarantee is feasible. Therefore, with *an in-depth technical analysis* on optimistic FTRL-based methods, it is possible to achieve similar theoretical results as those presented in our paper for composite SEA.
>
> However, we would like to emphasize that **in the dynamic regret minimization, our OMD-based methods offer more advantages compared to FTRL-based methods**. As stated in A3 to Reviewer wJwu, our methods can adapt to non-stationary environments and minimize the more general dynamic regret, whereas FTRL-based methods, to the best of our knowledge, have not been proven to ensure the similar theoretical results in online composite optimization. Moreover, we note that in the standard online optimization without the regularizer,  Jacobsen and Cutkosky [2022] have shown that parameter-free variants of FTRL-based methods cannot deliver a dynamic regret bound better than $O(P_T \sqrt{T})$ where $P_T$ reflects the non-stationarity of the environments. This finding partially reveals the limitation of FTRL-based methods in the dynamic regret minimization.
>
> ---
> **Q4**: When the function $f_t(\cdot)$ is non-smooth, can we still apply the proposed algorithms?
>
> **A4**: Below, we discuss the importance of the smoothness in our methods, and the potential modifications  for handling the non-smooth $f_t(x)$.
>
> - **Importance of smoothness.** In our analysis, the importance of the smoothness on $f_t(x)$ lies in decoupling the stochastic and adversarial aspects of the environments from the performance of algorithms. To be more precise, the smoothness is employed to extract $\sigma_{ 1:T }^2$ and $\Sigma_{1:T}^2$, each of which capture the stochastic and adversarial difficulties in the composite SEA model respectively, from the difference between $\nabla f_t(x_t)$ and $M_t$. By choosing the  optimism $M_{t} = \nabla f_{t-1}(x_{t-1})$, such an extraction introduces only an additional stability term, i.e., $\sum_{t=2}^T || x_t  -  x_{t-1} ||_2^2$, which is typically controllable and can be retracted with careful analysis;
>
> - **Modification to non-smoothness.** To handle the non-smooth $f_t(x)$, a classical approach is to apply the implicit update  [Campolongo and Orabona, 2020, Chen and Orabona, 2023], which  replaces the linear approximation of $f_t(x)$ in the update rules with the original function  $f_t(x)$ itself. Following the idea of the implicit update, one possible attempt  is to modify the update rules of OptCMD in the following ways:
> \begin{align}
> \hat{x}\_{t+1}=&\arg\min\_{x\in\mathcal{X}}[\left<\nabla f_t(x\_t),x\right>+r(x)+\mathcal{B}^{\mathcal{R}\_t}(x,\hat{x}\_t)]\\\\x\_{t+1}=&\arg\min\_{x\in\mathcal{X}}[f_t(x)+r(x)+\mathcal{B}^{\mathcal{R}\_{t+1}}(x,\hat{x}\_{t+1})]
> \end{align}
> where $\langle M_{t+1},x\rangle$ is replaced with $f_t(x)$. As mentioned above, the smoothness of $f_t(x)$ is crucial for revealing the adversarial and stochastic aspects of the environments. Therefore, in the non-smooth setting, the primary challenge lies in designing an analysis strategy that can still capture the two aspects of the environments without using the smoothness. This seems highly non-trivial, and we plan to investigate it in future work.
>
> **Reference**.
>
> [1] A. Jacobsen and A. Cutkosky. Parameter-free mirror descent. COLT, 2022.
>
> [2] N. Campolongo and F. Orabona. Temporal variability in implicit online learning. NeurIPS, 2020.
>
> [3] K. Chen and F. Orabona. Generalized implicit follow-the-regularized-leader. ICML, 2023.
>
> ---
> We hope that our responses can address your concerns, and we are also happy to provide further clarifications during the reviewer-author discussions if necessary.

---

### Official Review · Reviewer_qCfX · 2024-07-10

**Soundness:** 3
**Presentation:** 3
**Contribution:** 3
**Rating:** 6
**Confidence:** 3

**Summary:**

This paper investigates online composite optimization within the SEA model. Specifically, it demonstrates that by appropriately adjusting the predictor and step size of the algorithm known as OptCMD, similar regret bounds to those in online optimization within the SEA model can be achieved for general convex functions, strongly convex functions, and exp-concave functions. Additionally, the paper proposes extending the universal algorithm for online optimization by Yan et al. [2023] to online composite optimization.

**Strengths:**

- This paper studies a problem setting considered important for practical applications.
- By making simple adjustments to existing methods, the desired theoretical results are obtained.
- The comparison with existing research is detailed, especially with Scroccaro et al. [2023].
- Overall, it is well-written and easy to follow.

**Weaknesses:**

- The techniques proposed in this paper do not offer significant novelty.

**Questions:**

- If the regularizer $r(\cdot)$ is time-dependent, can the same regret upper bound be achieved with the current analysis? Or are there parts of the analysis that would require non-trivial adjustments?

**Limitations:**

The limitations of this study are discussed in the conclusion.

---

> ### Author Rebuttal · Authors · 2024-08-04
>
> **Many thanks for the constructive reviews!**
>
> ---
> **Q1**: The techniques proposed in this paper do not offer significant novelty.
>
> **A1**: We acknowledge that our research is partially inspired by existing studies, and some techniques employed may not be entirely novel. However, we would like to emphasize that introducing these techniques into the composite SEA model and modifying them to achieve favorable results remain highly non-trivial. Additionally, there also exist unique challenges in our investigated problem (see A1 to Reviewer ZP8k), which necessitate novel technical innovations to address them.
>
> In the following, we summarize our technical novelties.
> - **An in-depth analysis.** In the composite SEA, a straightforward approach is to simply adjust the parameters of OptCMD according to $(6)$, deliver the intermediate result by the original analysis in Scroccaro et al. [2023], and then take the expectations. However, this approach can only ensure much looser bounds compared to ours (see *A straightforward attempt* in A1 to Reviewer ZP8k). To address this issue, we reorganize and dig into the analysis of OptCMD. At the intermediate steps, we isolate the quantities related to $\sigma_{1:T}^2$ and $\Sigma_{1:T}^2$ (see Lines $505$-$508$ for the general convex case), and carefully manage the effect of expectations on them, ultimately achieving a series of favorable theoretical guarantees;
> - **A novel investigation.** Compared to the original study for OptCMD [Scroccaro et al., 2023], we further explore the exp-concave case, and provide an in-depth analysis for OptCMD with the *new* configurations $(10)$. It should be noticed that our result for the exp-concave case is versatile. On the one hand, it can reduce to the existing bound of $O(d \log T/(\alpha T))$ in the fully stochastic environments; on the other hand, it can deliver a tighter regret bound of $O((d/\alpha) \log V_T)$ than existing results in the purely adversarial environments. Detailed discussions can be found in Lines $245$-$250$;
> - **A crafted universal strategy.** Our universal algorithm is based on the two observations: *the time-invariance of $r(x)$* and *the accessibility of expert decisions for the current round*. These observations provide the foundation for utilizing information from $r(x)$ to track the expert performance. Based on them, we carefully design our universal algorithm with the two distinctiveness: (i) using the expert decisions for the current round to update weights; (ii) choosing the multi-level MsMwC with the composite surrogate loss and optimism in $(13)$ and $(14)$ as the meta-algorithm. With rigorous analysis, we demonstrate that our universal algorithm is able to deliver the desired regret bounds for three kinds of functions simultaneously, without prior knowledge of the function type. To the best of our knowledge, this is the *first* universal algorithm that can explicitly support composite loss functions.
> ---
> **Q2**: If the regularizer $r(\cdot)$ is time-dependent, can the same regret upper bound be achieved with the current analysis? Or are there parts of the analysis that would require non-trivial adjustments?
>
> **A2**: Thanks for the insightful question! We notice that in the purely adversary setting, there exist several efforts investigating the composite loss with the time-dependent regularizer [Scroccaro et al., 2023, Hou et al., 2023]. However, under the more general composite SEA model, analyzing the time-dependent regularizer is quite involved, and necessitates a more in-depth exploration. To be precise, we present the following updating rules of OptCMD for  the time-dependent $r_t(x)$:
> \begin{align}
> \hat{x}\_{t+1}=&\arg\min\_{x\in\mathcal{X}}[\left<\nabla f_t(x\_t),x\right>+r_t(x)+\mathcal{B}^{\mathcal{R}\_t}(x,\hat{x}\_t)]\\\\x\_{t+1}=&\arg\min\_{x\in\mathcal{X}}[\left<M_{t+1},x\right>+P_{t+1}(x)+\mathcal{B}^{\mathcal{R}\_{t+1}}(x,\hat{x}\_{t+1})]
> \end{align}
> where $M_{t+1}$ and $P_{t+1}(x)$ denotes the estimations for $\nabla f_{t+1}(\mathbf{x}\_{t+1})$ and $r_{t+1}(x)$ in the next round, respectively. Then, we consider the following two scenarios:
>
> - **The known $r_t(x)$ setting**, in which $r_t(x)$ is determined and known to the learner in advance. In this setting, it is natural to choose $P_{t+1}(x) = r_{t+1}(x)$ at the round $t$. In this way, there is no estimation error on $r_t(x)$, so that we can achieve the same theoretical guarantees as those in our paper;
> - **The unknown $r_t(x)$ setting**, in which $r_t(x)$ is selected by the environments either stochastically, adversarially, or in a manner that interpolates between the two extremes, and is only revealed after the learner submits the decision $x\_t$. In this setting, we can only choose $P_{t+1}(x) = r_{t}(x)$ at the round $t$, and hence, an estimation error on $r_t(x)$, such as $F_T=\sum_{t=1}^T\sup_{x\in\mathcal{X}}|r_{t}(x)-r_{t-1}(x)|$, is inevitably introduced into the final regret bound. Therefore, to capture the variation of $r_t(x)$ in the composite SEA model, it is necessary to incorporate quantities in terms of $r_t(x)$ that capture the adversarial and stochastic aspects of the environments, which will require additional analysis. Furthermore, it should be noticed that $r_t(x)$ is assumed to be non-smooth, and thus, existing analytical techniques on $f_t(x)$ cannot be directly applied to $r_t(x)$, which also demands extra efforts.
>
> We are deeply grateful to the reviewer for raising the insightful question. We believe that investigating the time-dependent $r_t(x)$ will further advance the understanding of composite SEA, and we plan to explore it in our future research.
>
>
> **Reference.**
>
> [1] R. Hou et al. Dynamic regret for online composite optimization. ArXiv, 2023.
>
> ---
> We hope that our responses can address your concerns, and we are also happy to provide further clarifications during the reviewer-author discussions if necessary.

---

> > ### Comment · Reviewer_qCfX · 2024-08-11
> >
> > I appreciate the detailed response.
> >
> > I will keep my score because the other reviews and the responses did not change my opinion that this paper mainly combines existing techniques (e.g., Yan et al. (2023), Sachs et al. (2022), and Chen et al. (2023)) and does not make a significant theoretical contribution.
> >
> > However, I believe that this paper makes a good contribution and is above the bar for acceptance.

---

> > > ### Author Response · Authors · 2024-08-11
> > >
> > > We greatly appreciate your kind response and support! In the revised version, we will provide clearer statements of the technical novelties. Thank you again for your constructive comments!

---

### Official Review · Reviewer_ZP8k · 2024-07-20

**Soundness:** 4
**Presentation:** 3
**Contribution:** 2
**Rating:** 5
**Confidence:** 3

**Summary:**

The authors study the online composite optimization under the Stochastically Extended Adversarial (SEA) model proposed by Sachs et al., 2022. They show that optimistic composite mirror descent (OptCMD), a variant of OMD by [Scroccaro et al., 2023], can achieve regret guarantees that match existing bounds for the SEA model without the regularizer. The bounds are derived for various classes of convex losses including convex functions, exp-concave functions, and strongly convex functions. Furthermore, the authors also propose a universal algorithm that achieves optimal bounds without requiring prior information of the function class.

**Strengths:**

The paper is well-written and ideas are presented clearly. The authors extend the existing algorithm OMD by [Scroccaro et al., 2023] to the composite loss setting. The main novelty lies in the careful selection of the algorithm's parameters that lead to a good regret bound. I have not gone through the technical details very carefully but the results look interesting and extend the known results from noncomposite to composite loss setting.

**Weaknesses:**

While the results look nice, my main concern lies in evaluating the novelty of the paper. I would be interested in knowing the challenges faced by the authors to extend the existing analysis of OMD by [Scroccaro et al., 2023] to the composite setting. Especially, what technical challenges does the new setting throw beyond a different selection of algorithm's parameters? Do these challenges require the authors to come up with any technical innovation in their proofs?

**Questions:**

Please see the above.

**Limitations:**

Please see the above.

---

> ### Author Rebuttal · Authors · 2024-08-04
>
> **Many thanks for the constructive reviews!**
>
> ---
> **Q1**: What technical challenges does the new setting throw beyond a different selection of algorithm's parameters? Do these challenges require the authors to come up with any technical innovation in their proofs?
>
> **A1**: Thanks for the insightful question! We would like to emphasize that since we focus on the expected regret minimization in composite SEA, simply adjusting the algorithm's parameters is insufficient to achieve the desired bounds. In other words, it is also necessary to delve deeper into the analysis and make the corresponding modifications at intermediate steps to carefully deal with the expectation operation. To illustrate this clearly, we take the general convex case as an example.
> - **A straightforward attempt.** One can configure OptCMD with the parameters in $(6)$, obtain the regret bound of $\mathrm{Regret}\_T = O(\sqrt{V_T})$ by the analysis of Scroccaro et al. [2023], and subsequently apply the expectation operation, i.e., $\mathbb{E}[\mathrm{Regret}\_T] = O(\mathbb{E}[\sqrt{V_T}])$ to deliver the expected bound under the composite SEA model. However, it can be verified that such a straightforward approach can only deliver a much looser bound compared to ours;
> - **The underlying reason and our solution.** The underlying reason lies in that the above attempt only takes the expectation after the holistic analysis, neglecting the impact of expectation on intermediate results. To address this issue, we reorganize and dig into the analysis of OptCMD. At the intermediate steps, we isolate the quantities related to $\sigma_{1:T}^2$ and $\Sigma_{1:T}^2$ (see Lines $505$-$508$), and carefully manage the effect of expectations on them, ultimately achieving the desired bounds. Similar strategies are also applied in the strongly convex and exp-concave cases.
>
> Additionally, we also face the following challenges:
>
> - **Analysis for the exp-concave case.** In the original study for OptCMD [Scroccaro et al., 2023], the analysis is limited to the general convex and strongly convex cases. In our paper, we extend the investigation to the *new* exp-concave case, and provide an in-depth analysis for OptCMD with the *new* configurations $(10)$. It should be noticed that our theoretical result for the exp-concave case is versatile. On the one hand, it can reduce to the existing results of $O(d \log T/(\alpha T))$ in the fully stochastic environments; on the other hand, it can deliver a tighter regret bound of $O((d/\alpha) \log V_T)$ than existing results in the purely adversarial environments. Detailed discussions can be found in Lines $245$-$250$;
> - **Design and analysis for the universal strategy.** As we have elaborated in Lines $289$-$290$, the primary challenge is to develop a meta-algorithm capable of tracking experts according to their performance on composite losses, while maintaining the overall regret bound. To address this issue, we propose our universal algorithm, which is based on the two observations: *the time-invariance of $r(\cdot)$* and *the accessibility of expert decisions for the current round*. Consequently, we carefully design our universal algorithm with the two distinctiveness: (i) using the expert decisions for the current round to update the weights; (ii) choosing the multi-level MsMwC as the meta-algorithm and employing the composite surrogate loss and optimism in $(13)$ and $(14)$ to track the expert performance. In analysis, we reveal that although the additional regularization term $r(\cdot)$ is included in the surrogate losses, it can be safely controlled through meticulous analysis, thereby eliminating its impact on the meta-regret. The corresponding analysis can be found in Lines $614$-$617$ for the general convex case, Lines $635$-$643$ for the strongly convex case and Lines $655$-$658$ for the exp-concave case.
>
> It should be emphasized that although some techniques employed in our work may not be entirely novel, introducing them to the composite SEA model and modifying them to deliver favorable results remain highly non-trivial.
>
> ---
> We greatly appreciate your constructive review and helpful feedback, and we will offer more detailed descriptions of the technical challenges and innovations in the revised version. We hope that our responses can address your concerns, and we are also happy to provide further clarifications during the reviewer-author discussions if necessary.

---

### Author Rebuttal · Authors · 2024-08-04

We thank all reviewers for their constructive comments and appreciations of our work. Since both Reviewers tdoW and wJwu  suggest conducting experiments to validate our theoretical results, we present the following general response regarding the empirical studies.

---
**Setup.** In our paper, we show that OptCMD with suitable configurations is able to achieve a series of favorable theoretical guarantees in composite SEA. Moreover, for the practical scenarios where the prior knowledge of loss functions is unavailable, we propose a novel universal strategy, called USC-SEA, which can achieve the desired regret bounds for three cases in composite SEA simultaneously. To verify our theoretical findings, we conduct experiments on the mushroom datasets from the LIBSVM repository [Chang and Lin, 2011], and consider the following online classification problem. At each round $t \in [T]$, the learner receives a data $(\mathbf{x}_t, y_t) \in \mathbb{R}^d \times$ {$-1, 1$} with $d = 112$. Then, the learner plays the decision $\mathbf{w}_t$ from the ball $\mathcal{X}$ with the diameter $D=20$, and suffers a composite loss
$$\phi_t(\mathbf{w}_t; \mathbf{x}_t, y_t) = f_t(\mathbf{w}_t; \mathbf{x}_t, y_t) + \lambda r(\mathbf{w}_t),$$
where we set the hyper-parameter $\lambda = 0.001$. The dataset used in the experiments is considered to be sampled from an unknown distribution, possessing the inherent stochastic property. To simulate the stochastically extended adversarial environments, we perturb the dataset by randomly flipping the labels of $10$% data as the adversarial corruptions. Then, we sequentially pass each data to the learner.

We consider the following three types of loss functions.
- For the general convex case, we choose the smooth and convex cross-entropy function as the time-varying function: $$f_t(\mathbf{w}_t; \mathbf{x}_t, y_t) = -y_t \log \sigma (\mathbf{x}\_{t}^\top \mathbf{w}_t) - (1-y_t) \log (1-\sigma(\mathbf{x}\_{t}^\top \mathbf{w}_t)),$$
where $\sigma(\cdot)$ denotes the sigmoid function, and utilize the $\ell_1$-norm regularizer $r(\mathbf{w}_t) = ||\mathbf{w}_t||_1$. Therefore, the composite function takes the form of:$$
\phi(\mathbf{w}_t; \mathbf{x}_t, y_t) = -y_t \log \sigma(\mathbf{x}\_{t}^\top \mathbf{w}\_t) - (1-y_t) \log (1-\sigma(\mathbf{x}\_{t}^\top \mathbf{w}_t)) + \lambda ||\mathbf{w}_t||_1;$$
- For the strongly convex case, we employ the cross-entropy functions with the $\ell_2$-norm regularizer as the time-varying function: $$f_t(\mathbf{w}_t; \mathbf{x}_t, y_t) = -y_t \log \sigma (\mathbf{x}\_{t}^\top \mathbf{w}_t) - (1-y_t) \log (1-\sigma(\mathbf{x}\_{t}^\top \mathbf{w}_t))+\delta||\mathbf{w}_t||_2^2,$$which is $2\delta$-strongly convex, and still leverage the $\ell_1$-norm regularizer. Hence, the composite loss function is in the form of:$$
\phi(\mathbf{w}_t; \mathbf{x}_t, y_t) = -y_t \log \sigma(\mathbf{x}\_{t}^\top \mathbf{w}\_t) - (1-y_t) \log (1-\sigma(\mathbf{x}\_{t}^\top \mathbf{w}_t))+\delta||\mathbf{w}_t||_2^2+\lambda||\mathbf{w}_t||_1;$$
- For the exp-concave case, we utilize the logistic function as the time-varying function:$$f_t(\mathbf{w}_t; \mathbf{x}_t, y_t) = \log (1 + \exp (-y_t \mathbf{w}_t^\top \mathbf{x}_t)),$$which is exp-concave and smooth [Hazan et al., 2014], and still employ the $\ell_1$-norm regularizer. The composite loss function is shown below:$$\phi(\mathbf{w}_t; \mathbf{x}_t, y_t) = \log (1 + \exp (-y_t \mathbf{w}_t^\top \mathbf{x}_t)) + \lambda ||\mathbf{w}_t||_1.$$

---
**Contenders.** For the general convex and strongly convex cases, we compare our methods with OGD [Zinkevich, 2003], COMID [Duchi et al., 2010] and Optimistic-OMD [Chen et al., 2023]. For the exp-concave case, we choose ONS [Hazan et al., 2007], ProxONS [Yang et al., 2023] and Optimistic-OMD [Chen et al., 2023] as the contenders. All parameters of each method are set according to their theoretical suggestions. For instance, in the general convex case, the learning rate is set as $\eta = ct^{-1/2}$ in OGD, and $\eta = cT^{-1/2}$ in COMID, $\eta_t = D (c + \bar{V}_{t-1})^{-1/2}$ in Optimistic-OMD  where $c$ denotes the hyper-parameter  selected   from {$10^{-3},10^{-2},\cdots,10^{4}$}.

---
**Results.** All experiments are repeated ten times, and we report the instantaneous loss, the cumulative loss and the average loss against the number of rounds in Figure 1 for the general convex case, Figure 2 for the strongly convex case and Figure 3 for the exp-concave case. From the experimental results, it is evident that adversarial corruptions cause considerable fluctuations in the instantaneous loss across different methods. Moreover, we observe that in the three cases of composite SEA, OptCMD incurs lower losses compared to baseline methods, and our universal method, USC-SEA, achieves similar performance to OptCMD without the prior knowledge of loss functions. This phenomenon can be attributed to their ability to adapt to the composite SEA environment, and their explicit support for handling the non-smooth component $r(\cdot)$.

**Reference.**

[1] C.-C. Chang and C.-J. Lin. Libsvm: A library for support vector machines. ACM TIST, 2011.

[2] E. Hazan et al. Logistic regression: Tight bounds for stochastic and online optimization. COLT, 2014.

[3] M. Zinkevich. Online convex programming and generalized infinitesimal gradient ascent. ICML, 2003.

[4] E. Hazan et al. Logarithmic regret algorithms for online convex optimization. FTML, 2007.

---

### Decision · Program_Chairs · 2024-09-25

**Decision:**

Accept (poster)

**Comment:**

This paper provides algorithms for the "stochastically extended adversary" setting in online optimization with composite losses. Happily, the resulting regret bounds are the intuitive result: the same as the non-composite case with the composite term not appearing in the bound.

Overall reviewers are very positive about this work, so it is recommended for acceptance.